# Native nucleosomes intrinsically encode genome organization principles

Sangwoo Park[1], Raquel Merino-Urteaga[2,3,12], Violetta Karwacki-Neisius[2,4,12], Gustavo Ezequiel Carrizo[5,12], Advait Athreya[6,12], Alberto Marin-Gonzalez[2,4], Nils A. Benning[2,3], Jonghan Park[7], Michelle M. Mitchener[8], Natarajan V. Bhanu[9], Benjamin A. Garcia[9], Bin Zhang[10], Tom W. Muir[8], Erika L. Pearce[5,11] & Taekjip Ha[1,2,4 ✉]

The eukaryotic genome is packed into nucleosomes of 147 base pairs around a histone core and is organized into euchromatin and heterochromatin, corresponding to the A and B compartments, respectively[1,2]. Here we investigated whether individual nucleosomes contain sufficient information for 3D genomic organization into compartments, for example, in their biophysical properties. We purified native mononucleosomes to high monodispersity and used physiological concentrations of polyamines to determine their condensability. The chromosomal regions known to partition into A compartments have low condensability and those for B compartments have high condensability. Chromatin polymer simulations using condensability as the only input, without any *trans* factors, reproduced the A/B compartments. Condensability is also strongly anticorrelated with gene expression, particularly near the promoters and in a cell type-dependent manner. Therefore, mononucleosomes have biophysical properties associated with genes being on or off. Comparisons with genetic and epigenetic features indicate that nucleosome condensability is an emergent property, providing a natural axis on which to project the high-dimensional cellular chromatin state. Analysis using various condensing agents or histone modifications and mutations indicates that the genome organization principle encoded into nucleosomes is mostly electrostatic in nature. Polyamine depletion in mouse T cells, resulting from either knocking out or inhibiting ornithine decarboxylase, results in hyperpolarized condensability, indicating that when cells cannot rely on polyamines to translate the biophysical properties of nucleosomes to 3D genome organization, they accentuate condensability contrast, which may explain the dysfunction observed with polyamine deficiency[3–5].

The nuclear genome is largely partitioned into two regions: the gene-rich and relatively open euchromatin and the gene-poor and relatively compact heterochromatin. With the advent of technologies such as Hi-C and chromatin tracing, the complex hierarchal organization of the genome is now being appreciated[1,2]. Each chromosome occupies its own territory in the nucleus; the chromosomes are partitioned into the A and B compartments on a multi-megabase (Mb) scale, and these are further segmented into topologically associated domains (TADs) and loops on a 1-Mb to 10-kilobase (kb) scale. Heterochromatin organization has been explained in terms of chromatin condensation, having either liquid-like[6,7] or gel-like[8] properties. The heterochromatin is AT-rich and has many non-coding repeat sequences, whereas highly transcribing genes usually have low AT content[9]. Histone post-translational modifications (PTMs) and histone variants also reflect the functional state of the chromatin[10].

Although the biological functions of genetic–epigenetic features have mainly been interpreted in the context of interacting partners, such as readers and writers of specific DNA sequences or epigenetic codes[11], their intrinsic physical properties can also have direct biological implications. DNA sequences with high AT content or a long poly(dA:dT) tract can have peculiar groove structures and curvature, which can have special roles in ionic interactions[12–14]. Histone PTMs could be important modulators for determining the intrinsic properties of nucleosomes[15]. Despite extensive knowledge of genome organization, there is little understanding of the biophysical driving force behind genomic compartmentation. In this

[1]Department of Biophysics and Biophysical Chemistry, Johns Hopkins University School of Medicine, Baltimore, MD, USA. [2]Howard Hughes Medical Institute and Program in Cellular and Molecular Medicine, Boston Children's Hospital, Boston, MA, USA. [3]Department of Biology, Johns Hopkins University, Baltimore, MD, USA. [4]Department of Pediatrics, Harvard Medical School, Boston, MA, USA. [5]Department of Oncology, The Bloomberg–Kimmel Institute for Cancer Immunotherapy, Johns Hopkins University School of Medicine, Baltimore, MD, USA. [6]Computational and Systems Biology Program, MIT, Cambridge, MA, USA. [7]College of Medicine, Yonsei University, Seoul, Republic of Korea. [8]Department of Chemistry, Princeton University, Princeton, NJ, USA. [9]Department of Biochemistry and Molecular Biophysics, Washington University School of Medicine St. Louis, St. Louis, MO, USA. [10]Department of Chemistry, MIT, Cambridge, MA, USA. [11]Department of Biochemistry and Molecular Biology, Johns Hopkins Bloomberg School of Public Health, Baltimore, MD, USA. [12]These authors contributed equally: Raquel Merino Urteaga, Violetta Karwacki-Neisius, Gustavo Ezequiel Carrizo, Advait Athreya. ✉e-mail: taekjip.ha@childrens.harvard.edu

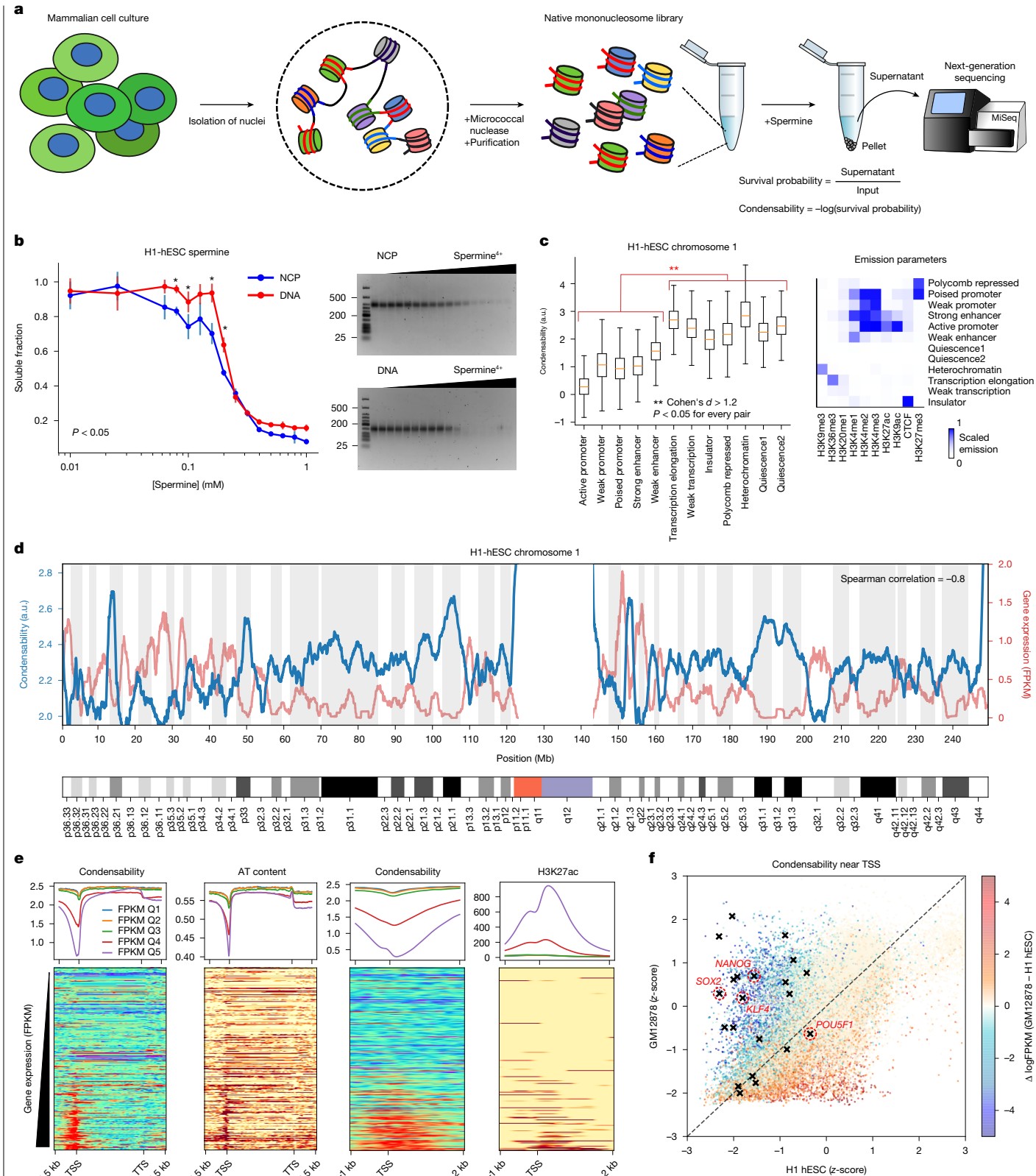

**Fig. 1 | See next page for caption.**

study, we investigate whether nucleosomes intrinsically encode the principles of genome organization, that is, whether individual nucleosomes are sufficient to spontaneously form large-scale organizations, such as the A and B compartments, and local organizations at promoters, enhancers and gene bodies without any chromatin readers, chromatin remodellers or further investment of energy. To address this issue, we developed an assay to measure the intrinsic condensability mediated by physiological condensing agents, and we applied it to human and mouse embryonic stem cells and differentiated cells.

**Fig. 1 | Condense-seq measures genome-wide single-nucleosome condensability. a**, Schematic of the condense-seq workflow. **b**, The total amount of NCP or nucleosomal DNA remaining in the supernatant was measured by ultraviolet–visible (UV–VIS) spectrometry. Left; graph of three biological replicates, error bars denote standard deviation, and the statistical significance of the difference between DNA and NCP is shown as a *P* value, obtained by two-sided Welch's *t*-test, marked with an asterisk: 0.0034, 0.06, 0.007 and 0.013, respectively. Right, their integrity was checked by 2% agarose gels; lane 1 is a low-molecular-weight DNA ladder, and other lanes are supernatant nucleosomes or nucleosomal DNA after condensation with various spermine concentrations. **c**, Genome segmentation into chromatin states based on histone PTM ChIP-seq data (right). All mononucleosomes of chromosome 1 were categorized and their condensability distribution for each chromatin state is shown (boxplot in which the centre is the median and the lower and upper bounds are the first and third quartiles, respectively). The *P* values were computed using two-sided Welch's *t*-test comparing the condensabilities between chromatin states. Cohen's *d* metric denotes the effect-size comparison over more than 7,000 nucleosomes for each state from two biological replicates (also shown in Extended Data Fig. 2i). **d**, RNA-seq data (red) and condensability (blue) over the entire chromosome 1 (Spearman correlation is −0.8 in 100-kb bins); positions are given in Mb. **e**, All genes were grouped into five quantiles according to the transcription level (quantiles 1–5 (Q1–Q5), in order of increasing transcription). Top, condensability, AT content and H3K27ac level along the transcription unit coordinate averaged for each quantile. Bottom, heat maps show the same quantities for each gene, rank ordered by increasing gene expression. **f**, Promoter condensability (averaged over a 5-kb window around the TSS) for H1-hESC and GM12878. Each gene is coloured according to its relative expression level in the two cell types. Black symbols indicate embryonic stem cell marker genes. FPKM, fragments per kilobase of transcript per million mapped reads; a.u., arbitrary units. Illustration in **a** created in BioRender (Park, S. (2025) https://BioRender.com/q73ofz1).

## Condense-seq of native mononucleosomes

We used various DNA- and nucleosome-condensing agents, including polyamines[16], cobalt hexamine[17], polyethylene glycol (PEG)[18], calcium[19], heterochromatin protein 1α (HP1α) and heterochromatin protein 1β (HP1β)[20] to induce condensation of native nucleosomes in vitro. Native mononucleosomes were prepared by hydroxy apatite purification after in-nuclei micrococcal nuclease digestion of the chromatin, followed by size selection to obtain monodisperse samples (Fig. 1a and Extended Data Figs. 1a–e and 2c). The nucleosome condensation experiment was first performed using various concentrations of spermine as a condensing agent (Fig. 1b). Spermine is a small biological metabolite and a prevalent polyamine in eukaryote nuclei[21]. We showed that native mononucleosomes remain intact after condensation, and we used single-molecule fluorescence resonance energy transfer[22] (FRET) to show that spermine, at concentrations that induce the formation of large nucleosome condensates, does not induce detectable unwrapping of nucleosomal DNA (Extended Data Fig. 1f–h). By sequencing the nucleosomes remaining in the supernatant and comparing them with the input control, each nucleosome could be localized along the genome and its survival probability after condensation could be estimated (Extended Data Fig. 2a,b). We defined 'condensability' (the propensity to be incorporated into the precipitate) as the negative natural log of the survival probability (Fig. 1a). Using this 'condense-seq' assay, we could determine genome-wide condensability at single-nucleosome resolution. We also validated that our condensability metric is indeed a measure tightly associated with how many nucleosomes survived in the supernatant after condensation, by showing that the nucleosome counts in the supernatant, not those of the input, are mainly responsible for the condensability contrast (Extended Data Fig. 2d–g). We also checked the reproducibility and robustness against the choice of nucleosome peak calling methods (Extended Data Fig. 2e,j,h).

## Condensability and gene expression

Chromosome-wide condensability maps for H1 human embryonic stem cells (H1-hESCs) are shown in Fig. 1d and Extended Data Fig. 3a,b. At a resolution of 1 Mb, condensability varies from 2 to 3, and it greatly increases in the subtelomeric and pericentromeric regions. Gene expression, as assessed by RNA-seq[23], shows a clear anticorrelation with condensability (a Spearman correlation of −0.8). At a much finer scale, condensability around the transcription start site (TSS) is the lowest for the most highly expressed genes and highest for those expressed least (Fig. 1e). These findings are surprising because they indicate that single native nucleosomes isolated from the cell have biophysical properties, high or low condensability, that are associated with low and high transcription, respectively, even though condensability was determined in vitro in the absence of any other factors normally present in vivo. Other features, such as AT content, CpG methylation density and levels of H3K9ac, H3K27ac and H3K4me3, were also dependent on gene expression, but individually they were poor predictors of condensability profiles across the promoter region (Fig. 1e and Extended Data Fig. 3c). For example, although AT content is also the lowest around the TSS in genes with the highest expression, its dip is approximately two-fold narrower than the condensability dip (Fig. 1e). Another example is H3K27ac, which, although stronger in highly expressed genes, does not match well with condensability in either width or rank order (Fig. 1e). Notably, even in highly expressed genes, condensability quickly increases as we examine regions farther away from the TSS and into the gene body (Fig. 1e).

Next, we used ChromHMM[24] to segment the genome into 12 chromatin states on the basis of histone modifications and observed differences in condensability depending on the chromatin state (Fig. 1c and Extended Data Fig. 2i). Promoters and enhancers show the lowest condensability, whereas heterochromatin, gene body, Polycomb repressed and quiescence state regions show the highest condensability. Furthermore, strength dependence was observed, with strong promoters and enhancers showing lower condensability than do weak promoters and enhancers. Overall, transcriptionally active chromatin states show low condensability compared with inactive states, with one exception: the gene body shows high condensability, and this is true even in highly expressed genes, as noted earlier (Fig. 1c).

In the genome browser view of an approximately 40-kb window of human chromosome 1 (Extended Data Fig. 3b), condensability obtained from H1-hESCs has two main minima approximately 2 kb in width and overlapping with *cis*-regulatory regions, a promoter and an enhancer. The depth of the minima is approximately two in natural log scale, indicating that the nucleosomes there are about 7.3 times, $e^2$, less condensable than average nucleosomes in probabilistic metric. Both overlapped with CpG islands and also with Dnase I hypersensitivity peaks, but these are much narrower than the condensability dips.

We next tested the possibility that the condensability contrast is driven mainly by AT content[14], and is therefore independent of cell type or cellular state, by performing condense-seq for a differentiated cell type, GM12878 (Extended Data Fig. 7). Condensability in the 5-kb region surrounding the TSSs of all annotated genes shows wide variations between the two cell types (Fig. 1f). Importantly, genes with higher expression in the differentiated cell (GM12878) than in the embryonic stem cell (H1-hESC) show lower condensability in the differentiated cell than in the embryonic stem cell. Therefore, condensability of the promoter region is cell type-dependent, excluding the possibility that cell type-independent features, such as AT content, are the primary determinant of promoter condensability. Notably, embryonic stem cell markers, such as *NANOG*, *SOX2* and *KLF4*, have promoter regions that are much less condensable in the embryonic stem cell than in the differentiated cell (Fig. 1f).

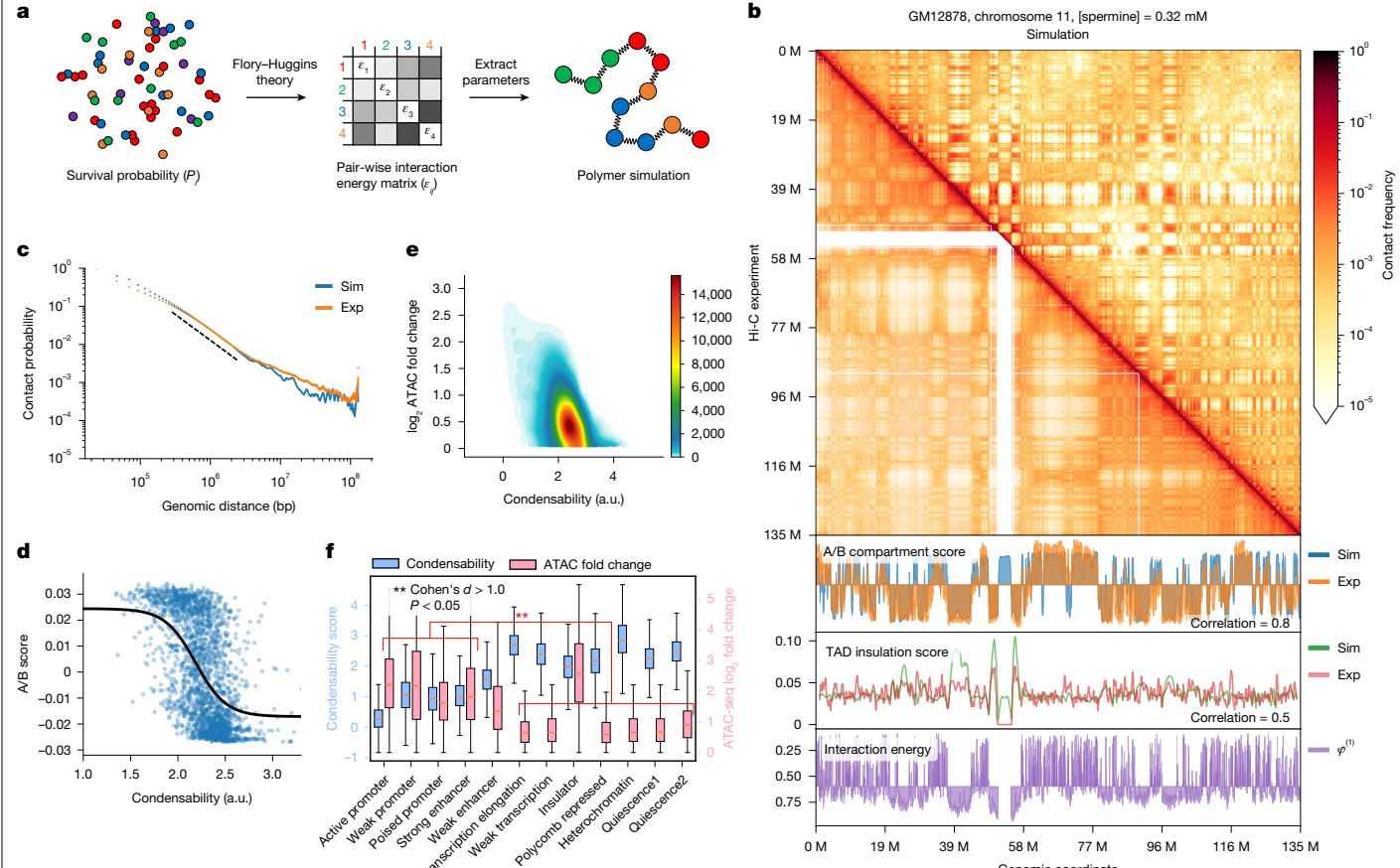

**Fig. 2 | 3D genome compartmentalization information is encoded in native mononucleosomes. a**, Nucleosome–nucleosome pair-wise interaction energies ($\varepsilon_{ij}$) were derived from the condense-seq measurements according to the Flory–Huggins theory. The chromatin polymer simulation was done using these interaction energies to predict the 3D chromatin structure solely from the nucleosome condensability. **b**, Comparison of contact probability matrix between the Hi-C data of GM12878 (lower-left triangle) and the polymer simulation (upper-right triangle). Bottom, the A/B compartment scores were computed using the Hi-C data or polymer simulation with interaction energies based on the condensability ($\varphi$). TAD insulation scores were also computed for the Hi-C data and polymer simulation. Pearson correlations between simulation (Sim) versus experimental (Exp) values are shown (0.8 for A/B compartment score and 0.5 for TAD insulation score comparison). **c**, Contact

probability versus genomic distance from the Hi-C experimental data (orange) and a polymer simulation (blue). The scale factor of exponential fitting is: simulation, $a = 1.2$; experimental, $a = 1.1$. **d**, A/B compartment score versus condensability in 100-kb bins. The black line is a logistic curve fit. **e**, Condensability versus chromatin accessibility (ATAC-seq fold change) in 1-kb bins (the colour bar represents the number of 1-kb bins in the 2D density plot with $20 \times 20$ bins). Spearman correlation = −0.46. **f**, Condensability and ATAC score versus ChromHMM chromatin state for chromosome 1. In the boxplots, the centre is the median and the lower and upper bounds are the first and third quartiles, respectively; $P$ values were computed using a two-sided Welch's $t$-test for comparing chromatin openness in different chromatin states; Cohen's $d$ was calculated for comparing the effect size over more than 100,000 genomic bins for each state from two biological replicates. a.u., arbitrary units.

We also applied condense-seq to mouse embryonic stem cells at embryonic day 14 (E14 mESCs) and found similar results, including the dependence of condensability on chromatin states, an anticorrelation between condensability and gene expression, and cell-type specificity (Extended Data Fig. 3d–g).

## Nucleosomes encode for A/B compartments

The chromosome-wide anticorrelation between condensability and gene expression raised the possibility that nucleosome condensability is closely associated with euchromatin or heterochromatin compartmentalization. We compared the condensability profile with the A/B compartment score obtained from the H1-hESC Micro-C data[23]. We observed a clear anticorrelation between the condensability and the A/B compartment score on the chromosome-wide Mb scale (Extended Data Fig. 4a) and on the 100-kb scale (Fig. 2d). At the finer scale of TADs and their boundaries that are determined by transacting factors such as cohesins and CTCF[25], the correlation between the experimental TAD insulation score and the predicted score based on condensability was

understandably weaker (Fig. 2b and Extended Data Fig. 4b). Genomic accessibility measured by ATAC-seq[26] also showed an anticorrelation with condensability, in which more-accessible or opened genomic regions were less condensable than less-accessible ones (Fig. 2e). This inverse relationship between chromatin openness and condensability was even more pronounced when compared across chromatin states (Fig. 2f and Extended Data Fig. 4c,d).

We also showed that the in silico chromatin polymer simulation of a human chromosome with pair-wise interaction energies derived from condensability alone as an input (Fig. 2a) can faithfully reproduce A/B compartments from the Hi-C data (Pearson correlation coefficient of 0.8 for GM12878) (Fig. 2b,c). This spatial segregation probably results from the exclusion of less-condensable chromatin from the compacted highly condensable core, and this is reminiscent of the inverted chromatin organization of rod photoreceptors[27]. Indeed, when AT-rich DNA and GC-rich DNA are co-condensed in the presence of spermine, they spontaneously form a spatially segregated structure in which an AT-rich DNA core is surrounded with GC-rich DNA, probably because of their differential condensabilities[14] (Extended Data Fig. 4e,f). Together,

our results imply that the native mononucleosomes intrinsically have, even in the absence of other factors, many of the biophysical properties needed for the large-scale A/B compartmentalization (around 80% in the case of GM12878 cells).

## Genetic and epigenetic basis

Next, we sought to identify the genetic and epigenetic features that determine nucleosome condensability. We observed a good correlation between the condensability and the AT content (Extended Data Fig. 5a), reminiscent of stronger polyamine-induced attractive interactions between AT-rich DNA compared with GC-rich DNA of the same length[14]. No significant correlation was found between condensability and dinucleotide periodicity associated with the rotational phasing of nucleosomal DNA[28] and extreme DNA cyclizability[29] (Extended Data Fig. 5b,c), which indicates that there are distinct biophysical mechanisms of nucleosome stability and condensability.

By analysing DNA methylation and histone chromatin immunoprecipitation followed by sequencing (ChIP-seq) data for H1-hESC in the Encyclopedia of DNA Elements (ENCODE) data portal[30], we investigated epigenetic features associated with nucleosome condensability (Extended Data Fig. 5d). Epigenetic marks associated with transcriptional activation were highly enriched in low-condensability partitions, with the lone exception of H3K36me3. Repressive epigenetic marks, such as H3K9me3 and CpG methylation density, were more enriched in high-condensability partitions. However, some of the other repressive marks, such as H3K27me3 and H3K23me2, were enriched in the least-condensable fraction (Extended Data Fig. 5d), potentially owing to confounding effects from poised promoters prevalent in embryonic stem cells, which simultaneously have both active and inactive marks, such as H3K27ac and H3K27me3, respectively[31], or bivalent promoters in the case of H3K23me2 (ref. 32). To reduce the confounding effects of diverse features occurring simultaneously in some nucleosomes, we stratified the data into subgroups that shared all features except one for comparison with condensability. This conditional correlation analysis showed that high condensability was the most strongly correlated with AT content, H3K36me and H3K9me3 (Fig. 3b). Low condensability was strongly correlated with histone acetylation in general and with *H2AFZ*, H3K4me1, H3K4me2, H3K4me3, H3K79me1 and H3K79me2. Machine-learning-based modelling also predicted the nucleosome condensability based on those genetic and epigenetic components as input with similar importance (Extended Data Fig. 5f–h).

We also used bottom-up mass spectrometry to identify histone PTMs enriched in supernatant/pellet/input native nucleosome samples before and after condensation by spermine (Extended Data Fig. 6a–c). By counting histone H3 and H4 peptides containing PTMs, we computed the enrichment of PTMs in the supernatant and compared them with unmodified peptides as the control (Extended Data Fig. 6a,b). Consistent with the genomic analysis based on ChIP-seq data, we found that the supernatant was depleted of repressive marks such as H3K9me3 and was strongly enriched in most of the acetylation marks, especially poly-acetylation marks. The H3K27 and H3K36 methylation marks did not show either clear enrichment or depletion, similar to the condense-seq analysis.

To investigate more directly how histone PTMs affect nucleosome condensation without contributions from the DNA sequence or cytosine methylation, we used a synthetic nucleosome PTM library formed on identical Widom 601 DNA sequences[33]. By performing condense-seq and demultiplexing using the appended barcodes, we obtained the condensability change for each PTM mark compared with controls that did not have any PTM marks (Fig. 3e). All single modifications, except for phosphorylation, showed a decrease in condensability relative to the unmodified control (Fig. 3d). Ubiquitylation was the most effective in making nucleosomes less condensable, followed by acetylation, crotonylation and methylation, in that order. The intrinsic solubilizing

effect of ubiquitin-like proteins has previously been demonstrated for SUMO[34]. Electrostatic interaction is a key determinant, as shown by the strong impact of acetylation and crotonylation, which add negative charges that would require more polyamines to neutralize the net negatively charged nucleosomes during condensation. Acetylation on histone tails has a much stronger effect than acetylation on the histone fold domain (Fig. 3d), having the strongest effect on the H4 tail, followed by the H2A, H2B and H3 tails, respectively. The H2A.Z variant showed significantly reduced condensability compared with the canonical histones (Fig. 3e), which is consistent with the conditional correlation analysis (Fig. 3b) and also with previous reports that H2A.Z makes oligonucleosomes more soluble, potentially owing to the different acidic patch structure of the variant[35,36]. A linear regression model trained on only the PTM library condensability data could qualitatively predict genomic nucleosome condensability (Extended Data Fig. 6d–f).

Next, to examine the effects of genomic DNA sequences on nucleosome condensation, we synthesized a 'reconstituted' nucleosome library composed of genomic nucleosomal DNA purified from GM12878 cells reconstituted with recombinant canonical histone octamers that were devoid of PTMs (Extended Data Fig. 7a,b). Remarkably, the reconstituted nucleosomes showed higher condensability overall compared with native nucleosomes (Extended Data Fig. 7c) and lost the chromatin state dependence (Extended Data Fig. 7d). They also lost the correlation with gene expression on a genome-wide scale (Extended Data Fig. 7e) and for individual genes near the TSS (Extended Data Fig. 7f). These results show the primary importance of histone PTMs for determining genomic nucleosome condensability.

In the cellular context, because genomic nucleosomes are decorated with the combinations of multiple PTMs and cytosine methylation in different sequence contexts, as shown in non-negative matrix factorization (NMF) clustering (Extended Data Fig. 5e), nucleosome condensation properties are likely to be a complex emergent outcome of the combined effects of the individual genetic and epigenetic features. If so, we may conclude that nucleosome condensability is a natural axis onto which to project the high-dimensional cellular chromatin state. We view condense-seq as a readily adoptable tool for studying functional genome organization in a variety of contexts.

## 3D genome through electrostatics

Polyamines are thought to induce condensation of DNA and nucleosomes by making ion bridges between negatively charged DNA[16]. If such charge–charge interactions are a major driving force, other ionic condensing agents should also induce condensation. We performed condense-seq on H1-hESC mononucleosomes using spermidine, cobalt hexamine, magnesium ions and calcium ions, as well as PEG (Extended Data Fig. 8a). For all condensing agents, chromosome-wide Mb-scale condensation profiles were anticorrelated with gene expression, and all ionic condensing agents showed good correlations with each other in terms of condensability at the 10-kb scale, except for calcium, which condensed mononucleosomes poorly (Fig. 3a and Extended Data Fig. 8b). Similarly, all ionic condensing agents also showed very strong correlations for condensation of the synthetic PTM library (Extended Data Fig. 9a–c,f). Intriguingly, charge-swap mutations on the acidic patch on histone H2A/B, which was previously suggested to be the nucleosome–nucleosome interaction interface[37], induced the largest condensability increase among PTM library members for all ionic condensing agents (Fig. 3e). Thus, this trend, combined with our observation that polymer simulations using nucleosome condensability as the sole input can predict A/B compartments (Fig. 2), further points to the electrostatic interaction between nucleosomes mediated by multivalent ions as a major driving force for large-scale genomic compartmentalization (see Supplementary Note 4 for further discussion).

Next, we performed condense-seq on H1-hESC nucleosomes using HP1α and HP1β proteins as condensing agents (Extended Data Fig. 8a).

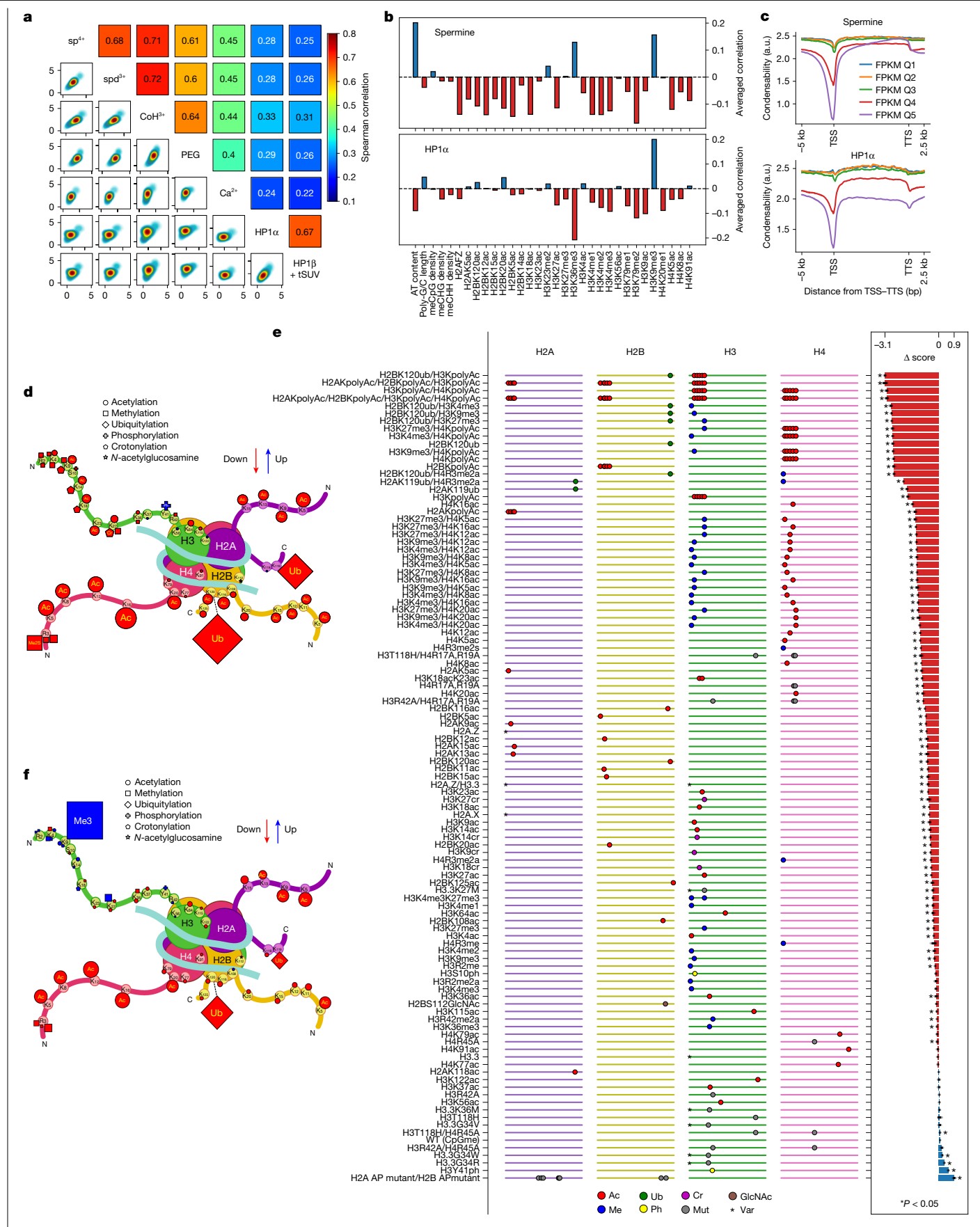

**Fig. 3 |** See next page for caption.

**Fig. 3 | Identification of the biophysical driving force of chromatin condensation and its genetic and epigenetic determinants. a**, Correlation of condensability scores for the condensing agents tested: spermine (sp$^{4+}$), spermidine (spd$^{3+}$), cobalt hexamine (CoH$^{3+}$), polyethylene glycol (molecular weight 8,000; PEG), Ca$^{2+}$, HP1α and HP1β/tSUV39H1 (HP1β + tSUV). **b**, Conditional correlations between condensability and various genetic and epigenetic factors for spermine (top) and HP1α (bottom). **c**, Condensability profiles versus gene unit position averaged over each of the five quantiles, from weakly expressed to highly expressed genes for spermine (top) and HP1α (bottom). **d–f**, Condense-seq results of the PTM library. The effects of single PTMs on nucleosome condensation are depicted in the cartoon structures for spermine (**d**) and HP1α (**f**). Each symbol represents a PTM of a specific type, as shown in the key, and its size is proportional to the strength of the effects. The colours of the marks indicate the direction of the effect (red, decrease condensability;

blue, increase condensability) compared with the unmodified control. All condensability scores of the PTM library using spermine as a condensing agent are shown (**e**). The library members were sorted from the lowest to the highest condensability scores from top to bottom. Left, the ladder-like lines represent each histone peptide from the N terminus (left) to the C terminus (right). Each mark on the line indicates the location of PTMs, and the shape of the marks represent the PTM type (Ac, acetylation; Me, methylation; Cr, crotonylation; Ub, ubiquitylation; Ph, phosphorylation; GlcNAc, GlcNAcylation; Mut, amino acid mutation; Var, histone variant). Right, the change in condensability scores of the various modified nucleosomes compared with the control nucleosomes without any PTMs is shown as a bar plot. Asterisks indicate statistical significance ($P < 0.05$, two-sided Welch's $t$-test used over three independent biological replicates) compared with the wild-type control. a.u., arbitrary units.

On the Mb scale, the chromosome-wide condensability profile was anti-correlated with gene expression, as in the case of ionic agents (Extended Data Fig. 8b). However, on the 10-kb scale, the condensability results for the ionic agents versus heterochromatin protein 1 (HP1) did not show good correlations (Fig. 3a). Using previously annotated data, we quantified the correlations between condensability and various markers of nuclear subcompartments: the lamina-associated domain (LAD)[30], nucleolar-associated domain (NAD)[38] and speckle-associated domain (SPAD)[39] (Extended Data Fig. 8c). For all condensing agents, condensability is strongly anticorrelated with nuclear speckle and transcription markers and weakly anticorrelated with Polycomb markers. Heterochromatin, nucleolar-associated and lamin-associated marks show a positive correlation with condensability, with the strongest correlation being observed between HP1-mediated condensability and the H3K9me3 marks. Differences between the ionic agents and HP1s were further identified in the ChromHMM genome segmentation; condensability is low at promoters and enhancers for all condensing agents, but the magnitude of this effect is much reduced for HP1 (Extended Data Fig. 8d). Interestingly, the gene body showed low condensability with HP1, in contrast to the high condensability with the ionic agents. Consistently, the condensability profile of HP1α from TSS to transcription termination site (TTS) also showed reduced condensability in highly expressed genes, not only near the TSS, but also along the gene body (Fig. 3c). Conditional correlations also revealed that condensability with HP1α is negatively correlated with H3K36me3 and positively correlated with H3K9me3 (Fig. 3b).

We also performed condense-seq on the PTM library using HP1α as the condensing agent. H3K9me3 profoundly increased nucleosome condensation by HP1α (Fig. 3f and Extended Data Fig. 9d), which is consistent with HP1α's role as an H3K9me3 heterochromatin mark reader[40,41]. Interestingly, regardless of PTM type, most PTMs on the H3 tail also showed a slight increase in HP1-induced condensation, and this trend was stronger at locations farther from the nucleosome core. This finding might indicate that HP1α could also recognize other PTMs on the H3 tail in a nonspecific manner, and/or that these H3 tail modifications may also affect nucleosome dynamics, thereby indirectly influencing interactions with HP1α[15]. Apart from the H3 tail modifications, most PTMs showed similar effects between HP1α and ionic agents, reducing condensability.

## Polyamine loss causes hyperpolarization

Polyamines are one of the most prevalent biological metabolites[21]. We performed condense-seq on mouse T cells, the activation and differentiation of which are crucially impacted by polyamines[3]. We isolated and activated CD8$^+$ T cells from control mice and mice with a T cell-specific knockout (KO) of ornithine decarboxylase (ODC) (Fig. 4b), which is a rate-limiting enzyme for polyamine synthesis, converting ornithine to putrescine, which can then be further metabolized to spermidine

and spermine (Fig. 4a). We also examined wild-type mouse CD8$^+$ T cells treated with difluoromethylornithine (DFMO), which is a chemical inhibitor of ODC[42]. For all three (control, *Odc* KO and +DFMO), native nucleosomes were purified and subjected to condense-seq with spermine (Fig. 4b and Extended Data Fig. 10a).

To enable a quantitative analysis of subtle differences across different conditions, we used another metric, condensation point ($c_{1/2}$), a spermine concentration at which the soluble fraction is half the input (Extended Data Fig. 10b). Thus, $c_{1/2}$ is inversely correlated with the previously defined condensability score (Extended Data Fig. 10i). The $c_{1/2}$ values of nucleosomes have a higher dynamic range in *Odc* KO and +DFMO cells than in wild-type cells (Extended Data Fig. 10c–h), such that disrupting polyamine synthesis seems to amplify the contrast, in which highly condensable nucleosomes become even more condensable, and poorly condensable nucleosomes become even less condensable (Fig. 4d). We propose that when cells cannot rely on endogenous polyamines to bring together more-condensable nucleosomes to form B compartments or to induce promoter condensation, they modify the nucleosomes to accentuate the condensability contrast. That is, following polyamine depletion, nucleosomes with biophysical properties associated with high condensability acquire changes to make their condensability even higher, and those with low condensability even lower. In support of this suggestion, similar trends of hyperpolarization were observed for individual nucleosomes that were categorized into different chromatin states (Fig. 4c), as well as in the condensability profiles of genes grouped into different quantiles according to their gene expression levels (Fig. 4e).

To investigate the possible local, gene-specific changes following polyamine depletion, we standardized the condensability score across different conditions using the $z$-score. ODC inhibition and *Odc*-KO induced $z$-score changes in single genes, $\Delta z$, are correlated between the two conditions (a Spearman's correlation coefficient of 0.6) (Extended Data Fig. 10j). Among the chromatin states, active and poised promoters were the most affected, showing the largest changes of $z$-scores in condensability following polyamine depletion (Extended Data Fig. 10k). Gene set enrichment analysis[43] showed that many T cell activation and other immune signalling processes were enriched among genes that showed significant increases in condensability, but a variety of developmental and differentiation processes were enriched among genes that showed significant reduction in condensability following *Odc* KO (Fig. 4f) or ODC inhibition (Fig. 4g). Development-related genes, which are repressed through H3K27me3 (ref. 44), were particularly strongly affected by *Odc* KO, and indeed, genes with the largest decreases in $z$-score of the promoter condensability following *Odc* KO (quintile 1; Fig. 4h) showed the greatest enrichment of H3K27me3 (Fig. 4h) in the wild type. The importance of the H3K27me3 mark was validated by a histone PTM immunostaining screen using flow cytometry that showed a global increase in H3K27me3 in *Odc*-KO CD8$^+$ T cells, which also showed a global increase in H3K36me3 (Extended Data Fig. 10l). This was further analysed by

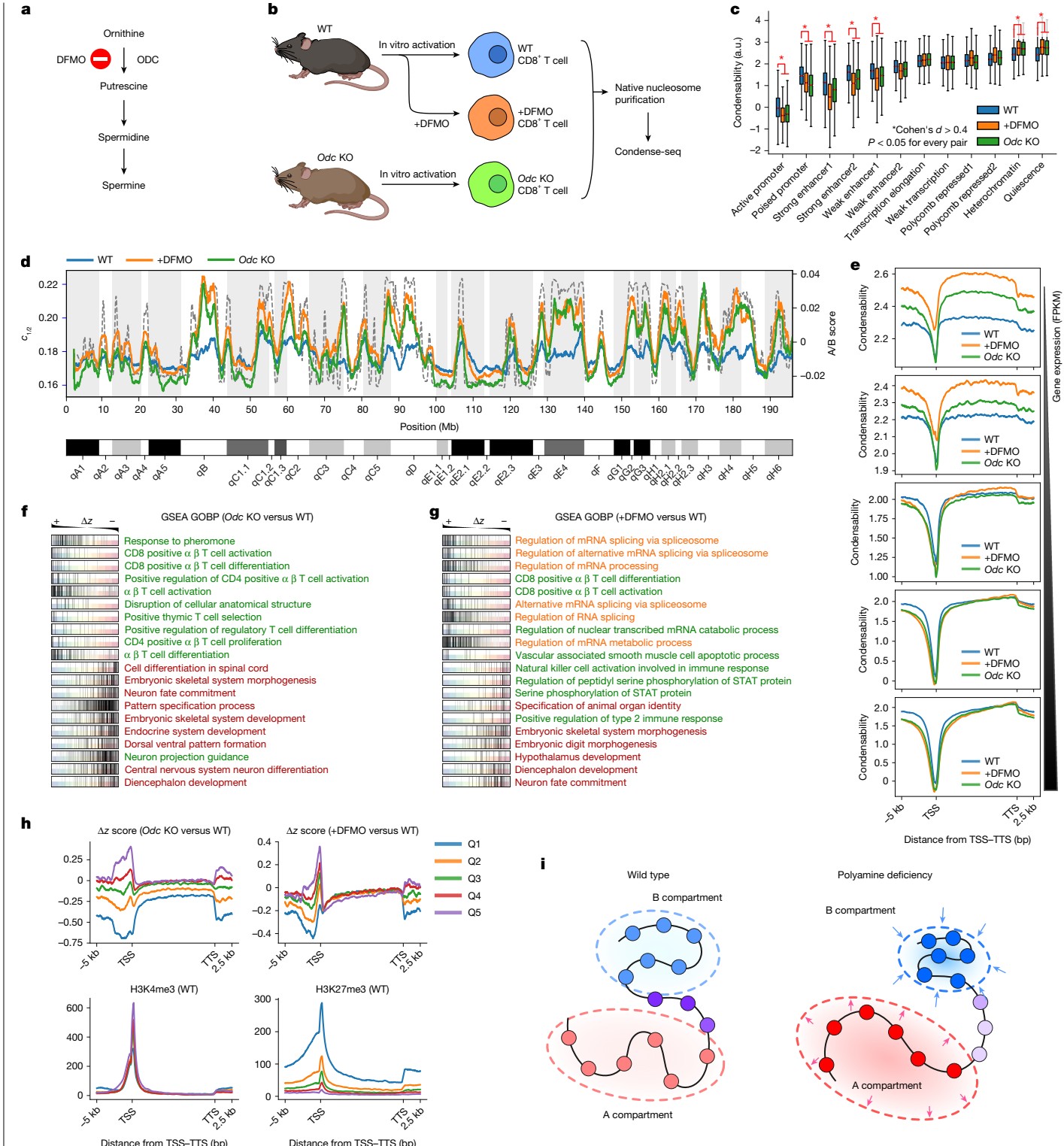

**Fig. 4 | See next page for caption.**

calibrated ChIP-seq experiments, which showed a small but significant increase in H3K27me3 following DFMO treatment, particularly at active chromatin regions, whereas H3K27ac levels were almost unchanged, with very slight decreases in heterochromatin regions (Extended Data Fig. 10m,n). Together, our results show that polyamine deficiency not only globally hyperpolarizes genome compartmentalization, making nucleosomes in B compartments and poorly expressed gene promoters more condensable and nucleosomes in A compartments and highly

expressed gene promoters less condensable, but also causes local chromatin disorganization, especially in developmental genes, which could lead to problems with cell differentiation (Fig. 4i).

## Discussion

Our results indicate that biophysical information that is important in large-scale organizations, such as A/B compartments, and in local

organizations, at promoters and enhancers, is electrostatically encoded in native nucleosome core particles. By showing that connectivity is not essential for heterochromatin-associated nucleosomes to condense more readily than euchromatin-associated nucleosomes, our data are synergistic with studies showing that 30-nm fibres do not form in cells[45]. Even when more-specific interactions between chromatin and proteins, such as HP1, Polycomb repressive complex, cohesin and CTCF, and other non-coding RNAs, are responsible for smaller-scale, function-directed chromosome organization, the intrinsic condensability of individual nucleosomes forms a biophysical backdrop that must be taken into consideration (Extended Data Fig. 8e).

**Fig. 4 | Polyamine deficiency globally hyperpolarizes but locally disorganizes chromatin condensability. a**, ODC is a key enzyme in polyamine biogenesis and is inhibited by DFMO. **b**, Mouse CD8[+] T cells were isolated and activated in vitro before condense-seq. **c**, Mononucleosome condensability distribution in various chromatin states classified using ChromHMM. The statistical significance ($P$ value) of the difference between polyamine-deficient conditions versus wild type was computed using two-sided Welch's $t$-test, and the effect size, Cohen's $d$, over more than 2,000 nucleosomes for each state from two biological replicates, was also computed for comparison. **d**, Condensation point ($c_{1/2}$) for chromosome 1 for +DFMO and *Odc* KO (solid lines show condensability; the dotted line shows the A/B score). **e**, Condensability over gene units averaged over genes belonging to five quantiles of gene expression. **f**,**g**, Gene set enrichment analysis (GSEA) of polyamine-deficient conditions *Odc* KO (**f**) and +DFMO (**g**) compared with the wild type. Genes were ordered by $\Delta z$, the $z$-score of condensability relative to the wild type, shown above. Each row corresponds to the Gene Ontology (GO) biological process (GOBP)

strongly enriched for strongly positive or strongly negative $\Delta z$ values, and genes belonging to that gene set are localized by tick marks. The top 10 positively and negatively enriched GO biological processes are shown. The enriched GO biological processes are clustered by their biological function (red, developmental; green, T cell activation and immunity; orange, mRNA splicing related). **h**, For each quantile of $\Delta z$ near the TSS (Q1–Q5), averaged $\Delta z$ versus transcription unit position is shown for *Odc* KO versus wild type (top left) and +DFMO versus wild type (top right), and averaged ChIP-seq signals in the wild type are shown for H3K4me3 (bottom left) and H3K27me3 (bottom right). **i**, Polyamine deficiency induces global hyperpolarization of chromatin compartmentalization but disrupts local chromatin organization (darker colours and the arrows shown for the polyamide deficiency condition depict hyperpolarized compartments), especially at genomic loci enriched with H3K27me3 marks. a.u., arbitrary units. Illustration in **b** created in BioRender (Park, S. (2025) https://BioRender.com/q73ofz1).

The differences in nucleosome condensability between H1-hESC and GM12878 show how compartmentalization changes after cellular differentiation; the genome-wide condensability in GM12878 shows the higher dynamic range and better correlation with A/B compartment scores (Extended Data Fig. 7d,e). Furthermore, the condensability near TSSs decreased deeply and widely, even affecting the gene body of highly transcribing genes of GM12878 (Extended Data Fig. 7f), whereas condensability on the gene body of H1-hESC is consistently high, regardless of gene expression level (Fig. 1c,e). This difference could be compensated for by expressing other heterochromatin proteins such as HP1, which polarizes the condensability of gene bodies according to transcription level in H1-hESC (Fig. 3c and Extended Data Fig. 8d). The PTM library data show that ubiquitylation, for either repressive (H2AK119Ub) or active (H2BK120Ub) marks, strongly impedes nucleosome condensation (Fig 3e), indicating that other factors must be recruited through chemical recognition to differentiate between the two ubiquitin modifications. Interestingly, in the micronuclei in which nuclear import is defective, both H2AK119Ub and H2BK120Ub are reduced, potentially contributing to more-condensed chromosomes in the micronuclei, which are also marked by reduced histone acetylation and increases in H3K36me3 (ref. 46). We were surprised that almost all PTMs, including charge-neutral methylations, reduce condensation. Overall, the direct physical effect of all these modifications is to increase the accessibility of chromatin, albeit to varying degrees, depending on the type (Fig. 3d and Extended Data Fig. 9a–c), which might serve as the initial physical opening of chromatin for docking epigenetic readers into action.

We wondered whether condensability drives differential gene expression, or whether it is a mere consequence of differential gene expression. The H3K36me3 marks, which are prevalent in highly transcribing gene bodies, do not show an enrichment in low-condensability partitions, indicating that the regions around the TSS, such as promoters and enhancers, rather than the gene body itself, are occupied by less-condensable nucleosomes. This is further supported by Chrom-HMM analysis (Fig. 1c) and meta-gene profiles (Fig. 1e). Therefore, high traffic by transcription machinery alone is not sufficient to lower nucleosome condensability, and we favour a model in which

cells regulate gene expression by modulating the condensability of promoter nucleosomes. High condensability in the gene body may help to prevent spurious initiation of transcription.

Although the nucleosome core particle (NCP), lacking linker DNA connecting nucleosomes in chromatin fibre, seems to contain sufficient information for large-scale genomic compartmentalization, and electrostatics can drive the compaction of NCPs, similar to that in nucleosome arrays[47], we do not neglect the possibility that the linker DNA may have an important role in genome organization through the modulation of nucleosome spacing[48], synergizing with the intrinsic condensabilities of individual NCPs. For example, the small reduction in condensability we observed for NCPs with H4K20me1 (Extended Data Fig. 5d), a modification known to induce decompaction in nucleosome arrays[49], indicates that some histone modifications may mainly impact condensation in arrays.

Polyamines, which exist at millimolar concentrations in eukaryotic cells[21], must have an important role in genome organization because, when they are depleted, cells try to compensate by accentuating the contrast in nucleosome condensability (Fig. 4). This hyperpolarization, which is consistent with the dual role of polyamine as a repressor and an inducer of gene expression, depending on the genes and cellular context, as previously reported[50], can result in various dysfunctions in cell differentiation[3], cancer[4] and immunity[5], through either direct interaction or metabolic perturbation of chromatin remodelling. Understanding this link, which shows how polyamines change the biophysical properties of chromatin, would be an interesting direction for future study.

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

## Methods

### Native mononucleosome purification

We used the hydroxyapatite (HAP) based protocol with minor modifications[51] (see Supplementary Note 1 for full details). In brief, we cultured mammalian cell lines, including human embryonic stem cells H1-hESC (WiCell), GM12878 (Coriell Institute) and ES-E14TG2a (a gift from Ian Chambers, University of Edinburgh), and collected approximately 100 million cells. Next, we purified the nuclei with 0.3% NP-40 buffer and performed MNase digestion at 37 °C for 10 min in the presence of protease-inhibitor cocktails and other deacetylation and dephosphorylation inhibitors. The soluble mononucleosomes were saved after centrifugation of the insoluble nuclei debris in a cold room. The nucleosome samples were incubated with hydroxyapatite slurry for 10 min, and then unbound proteins were removed by repetitive washing with intermediate salt buffers. Finally, the nucleosomes were eluted with phosphate buffer from the hydroxyapatite slurry. The eluted fraction was checked by extracting DNA from the nucleosome through phenol-chloroform extraction and running a 2% agarose gel. The HAP elution contained mononucleosomes, naked DNA and oligonucleosomes. We applied further size selection of mononucleosomes using Mini Prep Cell (Biorad) gel-based size-selection purification. The quality of the final mononucleosome sample was checked by running a 2% agarose gel and a 20% SDS–PAGE gel. The purified mononucleosomes were stored on ice in a cold room for less than a week before the condensation reaction, or they were frozen in liquid nitrogen with 20% glycerol for long-term storage at −80 °C. All cell lines used in this study were routinely tested for mycoplasma contamination and confirmed to be negative throughout the duration of the study.

### Nucleosome condensation assay

The purified native mononucleosome sample was extensively dialysed into 10 mM Tris pH 7.5 buffer through several buffer exchanges using an Amicon Ultra 10-kDa filter (MilliporeSigma). In each condensation reaction, the final concentration of nucleosome or DNA was 50 ng $\mu l^{-1}$ as DNA weight, and BSA was added to the final 0.2 mg $ml^{-1}$ to stabilize the nucleosome core particle. The condensation buffer condition was 10 mM Tris pH 7.5 with more salt depending on the condensing agents (50 mM NaCl for spermine and 250 mM NaCl for PEG (molecular weight, 8 kDa)). We prepared 8–16 samples with different concentrations of condensing agents simultaneously. They were incubated at room temperature for 10 min and centrifuged at 16,000$g$ for 10 min, and the supernatant was saved. The soluble-nucleosome concentration was measured using a Nanodrop UV spectrometer, and the nucleosome sample integrity was checked by running the 2% agarose gel (Supplementary Fig. 1). The rest of the nucleosomes in the supernatant were saved for use in high-throughput sequencing.

### Next-generation sequencing and library preparation

Using phenol-chloroform extraction, genomic DNA was extracted from the nucleosome, which was either the input control sample or the supernatant saved from the nucleosome condensation assay. The extracted DNA sample was then washed several times with distilled water using an Amicon Ultra 10-kDa filter (MilliporeSigma). Using the NEBNext Ultra II DNA library preparation kit (NEB), the DNA was adapter-ligated and indexed for Illumina next-generation sequencing (NGS). The final indexing PCR was conducted in 5–7 cycles. We used a HiSeq 2500 or a NovaSeq 6000 platform (Illumina) for 50 bp-by-50 bp pair-end sequencing. In each experimental condition, we sequenced the samples over multiple titration points to get data with 10-kb resolution but deeply sequenced a few selected titration points to achieved approximately 20× coverage of the entire human genome at single-nucleosome resolution. In this paper, we focused mainly on the titration points near complete depletion of the solution fraction, in which we could observe the highest contrast of nucleosome condensabilities with strong selection power

(for example, [spermine] = 0.79 mM in Fig. 1b and [HP1α] = 6.25 μM in Extended Data Fig. 8a).

### Genetic and epigenetic datasets

All the genome references and epigenetic data used in this work, including DNA methylation, histone ChIP-seq and Hi-C, are shown in Supplementary Tables 1–11.

### Computation of genome-wide nucleosome condensability

First, we obtained coverage profiles along the genome for input control and for the supernatant sample of each titration after the alignment of pair-end reads on the hg38 human genome assembly using Bowtie2 software[52]. On the basis of the coverage profile of the input control data, the position of each mononucleosome was localized by calling the peaks or finding the local maxima of the coverage profile. Beginning by randomly choosing a peak, the algorithm searched for all peaks in both directions, not allowing overlaps of more than 40 bp between 147-bp peak windows. For each nucleosome peak, the area of coverage in a window (we picked 171 bp as the window size) was computed for both the control and supernatant samples. The ratio of supernatant versus input read coverage area was combined with the titration curve measured by a UV–VIS spectrometer during the nucleosome condensation assay to estimate the survival probability of nucleosomes in the supernatant after condensation. Then, the negative natural log of this survival probability was used as a condensability metric for each mononucleosome peak. For the finer regular sampling used in plotting metagene profiles, the genome was binned into a 171-bp window with 25-bp sliding steps to compute the coverage area and the condensability scores. For a larger scale, we binned the genome into 1 kb or 10 kb and counted the reads aligned onto each bin to compute the condensability scores as the negative natural log of the ratio of supernatant to input read counts or the estimated survival probability inferred from the titration data. To avoid taking log of zero values, we added one pseudo-count to each input and supernatant read counts during the condensability calculation.

### Computation of a condensation point, $c_{1/2}$

The condensation point, $c_{1/2}$, was computed by using the survival probabilities of nucleosomes in multiple spermine concentrations. For each 10-kb genomic bin, we estimated the nucleosome counts in the input and supernatants after condensation in different spermine concentrations. We obtained the data points of spermine concentrations versus the soluble fraction of nucleosomes and fitted them with a logistic function. We then defined $c_{1/2}$ as the spermine concentration when the soluble fraction was half of the input.

### Using $z$-score computations as an enrichment metric

We used the $z$-score as the enrichment metric for genetic and epigenetic features. For example, we counted the number of CpG dinucleotides in each mononucleosome and standardized their distribution by subtracting the mean across all nucleosomes and dividing it by the standard deviation. Thus, each mononucleosome was assigned with a $z$-score of the CpG dinucleotide counts as the metric of how enriched or depleted the CpG was compared with the average in the unit of standard deviation. For the partitioned or grouped dataset of the quantile analysis, we used the averaged $z$-score for each partition as the enrichment metric.

### Data stratification and conditional correlation

To minimize the confounding effects between the genetic and epigenetic features of nucleosome condensation, the data were divided into subgroups that had one varying test variable, but all other variables were constant. For example, to evaluate whether AT content was correlated with condensability, the data were divided into smaller groups with the same genetic and epigenetic features, such as H3K4me3 and CpG methylations, except for AT content. In each stratified subgroup,

we checked the correlation between AT content and condensability. We then defined the conditional correlation between AT content and condensability as the weighted average of all correlations over the stratified subgroups, weighted according to the data size of each subgroup. In practice, it was difficult to obtain enough data for each stratified subgroup when the feature set is high dimensional. In this case, we discretized each genetic–epigenetic feature into a specific number. All histone ChIP-seq scores were discretized into 10 numbers, and other scores were discretized into 100 numbers.

## NMF decomposition
The genetic–epigenetic features of all mononucleosomes in chromosome 1 were linearly decomposed into ten basis property classes using a Scikit-learn NMF Python package. The nucleosomes were clustered into each property class with the highest component value in linear decomposition.

## Machine-learning models
First, we randomly selected 0.1 million nucleosomes from chromosome 1 for machine learning. For this dataset, the ridge regressor, supported vector regressor, gradient-boosting regressor, random-forest regressor and multilayer perception regressor were trained and validated using tenfold cross-validations. All machine-learning training and predictions were done using the Scikit-learn Python package. All analysis details are available and documented as IPython notebooks in our Github repository (https://github.com/spark159/condense-seq).

## Predicting the condensability of mononucleosomes
The condensability scores of mononucleosomes, as measured in H1-hESC cell lines using a spermine concentration of 0.79 mM, were predicted as a linear combination of the condensability scores of each PTM library member nucleosome measured at the same spermine concentration. For each PTM, the ChIP-seq signals on mononucleosomes were normalized by dividing them by the average ChIP-seq signal of the nucleosomes on chromosome 1, enabling comparison of different histone modifications at the same magnitude. The average of three measurements was used as the condensability score for each PTM. We restricted our analysis to mononucleosomes with at least six different types of PTM to prevent condensability from being influenced predominantly by PTMs not analysed in this study. The linear model was constructed as follows:

$$C_{mono} = \sum_{PTM} [\text{ChIP}_{PTM}] \times [C_{PTM}],$$

where $C_{mono}$ represents the predicted condensability of a mononucleosome, $\text{ChIP}_{PTM}$ indicates the normalized ChIP-seq signal and $C_{PTM}$ denotes the condensability of PTM-library nucleosomes. For further analysis, mononucleosomes were stratified using ChromHMM, and the predicted condensability of each chromatin state was compared with its measured counterpart (Extended Data Fig. 6d–f).

## Nucleosome reconstitution with canonical human octamers
Individual human histones H2A, H2B, H3.1 and H4 were purchased from the Histone Source (Colorado State University) and the octamers were reconstituted and purified following the standard protocol[53]. Then nucleosomes were reconstituted using Widom 601 DNA or purified genomic DNA by following the standard gradient salt-dialysis protocol[54]. Nucleosomes were further purified using Mini Prep Cell (Bio-Rad) to eliminate naked DNA or other by-product contaminants. For the PTM-library condense-seq experiment, the background reconstituted nucleosomes were made of Widom 601 DNA designed to have the same length and sequence as in the PTM library but with different primer-binding sequences, so it could not be amplified along with the library members. For the reconstitution of genomic DNA from GM12878, the genomic nucleosomal DNA was carefully purified at a size

of 150 bp by 6% PAGE purification (Bio-rad Mini Prep Cell) following the phenol-chloroform extraction of DNA from HAP-purified mononucleosomes. A histone octamer titration was required for each DNA batch because very small increments of octamer can induce aggregation and loss of mononucleosome yield. Reconstituted nucleosomes were further purified using a 6% polyacrylamide 29:1 Native PAGE column (Bio-Rad Mini Prep Cell). To increase the stability of mononucleosomes during PAGE separation, 0.02% NP40 was added to the column, running and elution buffers. Nucleosomes containing fractions were concentrated and stored on ice at 4 °C for immediate use.

## Purification of the HP1α and HP1β tSUV39H1 complex
We expressed and purified HP1α following the previous protocol[6]. In brief, we expressed HP1α with a His₆ affinity tag in *Escherichia coli* Rosetta (DE3) strains (MilliporeSigma) at 18 °C overnight. After cell lysis, the protein was first purified by cobalt-NTA affinity purification. The His tag was then cleaved by TEV protease, which was removed by anion-exchange purification using a HiTrap Q HP column (GE Healthcare). The HP1α was further purified by size selection using a Superdex-75 16/60 size-exclusion column (GE Healthcare). The HP1β with a truncated SUV39H1 complex (HP1β tSUV39H1) was similarly purified following a previous protocol[20].

## Nucleosome condensation assay of the PTM library
The PTM library was prepared as previously described[33]. The nucleosome condensation reaction of the PTM library was performed similarly, as described for the native mononucleosomes. However, because of the limited amount of the PTM-library sample, we spiked only a 1% (v/v) sample amount of the library into 99% (v/v) of reconstituted human nucleosomes as background for the condensation reaction. For condensation experiments using HP1α, a final concentration of 50 ng μl⁻¹ of DNA or nucleosome (DNA weight) was used in the reaction buffer (10 mM Tris-HCl pH 7.5, 100 mM NaCl, 0.2 mg ml⁻¹ BSA) with 5% (v/v) PEG 8000 as a crowding agent. Various amounts of HP1α were added to start the condensation.

## NGS library preparation and sequencing of the PTM library
The DNA sample was purified by phenol-chloroform extraction followed by several washes with distilled water using an Amicon Ultra filter (MilliporeSigma). The DNA library was then prepared for Illumina NGS sequencing by PCR using Phusion HF master mix (NEB) and custom indexed primers for the PTM library[33]. During amplification, the background nucleosome DNA was not amplified because it has different primer-binding sequences. We used MiSeq (Illumina) for sequencing libraries with custom primers, following previous protocols[33].

## Condensability calculation for the PTM library
The PTM library was de-multiplexed on the basis of the DNA hexamer barcodes by using a custom Python script and Bowtie2 aligner[52]. Then we approximated the nucleosome counts using information about the total soluble fraction, which was measured by a UV–VIS spectrometer, and the fraction of the individual members in the library, which was measured by Illumina sequencing. Finally, we computed the survival probability of each member in the library, which is the number of the remaining nucleosomes in the solution after condensation over input control. A negative log of survival probability was used for the condensability metric. For the PTM library, condensability averaged over many titration points was used as a condensability score for further analysis.

## Nucleosome–nucleosome interaction-energy calculations
Coarse-grained molecular-dynamics simulations of chromatin were done using OpenMM software[55]. Chromatin was modelled as beads-on-a-string polymers with each bead representing a genomic segment 25 kb long. Energy terms for bonds, excluded volume, spherical confinement and sequence-dependent contacts were defined.

Sequence-dependent contact energies were parameterized using read counts from condense-seq experiments. Contact probability matrixes were computed from these simulation trajectories and compared with experimental Hi-C contact maps. Full simulation details are provided in the Supplementary Note 2.

## Mouse CD8⁺ T cell culture and in vitro activation

Wild-type C57BL/6 mice and mice expressing Cre recombinase (CD4Cre) under the control of the CD4 promoter and *Rosa²⁶eYFP* were purchased from Jackson Laboratories, and *Odcflox/flox* mice were purchased from the KOMP repository. For experiments involving epigenetic marks, the spleen of *Odcflox/flox* or *Odc⁺/⁺ Rosa²⁶eYFP* mice were used to isolate and transduce T cells in vitro. All mice were bred and maintained in specific pathogen-free conditions under protocols approved by the Animal Care and Use Committee of Johns Hopkins University, in accordance with the Guide for the Care and Use of Animals. Mice used for all experiments were littermates and were matched for age and sex (both male and female mice were used). Mice of all strains were typically 8–12 weeks of age. Naive CD8⁺ T cells were isolated from the spleens of mice 8–12 weeks old using a negative-selection CD8 T cell kit (MojoSort Mouse CD8 T Cell Isolation Kit) according to the manufacturer's protocol. Isolated T cells ($1 \times 10^6$ per ml) were activated using plate-bound anti-CD3 (5 µg ml⁻¹) and soluble anti-CD28 (0.5 µg ml⁻¹) in T cell media (1640 Roswell Park Memorial Institute medium with 10% fatal calf serum, 4 mM L-glutamine, 1% penicillin/streptomycin and 55 µM β-mercaptoethanol) supplemented with 100 U ml⁻¹ rhIL-2 (Peprotech). Cells were cultured at 37 °C in humidified incubators with 5% $CO_2$ and atmospheric oxygen for 24 h after activation. After 48 h, T cells were removed from anti-CD3 and anti-CD28 and cultured at a density of $1 \times 10^6$ per ml in rhIL-2 (100 U ml⁻¹) at 37 °C for 7 days, with a change of media and fresh rhIL-2 every 24 h. To inhibit ODC, cells were incubated with 2.5 mM DFMO for 24 h at day 6 of culture. *Odc⁻/⁻*, wild-type and DFMO-treated cells were collected at day 7 for chromatin isolation and sequencing.

## Lentiviral production and cell transduction

HEK293T cells were transfected using Lipofectamine 3000 (Thermo Fisher Scientific) with the lentiviral packaging vectors pCAG-eco and psPAX.2 plus Cre-expressing vector pLV-EF1-Cre-PGK-Puro (all obtained from Addgene). The produced lentivirus was collected from the supernatant of the cells. CD8⁺ naive T lymphocytes isolated from *Odc⁺/⁺ Rosa²⁶eYFP* mice or Odcflox/flox *Rosa²⁶eYFP* were transduced by centrifugation in the presence of polybrene (8 mg ml⁻¹) in a plate treated with anti-CD3 (5 µg ml⁻¹), soluble anti-CD28 (0.5 µg ml⁻¹) and 100 U ml⁻¹ rhIL-2. The virus was removed after 6 h and fresh media containing anti-CD28⁺ IL-2 was added again. After two days, the transduced cells were selected by flow cytometry and sorted by expression of YFP (Cre⁺ cells) in the CD8⁺ live-cell population and cultured in the presence of 100 U ml⁻¹ rhIL-2 for two more days.

## Assessment of epigenetic marks by flow cytometry

Transduced CD8⁺ YFP⁺ sorted T cells from *Odc⁺/⁺* and *Odcflox/flox* were fixed and stained for intracellular immunostaining. The measurement of the histone methylation and acetylation marks enrichment was done using flow cytometry for sorted CD8⁺ eYFP⁺ T cells from *Odc⁺/⁺* and *Odcflox/flox* (wild type and KO, respectively) mice, and they were fixed for 60 min at room temperature using a FOXP3 permeabilization kit (eBioscience) and stained for 90 min with primary antibodies against H3K36me3 (Polyclonal, from Abcam), H3K4me3 (clone C42D8), H3K27ac (clone D5E4), H3K27me3 (clone C36B11), H3K9ac (clone C5B11) and rabbit monoclonal antibody IgG isotype control (DA1E) (all from Cell Signaling Technology unless stated otherwise) and stained for 30 min with donkey anti-rabbit IgG (H + L) Highly Cross-Adsorbed Secondary Antibody, Alexa Fluor Plus 647 (Thermo) at room temperature. Cells were gated on diploid cells with 'single' DNA content based on FxCycle staining (Thermo Fisher) in the live-cell gate.

## Histone PTM enrichment measurement

For the mass-spectrometry measurement, native mononucleosomes were purified from the GM12878 cell line and a nucleosome condensation assay was similarly performed using spermine (250 ng µl⁻¹ nucleosome, 0.079 mM spermine in 10 mM Tris-HCl pH 7.5 buffer at room temperature). The input/soluble/pellet nucleosome sample was washed several times in 10 mM Tris-HCl pH 7.5 buffer using an Amicon Ultra filter (10-kDa cut-off) to remove spermine and kept at 70 °C for 20 min to dissociate DNA from the histones. The free DNA was further removed in the desalting step of the mass-spectrometry process. About 20 µg of purified histone was derivatized using propionic anhydride[56] followed by digestion with 1 µg trypsin for bottom-up mass spectrometry. The desalted peptides were then separated in a Thermo Scientific Acclaim PepMap 100 C18 HPLC Column (250 mm length, 0.075 mm internal diameter, reversed-phase, 3 µm particle size) fitted on a Vanquish Neo UHPLC system (Thermo Scientific) using an HPLC gradient as follows: 2% to 35% solvent B (A = 0.1% formic acid; B = 95% MeCN, 0.1% formic acid) over 50 min, to 99% solvent B in 10 min, all at a flow rate of 300 nl min⁻¹. About 5 µl of a 1 µg µl⁻¹ sample was injected into a QExactive-Orbitrap mass spectrometer (Thermo Scientific) and a data-independent acquisition was carried on, as described previously[56]. In brief, full-scan mass spectrometry ($m/z$ 295–1,100) was acquired in an Orbitrap with a resolution of 70,000 and an AGC target of $1 \times 10^6$. Tandem mass spectrometry was set in centroid mode in the ion trap using sequential isolation windows of 24 $m/z$ with an AGC target of $2 \times 10^5$, a CID collision energy of 30 and a maximum injection time of 50 ms. The raw data were analysed using in-house software, EpiProfile[57]. The chromatographic profile and isobaric forms of peptides were determined using precursor and fragment-extracted ions. The data were output as peptide relative ratios (percentages) of the total area under the extracted ion chromatogram of a particular peptide form to the sum of unmodified and modified forms belonging to the same peptide with the same amino acid sequence. The $\log_2$-transformed fold change in the peptide relative ratio in the soluble/pellet fraction versus the input was computed as the enrichment metric. Using the unmodified peptide as the reference, the difference in fold change between the PTM modified peptide and the unmodified peptide was computed and plotted as a heatmap.

## Calibrated ChIP-seq

We followed a published ChIP protocol[58] with minimal modifications. Antibody-conjugated beads were prepared by adding 50 µl of Protein A beads per ChIP reaction (Thermo Fisher) to a 2 ml tube, washing twice with 1 ml of blocking buffer (0.5% BSA in PBS) and resuspending in 100 µl blocking buffer per ChIP reaction. Antibody was then added to the beads (4 µl of H3K27ac antibody (Novus ab4729) and 4 µl of H3K27me3 (Novus ab192985) plus 2 µg of spike-in antibody (Active-Motif) per reaction), and the mixture was incubated with rotation for 1–3 h. Crosslinked cell pellets were resuspended in 4 ml of lysis buffer LB1 (50 mM HEPES, 140 mM NaCl, 1 mM EDTA, 10% glycerol, 0.5% Igepal CA-630, 0.25% Triton X-100, pH adjusted to 7.5, 1× protease inhibitors) and incubated in LB1 for 10 min at 4 °C with rotation. Cells were then spun down at 2,000*g*, at 4 °C for 3 min. The supernatant was discarded and pellets were resuspended in 4 ml of LB2 (10 mM Tris-HCl pH 8, 200 mM NaCl, 1 mM EDTA, 0.5 mM EGTA, pH 8.0, 1× protease inhibitors) and incubated at 4 °C with rotation for 5 min, then spun down (with the same settings). The supernatant was removed and cells were then resuspended in 1.5 ml of LB3 (10 mM Tris-HCl pH 8, 100 mM NaCl, 1 mM EDTA, 0.5 mM EGTA, 0.1% Na-deoxycholate, 0.5% *N*-lauroylsarcosine, pH 8.0, 1× protease inhibitors) and transferred to 2-ml tubes. Sonication was performed using a Fisher 150E Sonic Dismembrator with the following settings: 50% amplitude, 30 s on, 30 s off for 12 min total time. The sonicated sample was spun down at 20,000*g* and 4 °C for 10 min, and the supernatant was transferred to a 5 ml tube. Then, 1.5 ml of LB3 (with no protease inhibitor), 300 µl of 10% Triton X-100, and 120 ng

of *Drosophila* spike-in chromatin (ActiveMotif) per 25 μg of ChIP'ed chromatin were added to each sample. The entire solution was mixed by inversion. The 2-ml tubes containing antibody-conjugated beads were placed on a magnetic rack, washed three times with 1 ml of blocking buffer, and resuspended in 50 μl of blocking buffer per ChIP reaction. We then transferred 50 μl of antibody-conjugated beads to each ChIP reaction and incubated them overnight at 4 °C with rotation. ChIP samples were transferred to a 1.5 ml LoBind tube, placed on a magnetic stand and washed six times with 1 ml RIPA buffer (50 mM HEPES, 500 mM LiCl, 1 mM EDTA, 1% Igepal CA-630, 0.7% Na-deoxycholate, pH 7.5) and once with 1 ml TBE buffer (20 mM Tris-HCl pH 7.5, 150 mM NaCl). The supernatant was discarded, and the beads were eluted in 50 μl elution buffer EB (50 mM Tris-HCl pH 8.0, 10 mM EDTA, 1% SDS) and incubated at 65 °C overnight with shaking at 1,000 rpm. We then added 40 μl TE buffer to the mixture to dilute the SDS, followed by 2 μl of 20 mg ml$^{-1}$ RNaseA (New England BioLabs), and samples were incubated for 15 min at 37 °C. Then, 4 μl of 20 mg ml$^{-1}$ Proteinase K (New England BioLabs) was added and the samples were incubated for 1 h at 55 °C. The genomic DNA was column purified and eluted in 41 μl of nuclease-free water. Sequencing libraries were prepared using the NEB Next Ultra II End Repair/dA-Tailing Module (New England BioLabs), using half volumes. Libraries were amplified with 10 (H3k27ac) or 13 (H3k27me3) cycles of PCR using single indexed primers. ChIP'ed DNA samples were then pooled, quantified with QuBit and qPCR (BioRad), and sequenced on a NextSeq 1000 Illumina machine using paired 2 × 50 bp reads. Reads were demultiplexed after sequencing using bcl2fastq and aligned to the mm10 genome using bowtie2. Samtools63 was used to filter for a mapping quality greater than or equal to 25, remove singleton reads, convert to BAM format and remove potential PCR duplicates and index reads.

### Two-colour smFRET imaging for nucleosome unwrapping

Biotinylated Cy3/Cy5 20N20 mononucleosomes (25 mM Hepes-KOH pH 7.6, 5% glycerol, 0.017% NP-40, 70 mM KCl, 3.6 mM MgCl2 and 0.1 mg ml$^{-1}$ BSA) were incubated in surface-functionalized chambers for 2 min. Free nucleosomes were flushed out with dilution buffer containing imaging additives (oxygen-scavenging system: 0.8% w/v dextrose, 2 mM Trolox, 1 mg ml$^{-1}$ glucose oxidase (Sigma-Aldrich) and 500 U ml$^{-1}$ catalase (Sigma-Aldrich)). Basal nucleosome fluorescent emission was recorded to control density and FRET signal before the addition of spermine. A total of 10 short movies (100 ms exposure time) of 20 frames each were taken (10 frames using Cy3 excitation and 10 frames using Cy5 excitation). Spermine was introduced to the imaging chamber in dilution buffer containing imaging additives and incubated for 10 min. Short movies were taken using the settings explained above. FRET histograms were generated from donor and acceptor fluorescent intensities of single molecules. The details of the nucleosome construct and single-molecule imaging conditions can be found in Supplementary Note 3.

### Single-molecule nucleosome pull-down assay

Biotinylated Cy3-H2A(K120C) 20N0 mononucleosomes were dialysed into 10 mM Tris pH 7.5 buffer through three buffer exchanges using an Amicon Ultra 10-kDa filter (MilliporeSigma). Nucleosomes were diluted to 7.5 nM and BSA was added to a concentration of 0.2 mg ml$^{-1}$. For condensation, 5 nM mononucleosomes were mixed with 0.4 mM spermine in 10 mM Tris pH 7.5 and 50 mM NaCl. The reaction was covered from light and incubated at room temperature for 10 min. Before immobilization, spermine-condensed nucleosomes were mixed and immediately diluted 50 times in 10 mM Tris pH 7.5, 50 mM NaCl and 0.4 mM spermine (pull-down buffer). Dilution flowed into neutravidin functionalized chambers and incubated for 10 min with the quartz slide facing down. The chamber was washed with pull-down buffer including imaging additives. Short movies of 20 frames (100 ms exposure time) were taken using Cy3 excitation. Laser intensity was regulated to control the intense fluorescent signal from large condensates immobilized on the single-molecule surface. A control experiment was done in which

spermine was removed from the condensation reaction and pull-down buffers. Nucleosomes were diluted 500-fold for immobilization and only single nucleosome spots were observed. Detailed information of nucleosome constructs and single-molecule imaging conditions are in Supplementary Note 3 and Supplementary Table 12.

### Reporting summary

Further information on research design is available in the Nature Portfolio Reporting Summary linked to this article.

### Data availability

Sequencing data have been deposited in the GEO database with accession number GSE252941. Source data are provided with this paper.

### Code availability

All condense-seq data analysis was conducted using custom Python scripts, which are available on GitHub at https://github.com/spark159/condense-seq and archived at Zenodo at https://doi.org/10.5281/zenodo.15036149 (ref. 59).

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

**Acknowledgements** We thank J. Yoo for formulating the initial idea; K. Onishi and A. Feinberg for advice about the experiments; and W. Timp, G. Mizuguchi and D. Kim for discussions. The HP1α plasmid was a gift from the G. Narlikar lab; the HP1β and SUV39H1 plasmids were gifts from the P. Li lab. E14Tg2a mESCs were received from I. Chambers. This work was supported by the National Science Foundation (NSF) Emerging Frontiers in Research and Innovation, Chromatin and Epigenetic Engineering (EFMA 1933303 to T.H.); the NSF Science and Technology Center 'Quantitative Cell Biology' (2243257 to T.H.), the National Institutes of Health (GM 122560 and DK 127432 to T.H.; R35 GM133580 to B.Z.; R01 GM086868, R01 CA240768 and P01 CA196539 to T.W.M.; R01 HD106051 and R01 AI118891 to B.A.G.; and R01 AI170599 to E.L.P.). B.A.G. was also supported by a grant from the St. Jude Chromatin Collaborative. M.M.M. was supported by an NIH postdoctoral fellowship (GM131632). T.H. is an investigator of the Howard Hughes Medical Institute.

**Author contributions** S.P. and T.H. designed the research. S.P. performed all aspects of the research and data analysis. S.P. and T.H. wrote the paper. R.M.U. and V.K.-N. synthesized and performed condense-seq using nucleosomes reconstituted with genomic DNA and recombinant histone octamers. R.M.U. conducted single-molecule FRET experiments. V.K.-N. cultured E14 mESCs and performed condense-seq using purified nucleosomes from them. G.E.C. cultured and purified CD8+ T cells from mouse *Odc*-KO lines and did flow-cytometry experiments. A.A. did interaction-energy calculations and chromatin-polymer simulations. N.A.B. helped with H1-hESC and GM12878 cell culture and nucleosome purification. A.M.-G. did ChIP-seq experiments. J.P. built the linear model for predicting genomic nucleosome condensability from PTM library data. M.M.M. advised on PTM library-based experiments. N.V.B. did bottom-up mass spectroscopy to identify histone PTMs. B.Z., B.A.G., T.W.M. and E.L.P. provided discussion and supported collaboration. All authors commented on the manuscript.

**Competing interests** The authors declare no competing interests.

**Additional information**
**Correspondence and requests for materials** should be addressed to Taekjip Ha.

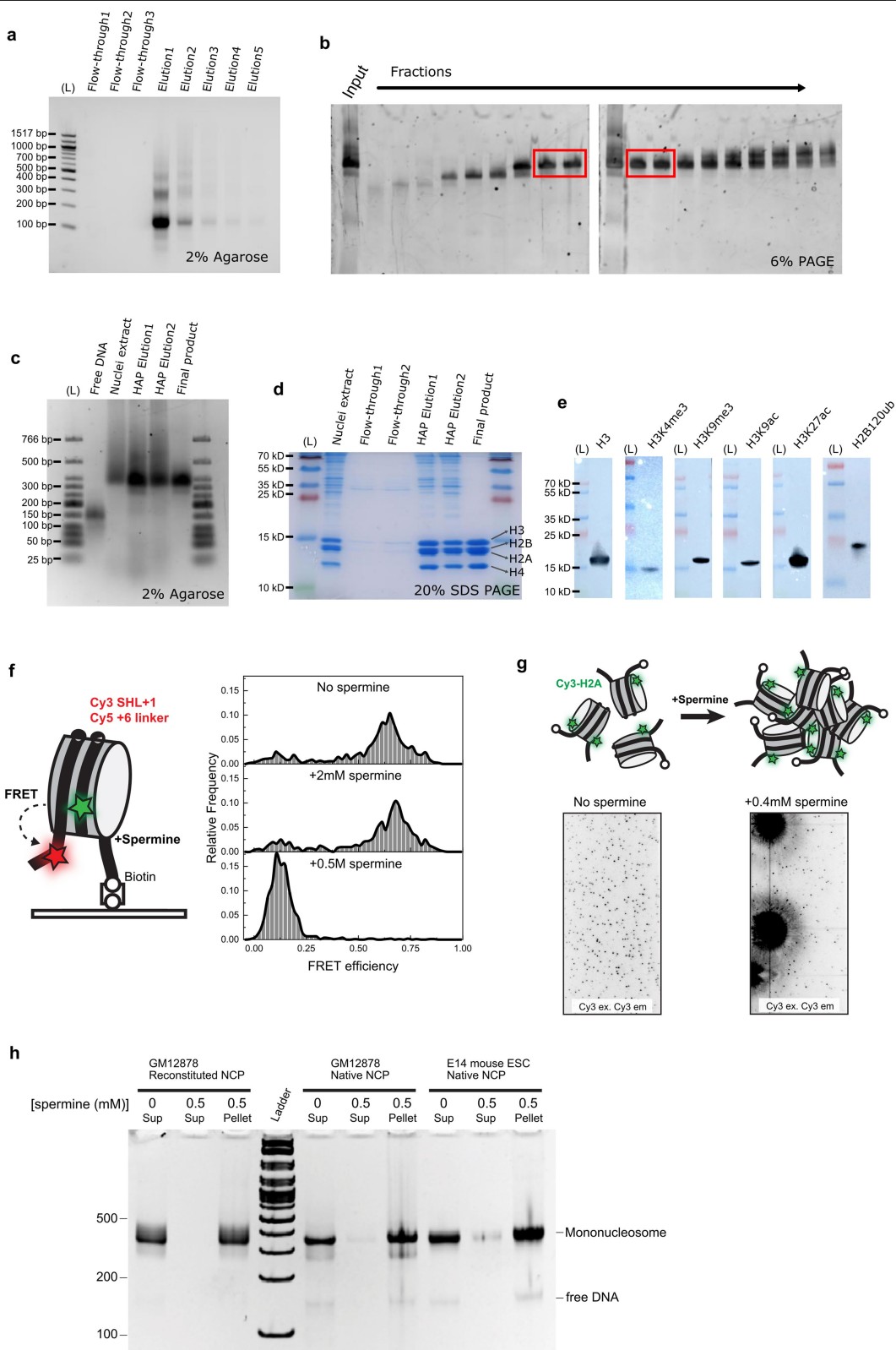

**Extended Data Fig. 1** | See next page for caption.

**Extended Data Fig. 1 | Intact native mono-nucleosomes obtained by hydroxyapatite (HAP) and size-selective purification. a**, After the HAP purification of MNase-treated chromatin, flow-through and elution samples were run in 2% agarose gel. The 1st lane is NEB 100 bp DNA Ladder (denoted as L). **b-e**, Mono-nucleosomes were selected through further size-selective purification of HAP elution. HAP elution input and each fraction of size-selection are shown in (b). Each purification step and the quality of final product was validated by running the samples in 2% agarose gel (**c**), SDS-PAGE gel showing only four histones without other proteins (**d**), and western blot for histone PTMs (**e**). (Ladder: NEB Low Molecular Weight DNA ladder is used for (c) and Thermo Scientific PageRuller used for (d-c)). **f**, Schematics of single molecule FRET analysis using a FRET pair (green for donor, red for acceptor) conjugated to DNA designed to show a FRET decrease upon DNA unwrapping (left). Single molecule FRET histograms (right) showed that there is no detectable unwrapping at the spermine concentration relevant to condense-seq (up to 2 mM). At 0.5 M spermine, DNA is unwrapped. **g**, Visualization of nucleosome condensates via total internal reflection fluorescence microscopy of Cy3 conjugated to H2A. Biotin (empty circle) is used to capture the nucleosomes on a passivated neutravidin-coated surface after incubation with and without 0.4 mM spermine prior to capture. Data show that 0.4 mM spermine is sufficient to induce nucleosome condensates in vitro. **h**, To confirm integrity of nucleosomes during polyamine-induced condensation, we ran a gel of nucleosome core particles (NCPs) before condensation (left lane in each category) and after solubilization following condensation in the presence of 0.5 mM spermine (right lane). The middle lanes show that most of NCPs have been condensed at 0.5 mM spermine. Resolubilized NCPs collected from the condensed pellet showed the same migration pattern as the input NCPs, demonstrating their integrity for GM12878 reconstituted NCPs, GM12878 native NCPs and E14 mESC NCPs. The nucleosome integrity was checked with similar results from three independent experiments.

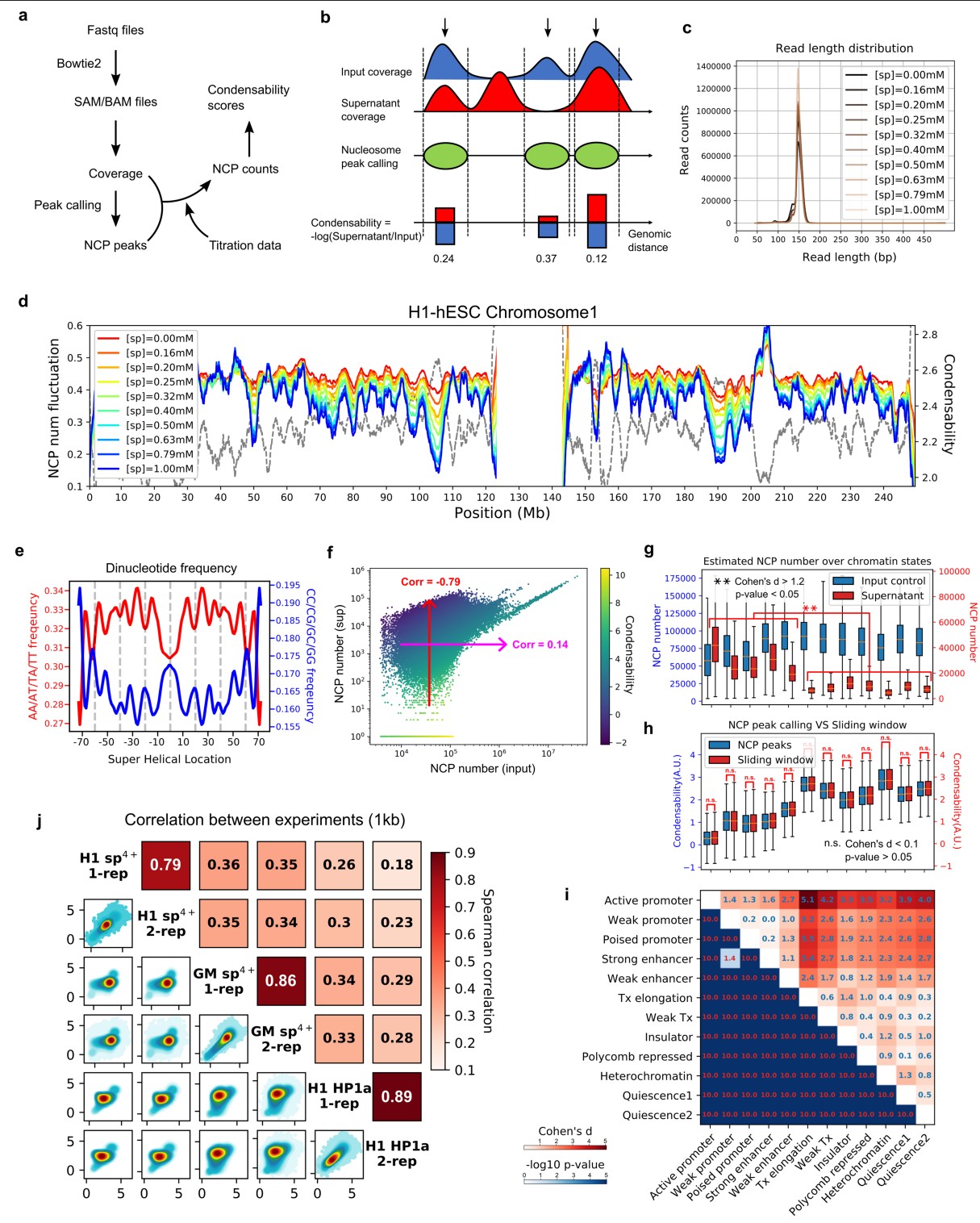

**Extended Data Fig. 2** | See next page for caption.

**Extended Data Fig. 2 | Computational pipeline and data quality controls for condense-seq. a**,**b**, The pipeline of Condense-seq analysis is composed of (i) reads alignment by Bowtie2, (ii) coverage calculations, (iii) mono-nucleosome peak calling for each local maximum of input coverage, (iv) absolute nucleosome count estimation using coverage area and soluble fraction changes from the titration data of the UV-VIS spectrometry measurement, and (v) compute condensability score as negative log of soluble fraction after condensation for each nucleosome. **c**, For quality control, we checked that the length distribution of nucleosomal DNA of nucleosomes remaining in the supernatant is mostly around at 150 bp for all concentrations of spermine used. (**d**) Nucleosome number fluctuation vs genomic position in Chr1. The input ([sp]=0 mM, red curve) shows mostly flat values, showing that there is no strong bias in the input. NCPs remaining in the supernatant show progressively strong bias at higher [sp]. **e**, The periodicity of AT-rich versus GC-rich dinucleotides, the hallmark indicator of nucleosome peaks, supports the nucleosomal source of DNA analyzed. **f**, Condensability is more highly correlated with the supernatant nucleosome number changes than the input (Spearman correlation coefficient −0.79 vs 0.14). **g**, Estimated NCP number for various ChromHMM chromatin states for input vs supernatant ([sp] = 0.79 mM). Analyses in (**d**), (**f**), and (**g**) collectively show that condensability score is mostly determined by the degree of how much nucleosomes are condensed, not by the variations in the input NCPs. **h**, Condensability determined via nucleosome peak calling and regular sliding windows gave almost identical results for various ChromHMM chromatin states (p-value > 0.05 and Cohen's d < 0.1 for every comparison). All boxplot centers represent median, and the lower/upper bounds is the 1$^{st}$/3$^{rd}$ quartile of data. **i**, The statistical significance (p-value using t-test) and effect size (Cohen's d) are computed for condensability difference between each pair of ChromHMM states (data in Fig. 1c and Extended Data Fig. 2h). Numeric values are shown for each cell for Cohen's d (top right triangle) and -log10 p-value (bottom left triangle). **j**, Correlations of condensability values between replicates. All statistics were computed via two-sided Welch's t-test over more than 7000 nucleosomes (g-i) or 40000 genomic bins (h) of each state from two biological replicates.

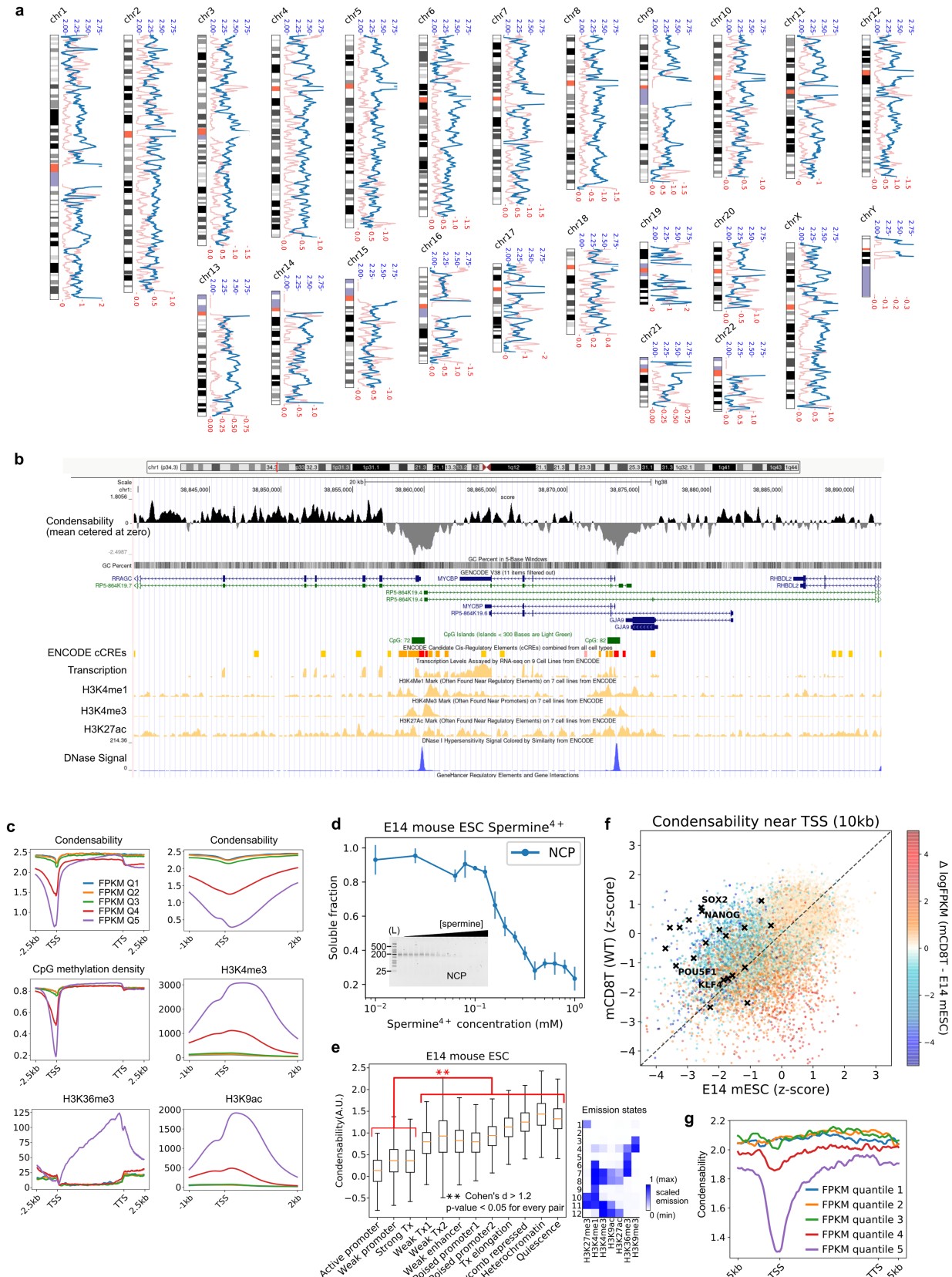

**Extended Data Fig. 3** | See next page for caption.

**Extended Data Fig. 3 | Condensability measurements of human embryonic stem cell (H1-hESC) and mouse embryonic stem cell (E14 mESC).**
**a**, Comparison between condensability (blue) and transcription level (red) along all chromosomes of H1-hESC. **b**, Snapshot of UCSC genome browser for the condensability profile of H1-hESC along with many other cis-regulatory elements. **c**, All genes were grouped into five quantiles according to the transcription level of H1-hESC (quantile 1 through 5 for increasing transcription). Condensability, methylated CpG density, and H3K36me3 along the transcription unit coordinate averaged for each quantile (left column). Views zoomed around TSS are shown for condensability, H3K4me3 and H3K9ac (right column). **d**, Native nucleosomes are prepared from mouse embryonic stem cells (E14 mESC) and condensed by spermine titration (the titration curve is the mean value of three replicates and error bar represents the standard deviation). NEB Low Molecular Weight DNA ladder was used for the first lane as marker.

**e**, Genome segmentation into chromatin states based on histone PTM ChIP-seq data (right). All mono-nucleosomes of chromosome 1 were categorized using ChromHMM, and their condensability distribution for each chromatin state is shown (boxplot: the center is median and the lower/upper bound is the $1^{st}/3^{rd}$ quartile of data). Statistically significant differences between ChromHMM states are noted. The statistics were computed via two-sided Welch's t-test over more than 400 nucleosomes of each state from two biological replicates. **f**. Promoter condensability (averaged over 10 kb window around TSS) for E14 mESC and mCD8 T cells. Each gene is colored according to their relative expression levels in the two cell types. Black symbols are for embryonic stem cell marker genes. **g**, All genes in chromosome 1 were grouped into five quantiles according to the transcription level (quantile 1 through 5 for increasing transcription) and condensability along the transcription unit coordinate averaged for each quantile is shown.

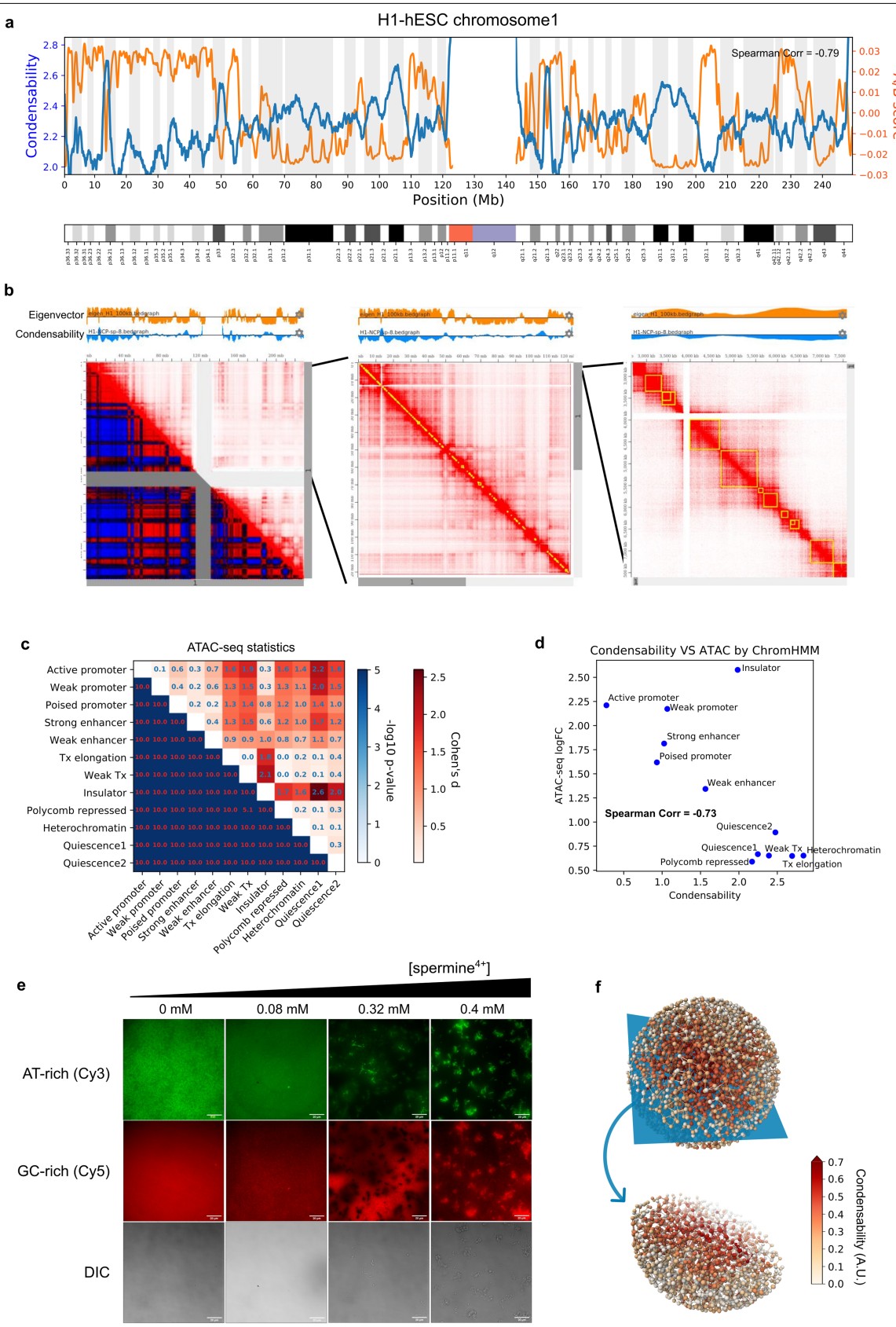

**Extended Data Fig. 4 |** See next page for caption.

**Extended Data Fig. 4 | Spatial separation of molecules promoted by condensability difference to compartmentalize the genome. a, b**, H1-hESC condensability (blue) and A/B compartment scores based on Micro-C data (orange) in mega base-pair resolution of chromosome 1 (**a**) and finer resolution (**b**). **c**, Statistical significance (p-value using t-test) and effect size (Cohen's d) were computed for ATAC-seq signal fold change differences between each pair of ChromHMM chromatin states for data shown in Fig. 2f. The statistics were computed via two-sided Welch's t-test over more than 100000 genomic bins of each state from two biological replicates. **d**, ATAC-seq fold change vs condensability for various ChromHMM states shows an anticorrelation (Spearman correlation coefficient is −0.73). **e**, PCR amplified AT-rich (Cy3 labeled) and GC-rich (Cy5 labeled) DNAs were mixed and condensed in spermine concentrations indicated. For each condition, DNA condensates were imaged using wide-field microscope. As spermine concentration increased, AT-rich DNAs formed a condensed core first, and GC-rich DNAs condensed over the AT-rich core at higher spermine concentrations, promoting the spatial separation between AT-rich versus GC-rich condensates. A similar result was observed from two independent experiments. **f**, Chromosome polymer simulation with condense-seq data using spermine as the only input (GM12878, chr12) shows that highly condensable chromatin is compacted into the core and the rest is excluded to generate spatially separate compartments.

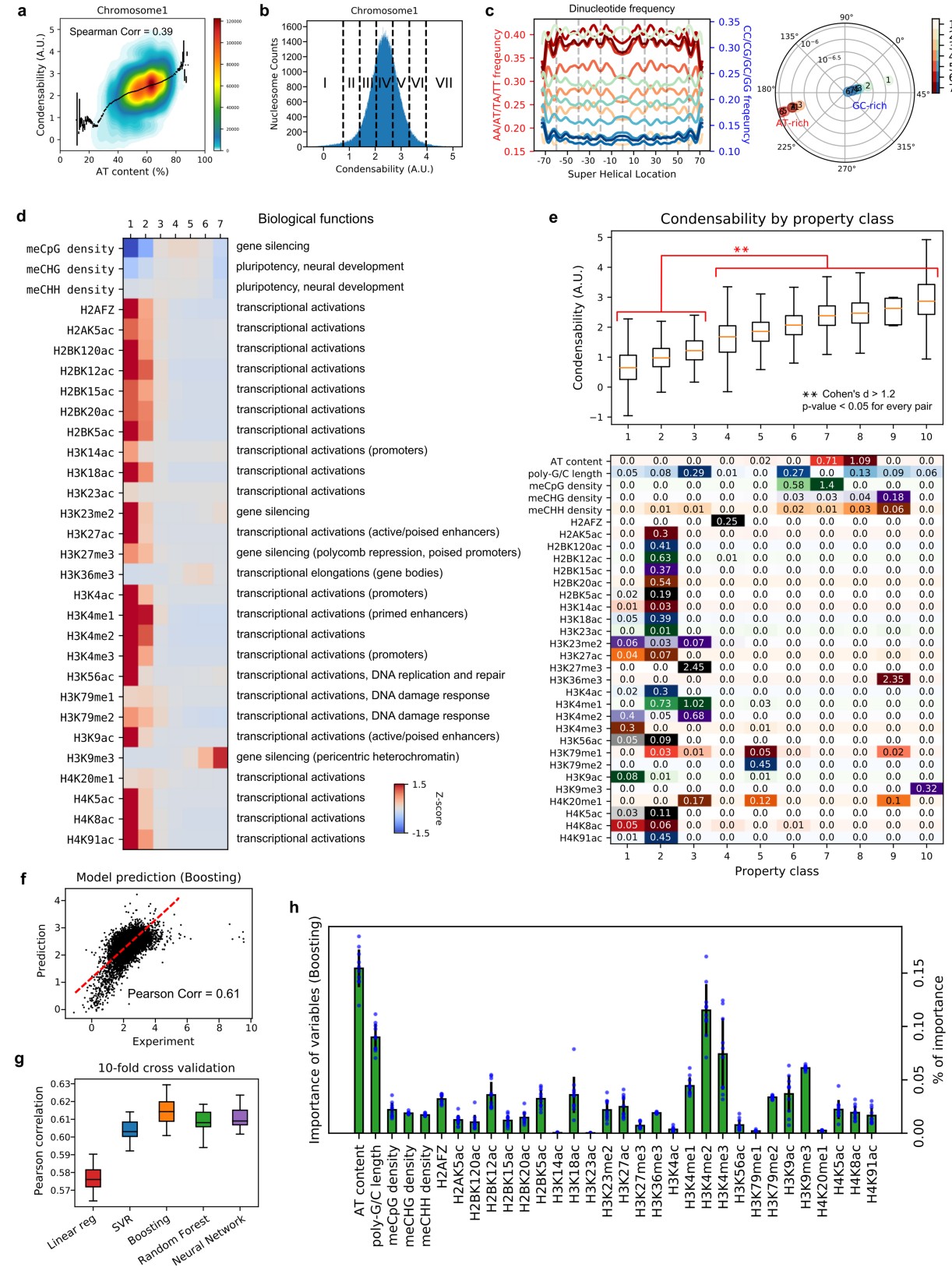

**Extended Data Fig. 5** | See next page for caption.

**Extended Data Fig. 5 | Deciphering the genetic and epigenetic determinants of genomic nucleosome condensation. a**, Scatter plot of the condensability of mono-nucleosomes in chromosome 1 and the AT contents of corresponding nucleosomal DNA. **b**, The nucleosome population was partitioned into seven partitions, from low to high condensability. **c**, The periodicity of AT-rich versus GC-rich dinucleotides. Average frequency of different dinucleotides vs position relative to nucleosome dyad for each of the seven partitions in (**b**) is shown (left). The amplitude and phase of dinucleotide frequency fluctuations vs position were computed using Fourier transformation and represented in a polar plot (right, radius: amplitude, angle: phase). **d**, The enrichment analysis of all DNA methylation and histone ChIP-seq data available in ENCODE over different condensability partitions from low to high (1–7 partitions in **b**). **e**, The genetic and epigenetic features of all mono-nucleosomes in chromosome 1 were linearly decomposed into 10 property classes by non-negative matrix factorization. Each property class has a specific combination of features, as shown in the matrix (lower panel). Every nucleosome was assigned to a representative property class with the largest contribution. After clustering, nucleosome condensabilities were plotted as boxplot for each class (upper panel) and p-values & Cohen's d were computed for condensability comparison across classes. In the boxplot, the center represents the median and the lower/upper bound shows the 1st/3rd quartile of data. The statistics were computed via the two-sided Welch's t-test over 7 to 500000 nucleosomes of each state from two biological replicates. **f-h**, Multivariate linear regression (linear reg), Supported Vector Machine regression (SVM), gradient boosting regression (Boosting), random forest regression (Random Forest), and neural networks were used to predict nucleosome condensability. All showed similar correlations between experimental values and predictions in 10 sampling replicates of 10-fold cross-validation (**f**,**g**). The importance of genetic–epigenetic features in prediction was computed using the boosting method shown as the bar plot of means from the 10 sampling replicates of 10-fold cross validation with error bar as the standard deviation (**h**).

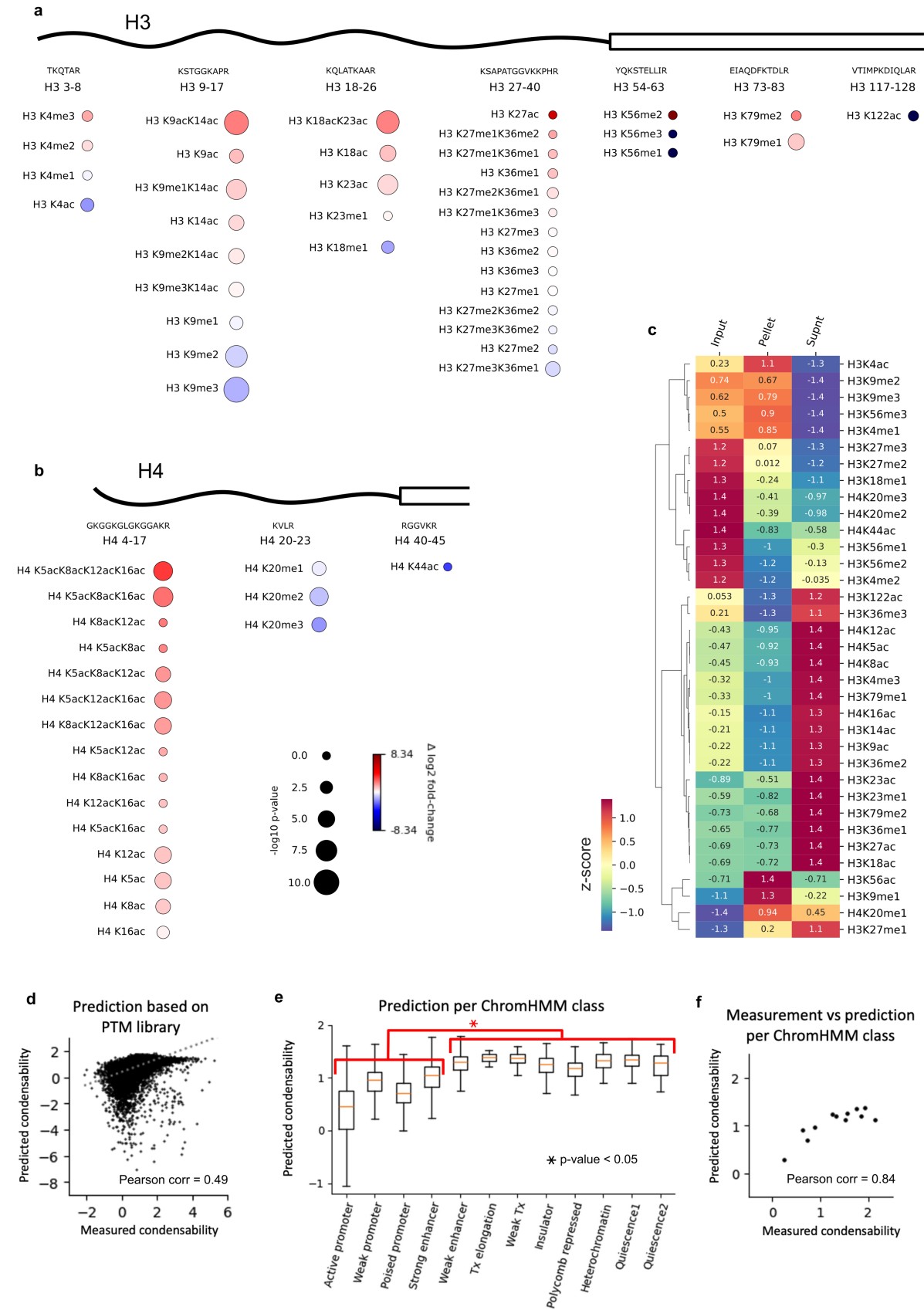

**Extended Data Fig. 6 | Mass spectrometry identification of histone PTM marks with biased enrichment during native mononucleosome condensation experiments. a,b,** Histone PTM marks detected in each histone H3/H4 peptide are shown. Its relative enrichment difference compared with the unmodified peptide is represented by color (red: more enriched in supernatant, blue: more depleted in supernatant) and its signification is represented by the size of bubble (-log p-value). The statistics were computed via the two-sided Welch's t-test over 4 technical replicates. **c,** Combinatorial histone PTM enrichment data was aggregated into single PTM modifications, and the relative enrichment in each phase of condensation (input/pellet/ supernatant) is shown in the z-score heat map. **d-f,** Only using the synthetic histone PTM library condensability data, the genomic nucleosome condensability of H1-hESC were predicted using linear regression model. The prediction shows a moderate correlation with experimental data at the single-nucleosome level (**d**), and could qualitatively reproduce the pattern of condensability change across different ChromHMM chromatin states (boxplot: the center is median and the lower/upper bound is the $1^{st}/3^{rd}$ quartile of data, statistics: two-sided t-test used for the comparison with 50–8000 nucleosomes of each ChromHMM state) (**e-f**).

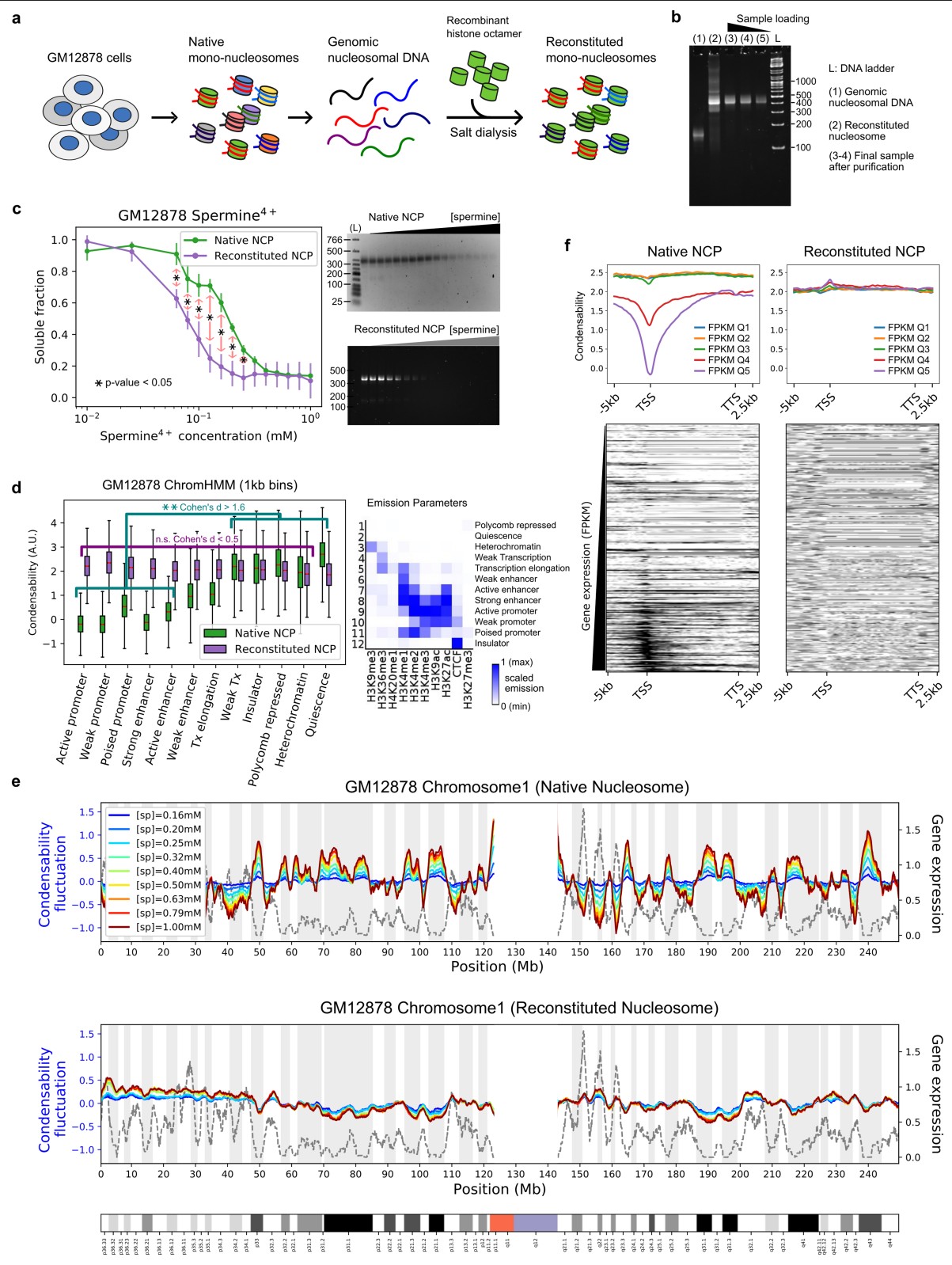

**Extended Data Fig. 7** | See next page for caption.

**Extended Data Fig. 7 | Condense-seq measurement of native and reconstituted mono-nucleosomes from GM12878 cells and the comparison between nucleosome condensability and their chromatin states. a**, Native mono-nucleosomes were purified from GM12878 cell line. For reconstituted nucleosomes, DNA was isolated and purified to size homogeneity before reconstitution with recombinant histone octamers without any PTMs, and was further purified. **b**, Pure reconstituted nucleosomes used in the condensation experiment are shown in 6% agarose gel. Samples included isolated genomic DNA from GM12878, the reconstituted nucleosomes, and final product after size-selection. (Ladder: NEB 100 bp DNA ladder used). **c**, Condensation was induced by adding spermine. Soluble fractions were measured using UV-VIS spectroscopy (left, the titration curves are plotted as the mean of three replicates with error bars as the standard deviation, and the asterisk represent the significantly different titration points when the p-value < 0.05 from the two-sided Welch's t-test from three replicates, and p-values are 0.006, 0.016, 0.016, 0.005, 0.003, 0.017, and 0.05 respectively) and ran in the 2% agarose gel (right). (Ladder: NEB Low Molecular Weight Ladder used). **d**, Native and reconstituted nucleosomes were grouped according to their ChromHMM states based on the combination of various PTMs Chip-seq data. Their condensabilities are shown in box plot for each chromatin state (green: native nucleosome, purple: reconstituted nucleosome), and the effect size of differences (Cohen's d) across the chromatin states was computed over more than 4000 nucleosomes of each state from two biological replicates (boxplot: the center is median and the lower/upper bound is the $1^{st}/3^{rd}$ quartile of data). **e**, The adjusted condensability score (after standardized by only mean, not variation, to compute the fluctuations) was plotted over human chromosome 1 for different spermine titration points (colored lines) and compared with the gene expression level (black dotted line). **f**. The condensability profiles of native and reconstituted nucleosomes from TSS to TTS for five quantiles based on the gene expression levels in the GM12878 cell line.

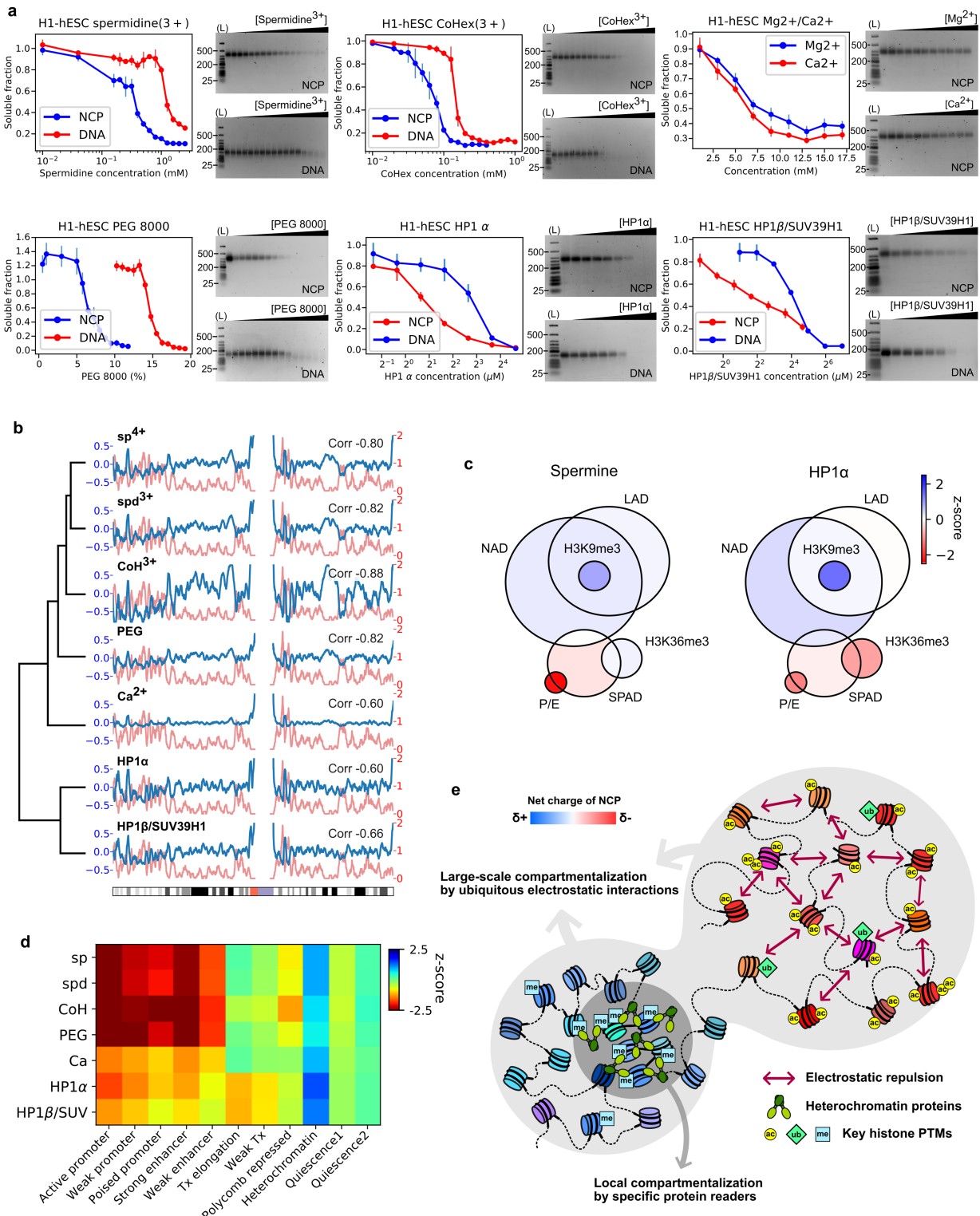

**Extended Data Fig. 8 | Condense-seq of H1-hESC native mono-nucleosomes using various condensing agents. a**, The soluble fraction of nucleosomes was measured by titrating the various condensing agents other than spermine, including spermidine, cobalt-hexamine, magnesium/calcium, PEG 8000, and HP1α, HP1β with SUV39H1 complex. The titration curves were plotted as the mean of three replicates with error bar as the standard deviation. (Ladder: NEB Low Molecular Weight DNA ladder used for the first lane of gels) **b**, Condensability scores were plotted over chromosome 1 (blue) and the Spearman correlation coefficient were computed compared with the gene expression level (red). Hierarchical clustering of the condensability profile shows that all ionic condensing agents (spermine/spermidine/cobalt-hexamine/PEG/calcium) are clustered together but other protein-based condensing agents (HP1α and HP1β) are clustered in a separate group. **c, d**, Comparison of condensability scores for different condensing agents across various nuclear compartments (**c**, LAD: lamina-associated domain, NAD: nucleolar-associated domain, SPAD: nuclear speckle-associated domain, P/E: promoter or enhancer) and chromatin states (**d**). **e**, Hypothetical hierarchal model of the biophysical driving force of chromatin organization: At a large scale, chromatin is compartmentalized via ubiquitous charge–charge interactions, but specific heterochromatin proteins are involved to generate local compartments that are smaller in scale but more specific function directed.

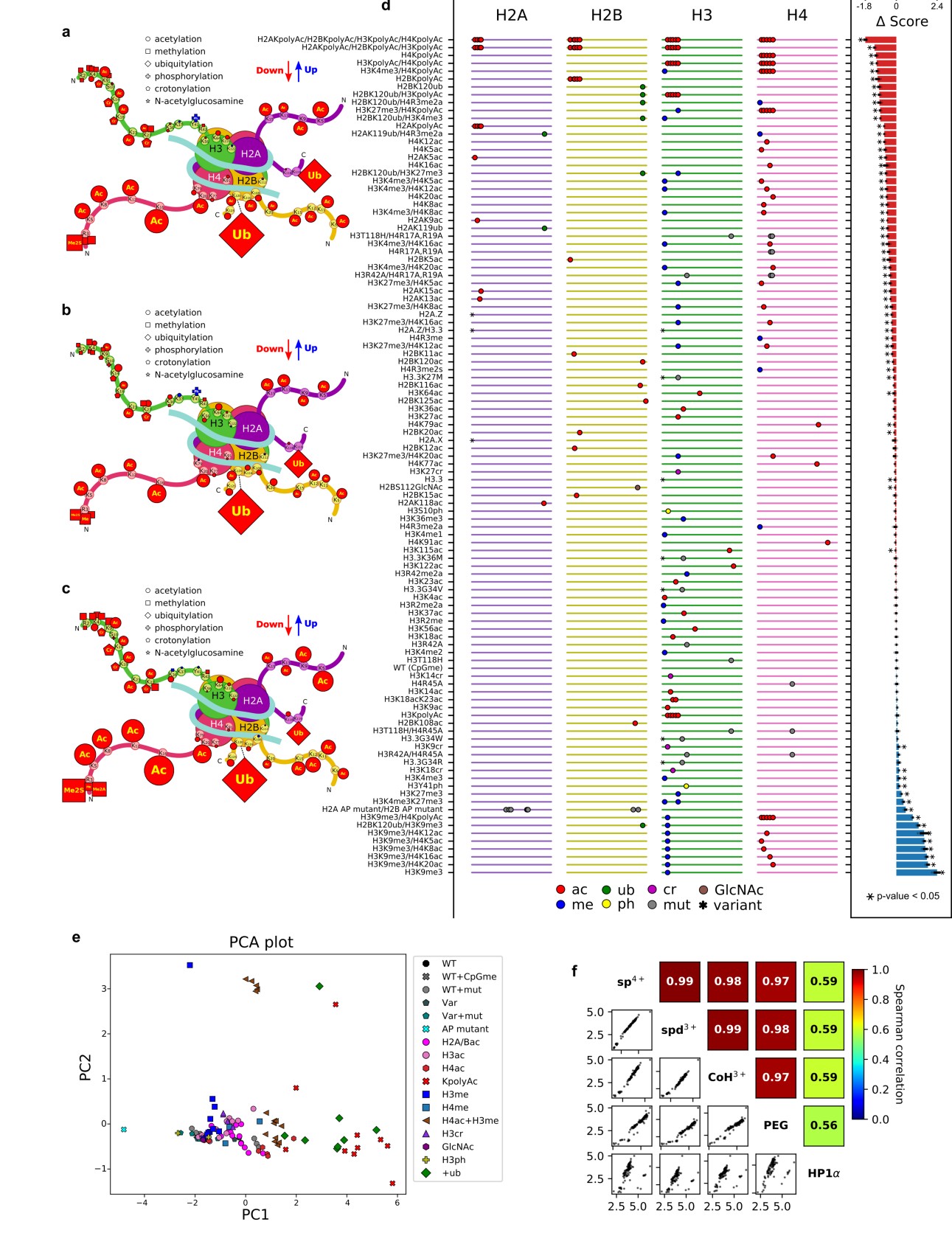

**Extended Data Fig. 9** | See next page for caption.

**Extended Data Fig. 9 | Summary of histone PTM effects on the nucleosome condensation by various condensing agents on the synthetic nucleosome library with PTM marks.** The effects of single PTMs on nucleosome condensation are depicted by the cartoons (**a**: spermidine, **b**: cobalt-hexamine, **c**: PEG 8000 as the condensing agent). Each symbol represents different types of PTMs as shown in the legend, and the size is proportional to the strength of effects. The colors of the marks indicate the direction of the effect (red: decreases condensation, blue: increases condensation) compared with the unmodified control. **d**, All condensability scores of the PTM library using HP1α as a condensing agent are summarized in the ladder bar plot. The library members are sorted from the lowest to the highest condensability scores from top to bottom. On the left panel, the ladder-like lines represent each histone subunit peptide from N-terminal (left) to the C-terminal (right). Each mark on the line indicates the location of the PTMs and the shape of the marks represents the PTM type (ac: acetylation, me: methylation, cr: crotonylation, ub: ubiquitylation, ph: phosphorylation, GlcNAc: GlcNAcylation, mut: amino acid mutation, var: histone variant). On the right panel, differences in the condensability score compared with the unmodified control are shown as bar plots for each member of the library. The asterisk on the bar-plot represents statistically significant (p-value < 0.05, and the two-sided Welch's t-test used over three biological replicates) values compared to the unmodified controls. **e**, PCA analysis was conducted by combining the condensability scores of all five condensing agents (spermine/spermidine/cobalt-hexamine/PEG 8000/ HP1α) into the five-dimensional state vector. In the PCA plot, each member of the library is represented by a symbols according to categories such as canonical wild-type nucleosome (WT), wild type with CpG methylation (WT+CpGme), mutations on wild type (WT+mut), nucleosome with histone variants (Var), mutations on histone variants (Var+mut), Acidic patch mutants (AP mutants), and nucleosomes with acetylation on H2A/B dimer (H2A/Bac), acetylation on H3 (H3ac), acetylation on (H4ac), having poly-acetylation (KpolyAC), methylation on H3 (H3me), methylation on H4 (H4me), acetylation on H4 and methylation on H3 (H4ac + H3me), crotonylation on H3 (H3cr), GlcNAcylation (GlcNAc), phosphorylation on H3 (H3ph), and ubiquitylation (+ub), all of which are shown in the figure legend. **f**, Comparison of condensability scores across different condensing agents. Scatter plots of condensability across different condensing agents are shown in the lower triangle, and the corresponding Spearman's correlations are shown in the upper triangle of the matrix.

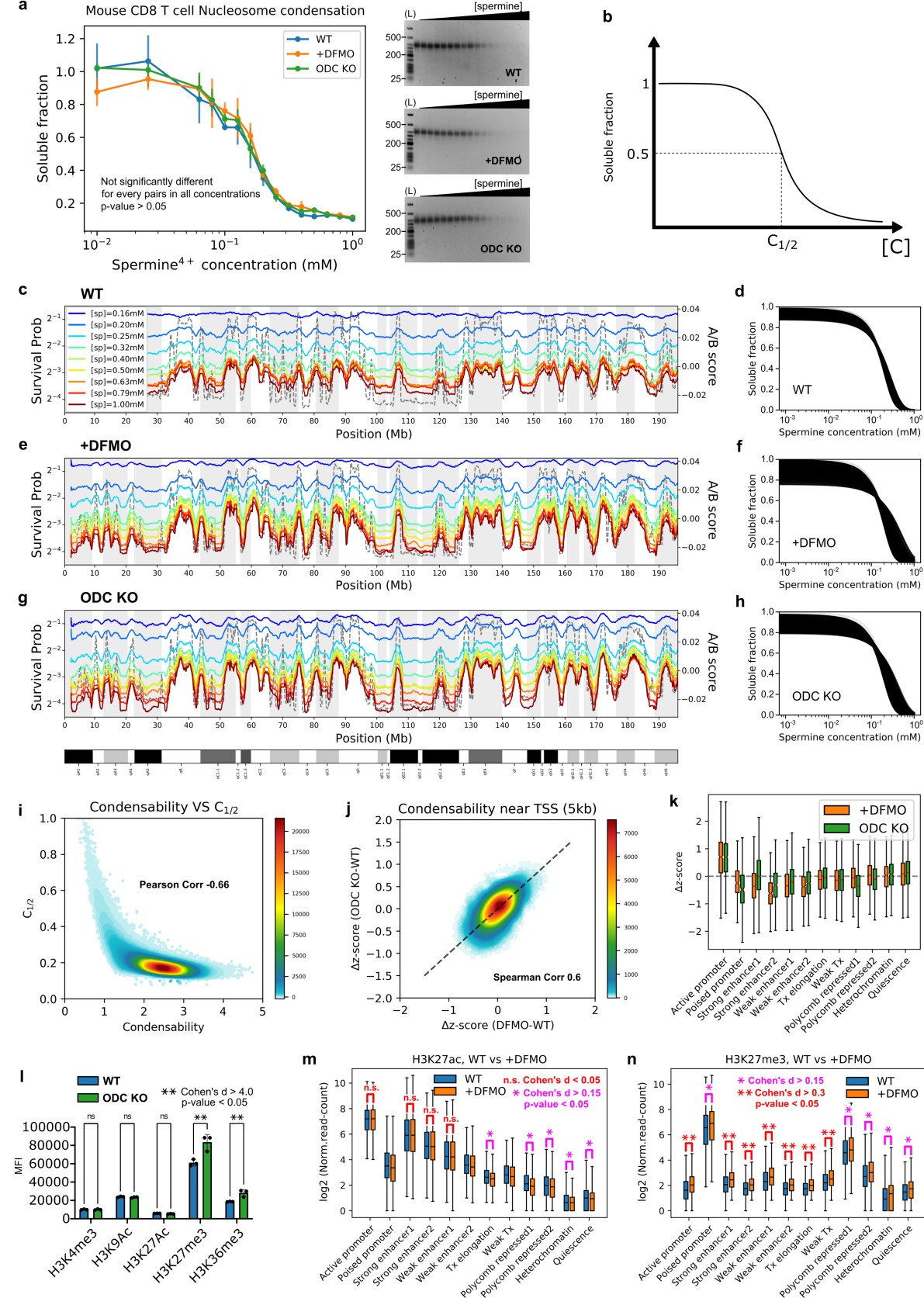

**Extended Data Fig. 10** | See next page for caption.

**Extended Data Fig. 10 | Condense-seq measurements of nucleosomes purified from mouse CD8⁺ T cells. a**, Soluble fractions were measured via UV-VIS spectroscopy and run in 2% agarose gel after condensation in various spermine concentrations (the titration curves were plotted as the mean of three replicates with error bar as the standard deviation, and there were no significant differences between wild-type/DFMO-treated/ODC-KO as shown p-value > 0.05 for any-pair). (Ladder: NEB Low Molecular Weight DNA ladder used for the first lane of gels) **b**, Condensation point ($c_{1/2}$) is defined by the concentration of condensing agent when the soluble fraction is half the input, so it is reversely correlated with condensability score. **c**–**h**, Soluble fractions of nucleosomes in various spermine concentrations were calculated and plotted over chromosome 1 in 10 kb resolution. $C_{1/2}$ was computed for each bin after fitting the soluble fraction change with a logistic function as shown fitting curves of all bins (**d**, **f**, **h**), and polyamine deficient conditions show broader distribution of condensation points. (**c**, **d**: wild type control, **e**, **f**: +DFMO, **g**, **h**: ODC KO) **i**, Condensability point ($c_{1/2}$) has inverse relationship with condensability scores of nucleosomes in mouse CD8 + T cells. **j**, The scatter plot of Δ z-score of condensability near TSS shows a high correlation between +DFMO and ODC KO. **k**, The Δ z-score of condensabilities is computed as the difference between the standardized condensability of +DFMO or ODC KO conditions and the wild type control and then categorized into the corresponding ChromHMM chromatin states over more than 300 nucleosomes of each state from two biological replicates (boxplot: the center is median and the lower/upper bound is the 1ˢᵗ/3ʳᵈ quartile of data). Flow cytometry data show the global changes of PTM marks in ODC knockout (ODC KO) CD8⁺ T cells vs wild type (WT) expressed as mean intensity fluorescence (MFI) from three biological replicates (p-values were computed via the two-sided Welch's t-test). (**l**) and also further verified using calibrated histone ChIP-seq for H3K27ac (**m**) and H3K27me3 marks (**n**) (the statistics were computed via the two-sided Welch's t-test over more than 10000 nucleosomes of each state from three biological replicates).

# Reporting Summary

Please do not complete any field with "not applicable" or n/a. Refer to the help text for what text to use if an item is not relevant to your study.
For final submission: please carefully check your responses for accuracy; you will not be able to make changes later.

## Statistics

For all statistical analyses, confirm that the following items are present in the figure legend, table legend, main text, or Methods section.

| n/a | Confirmed | |
|---|---|---|
| ☐ | ☑ | The exact sample size (*n*) for each experimental group/condition, given as a discrete number and unit of measurement |
| ☐ | ☑ | A statement on whether measurements were taken from distinct samples or whether the same sample was measured repeatedly |
| ☐ | ☑ | The statistical test(s) used AND whether they are one- or two-sided<br>*Only common tests should be described solely by name; describe more complex techniques in the Methods section.* |
| ☑ | ☐ | A description of all covariates tested |
| ☑ | ☐ | A description of any assumptions or corrections, such as tests of normality and adjustment for multiple comparisons |
| ☐ | ☑ | A full description of the statistical parameters including central tendency (e.g. means) or other basic estimates (e.g. regression coefficient) AND variation (e.g. standard deviation) or associated estimates of uncertainty (e.g. confidence intervals) |
| ☐ | ☑ | For null hypothesis testing, the test statistic (e.g. *F*, *t*, *r*) with confidence intervals, effect sizes, degrees of freedom and *P* value noted<br>*Give P values as exact values whenever suitable.* |
| ☑ | ☐ | For Bayesian analysis, information on the choice of priors and Markov chain Monte Carlo settings |
| ☑ | ☐ | For hierarchical and complex designs, identification of the appropriate level for tests and full reporting of outcomes |
| ☐ | ☑ | Estimates of effect sizes (e.g. Cohen's *d*, Pearson's *r*), indicating how they were calculated |

*Our web collection on statistics for biologists contains articles on many of the points above.*

## Software and code

Policy information about availability of computer code

| Data collection | No specific software was used for data collection |
|---|---|
| Data analysis | Bowtie2 (v2.4.1) was used for NGS read alignment, and Samtools (v1.10) was used for accessing BAM files. Python2 (v2.7.18) was utilized for all other custom analyses. Further details on data analysis and custom scripts can be found in the GitHub repository: https://github.com/spark159/condense-seq. |

For manuscripts utilizing custom algorithms or software that are central to the research but not yet described in published literature, software must be made available to editors and reviewers. We strongly encourage code deposition in a community repository (e.g. GitHub). See the Nature Portfolio guidelines for submitting code & software for further information.

## Data

Policy information about availability of data

All manuscripts must include a data availability statement. This statement should provide the following information, where applicable:
- Accession codes, unique identifiers, or web links for publicly available datasets
- A description of any restrictions on data availability
- For clinical datasets or third party data, please ensure that the statement adheres to our policy

NCBI GEO under accession number GSE252941

# Research involving human participants, their data, or biological material

Policy information about studies with human participants or human data. See also policy information about sex, gender (identity/presentation), and sexual orientation and race, ethnicity and racism.

| | |
|---|---|
| Reporting on sex and gender | NA |
| Reporting on race, ethnicity, or other socially relevant groupings | NA |
| Population characteristics | NA |
| Recruitment | NA |
| Ethics oversight | NA |

Note that full information on the approval of the study protocol must also be provided in the manuscript.

# Field-specific reporting

Please select the one below that is the best fit for your research. If you are not sure, read the appropriate sections before making your selection.

[V] Life sciences    [ ] Behavioural & social sciences    [ ] Ecological, evolutionary & environmental sciences

For a reference copy of the document with all sections, see nature.com/documents/nr-reporting-summary-flat.pdf

# Life sciences study design

All studies must disclose on these points even when the disclosure is negative.

| | |
|---|---|
| Sample size | The particular sample size calculation is not relevant for this study. However, we always numerate data size, for example, nucleosome numbers, in the figure legend for any statistical comparison. |
| Data exclusions | No data exclusion were conducted in this study |
| Replication | All biochemical studies were performed in triplicate (n=3) for all experiments. Genomic studies from cell lines were all carried out in duplicate. More specific summary is the following. |
| Randomization | 3 biological replicates (Nucleosome condensation assay), 2 biological replicates (Condense-seq with genomic nucleosomes), 3 biological replicates (Histone Chip-seq), 3 biological replicates (Immunofluorescence), 3 biological replicates (PTM library) |
| Blinding | The randomization was not required at any other stage of this study. |
| | Blinding was not necessary in this study since it was not a clinical trial involving human patients. |

# Behavioural & social sciences study design

All studies must disclose on these points even when the disclosure is negative.

| | |
|---|---|
| Study description | |
| Research sample | |
| Sampling strategy | |
| Data collection | |
| Timing | |
| Data exclusions | |
| Non-participation | |
| Randomization | |

# Ecological, evolutionary & environmental sciences study design

All studies must disclose on these points even when the disclosure is negative.

| | |
|---|---|
| Study description | |
| Research sample | |
| Sampling strategy | |
| Data collection | |
| Timing and spatial scale | |
| Data exclusions | |
| Reproducibility | |
| Randomization | |
| Blinding | |

Did the study involve field work?  ☐ Yes   ☐ No

## Field work, collection and transport

| | |
|---|---|
| Field conditions | |
| Location | |
| Access & import/export | |
| Disturbance | |

# Reporting for specific materials, systems and methods

We require information from authors about some types of materials, experimental systems and methods used in many studies. Here, indicate whether each material, system or method listed is relevant to your study. If you are not sure if a list item applies to your research, read the appropriate section before selecting a response.

### Materials & experimental systems

| n/a | Involved in the study |
|---|---|
| ☐ | ☑ Antibodies |
| ☐ | ☑ Eukaryotic cell lines |
| ☑ | ☐ Palaeontology and archaeology |
| ☐ | ☑ Animals and other organisms |
| ☑ | ☐ Clinical data |
| ☑ | ☐ Dual use research of concern |
| ☑ | ☐ Plants |

### Methods

| n/a | Involved in the study |
|---|---|
| ☐ | ☑ ChIP-seq |
| ☑ | ☐ Flow cytometry |
| ☑ | ☐ MRI-based neuroimaging |

## Antibodies

Antibodies used

Validation

Western Blot: H3 (polyclonal, abcam, Cat# ab1791), H3K4me3 (C42D8, Cell Signaling Technology, Cat# 9751), H3K9me3 (polyclonal, abcam, Cat# ab8898), H3K9ac (C5B11, Cell Signaling Technology, Cat# 9649), H3K27ac (polyclonal, abcam, ab4729), H2BK120ub (D11, Cell Signaling Technology, Cat# 5546). Chip-seq: H3K27ac (polyclonal, abcam, Cat# ab4729), H3K27me3 (EPR18607, abcam, Cat# ab192985). Immunofluorescence: H3K36me3 (polyclonal, abcam, Cat# ab9050), H3K4me3 (C42D8, Cell Signaling Technology, Cat# 9751), H3K27ac (D5E4, Cell Signaling Technology, Cat# 8173), H3K27me3 (C36B11, Cell Signaling Technology, Cat# 9733), H3K9ac (C5B11, Cell Signaling Technology, Cat# 9649)

All validation studies and other referenced research were conducted and can be found through commercial vendors, accessible via catalog numbers on their websites.

# Eukaryotic cell lines

Policy information about cell lines and Sex and Gender in Research

| | |
|---|---|
| Cell line source(s) | Cell lines used in this study: H1-hESC (WiCell), GM12878 (Coriell Institute), ES-E14TG2a (ATCC) |
| Authentication | Cell line authentication details can be found on the vendor's website. |
| Mycoplasma contamination | All cell lines used in this study were routinely tested for mycoplasma contamination and confirmed to be negative throughout the duration of the study. |
| Commonly misidentified lines (See ICLAC register) | None of the cell lines used in this study are known to be commonly misidentified. |

# Palaeontology and Archaeology

| | |
|---|---|
| Specimen provenance | |
| Specimen deposition | |
| Dating methods | |

☐ Tick this box to confirm that the raw and calibrated dates are available in the paper or in Supplementary Information.

| | |
|---|---|
| Ethics oversight | |

Note that full information on the approval of the study protocol must also be provided in the manuscript.

# Animals and other research organisms

Policy information about studies involving animals; ARRIVE guidelines recommended for reporting animal research, and Sex and Gender in Research

| | |
|---|---|
| Laboratory animals | C57BL/6 wild-type and ODC (flox/flox) mice with Cre recombinase (purchased from Jackson Laboratories) were used in this study. Mice of all strains were typically 8–12 weeks old. All mice were housed in individually ventilated microisolator cages in a facility maintained at Johns Hopkins University, on a 12 hr light/dark cycle maintained at 40-60% humidity and 72 degree F+/-2 degrees. |
| Wild animals | No wild animals were used in the study. |
| Reporting on sex | In this study, we investigated the effect of polyamine depletion on chromatin organization by comparing wild-type and knockout mice. As gender was not the focus of this study and not a critical factor, mice were randomly assigned to groups regardless of sex. |
| Field-collected samples | No field collected samples were used in the study. |
| Ethics oversight | All mice were bred and maintained under specific pathogen free conditions under protocols approved by the Animal Care and Use Committee (IACUC) of the Johns Hopkins University, Baltimore, USA, in accordance with the Guide for the Care and Use of Animals. |

Note that full information on the approval of the study protocol must also be provided in the manuscript.

# Clinical data

Policy information about clinical studies

All manuscripts should comply with the ICMJE guidelines for publication of clinical research and a completed CONSORT checklist must be included with all submissions.

| | |
|---|---|
| Clinical trial registration | |
| Study protocol | |
| Data collection | |
| Outcomes | |

# Dual use research of concern

Policy information about dual use research of concern

## Hazards

Could the accidental, deliberate or reckless misuse of agents or technologies generated in the work, or the application of information presented in the manuscript, pose a threat to:

| No | Yes | |
|---|---|---|
| ☐ | ☐ | Public health |
| ☐ | ☐ | National security |
| ☐ | ☐ | Crops and/or livestock |
| ☐ | ☐ | Ecosystems |
| ☐ | ☐ | Any other significant area |

## Experiments of concern

Does the work involve any of these experiments of concern:

| No | Yes | |
|---|---|---|
| ☐ | ☐ | Demonstrate how to render a vaccine ineffective |
| ☐ | ☐ | Confer resistance to therapeutically useful antibiotics or antiviral agents |
| ☐ | ☐ | Enhance the virulence of a pathogen or render a nonpathogen virulent |
| ☐ | ☐ | Increase transmissibility of a pathogen |
| ☐ | ☐ | Alter the host range of a pathogen |
| ☐ | ☐ | Enable evasion of diagnostic/detection modalities |
| ☐ | ☐ | Enable the weaponization of a biological agent or toxin |
| ☐ | ☐ | Any other potentially harmful combination of experiments and agents |

# Plants

| Seed stocks | |
|---|---|
| Novel plant genotypes | |
| | |
| Authentication | |

# ChIP-seq

## Data deposition

☑ Confirm that both raw and final processed data have been deposited in a public database such as GEO.

☑ Confirm that you have deposited or provided access to graph files (e.g. BED files) for the called peaks.

| Data access links
*May remain private before publication.* | NCBI GEO under accession number GSE252941 |
|---|---|
| Files in database submission | Detail information could be found at GSE252941 |
| Genome browser session
(e.g. UCSC) | no longer applicable |

## Methodology

| Replicates | 3 biological replicates |
|---|---|
| Sequencing depth | All raw and processed data deposited at GSE252941 |
| Antibodies | Detail information could be found at method section of main text |
| Peak calling parameters | Detail information could be found at method section of main text |
| Data quality | All raw and processed data deposited at GSE252941 |

| Software | Bowtie2 and custom python scripts |
|---|---|

# Flow Cytometry

## Plots

Confirm that:

☐ The axis labels state the marker and fluorochrome used (e.g. CD4-FITC).

☐ The axis scales are clearly visible. Include numbers along axes only for bottom left plot of group (a 'group' is an analysis of identical markers).

☐ All plots are contour plots with outliers or pseudocolor plots.

☐ A numerical value for number of cells or percentage (with statistics) is provided.

## Methodology

| Sample preparation | |
|---|---|
| Instrument | |
| Software | |
| Cell population abundance | |
| Gating strategy | |

☐ Tick this box to confirm that a figure exemplifying the gating strategy is provided in the Supplementary Information.

# Magnetic resonance imaging

## Experimental design

| Design type | |
|---|---|
| Design specifications | |
| Behavioral performance measures | |

| Imaging type(s) | |
|---|---|
| Field strength | |
| Sequence & imaging parameters | |
| Area of acquisition | |

Diffusion MRI ☐ Used ☐ Not used

## Preprocessing

| Preprocessing software | |
|---|---|
| Normalization | |
| Normalization template | |
| Noise and artifact removal | |
| Volume censoring | |

## Statistical modeling & inference

| Model type and settings | |
|---|---|
| Effect(s) tested | |

Specify type of analysis: ☐ Whole brain ☐ ROI-based ☐ Both

Statistic type for inference

(See Eklund et al. 2016)

Correction

## Models & analysis

| n/a | Involved in the study |
|---|---|
| ☐ ☐ | Functional and/or effective connectivity |
| ☐ ☐ | Graph analysis |
| ☐ ☐ | Multivariate modeling or predictive analysis |

Functional and/or effective connectivity

Graph analysis

Multivariate modeling and predictive analysis

