## [Peer Review File · Nature]

Native nucleosomes intrinsically encode genome organization principles

Corresponding Author: Professor Taekjip Ha

Version 0:

Reviewer comments:

Referee #1

(Remarks to the Author)

In this paper, S. Park et al. report an extremely elegant new approach, condense-seq, to study fundamental principles of genome organization. Condense-seq involves extracting and digesting nuclear chromatin to purify native mononucleosomes. Upon exposure of these native mononucleosomes to condensating agents, the authors use next-generation sequencing to derive, for each sequence, the fraction of nucleosomes in the supernatant (in comparison to the condensed precipitate) versus the input. This 'condense-seq' workflow thus yields a genome-wide measure of condensability that the authors correlate with chromatin organization. Specifically, they demonstrate a striking anticorrelation of this condensability with expression levels as per RNA-seq data. The authors demonstrate that the information contained in these mono-nucleosomes is an important determinant for sorting them into either A or B compartments. The biophysical properties of mono-nucleosomes are determined by their DNA sequence and epigenetic marks. In order to separately probe the contributions from these two factors, the authors therefore analyze the condensability of nucleosomes from two different human cell lines. Additionally, they employ a barcoded library of nucleosomes carrying post-translational modifications (PTMs) to elucidate their effect on condensability. Intriguingly, these experiments demonstrated that all PTMs included in this study attenuate condensation.

This is a well-executed study which provides extremely timely and novel information about one of the most fundamental questions in the organization of genetic information. The data in this study are of exceptional quality, and the methodology is novel and unique. I have no doubt that this will be a landmark paper and it is a pleasure to enthusiastically recommend it for publication, after the following points have been addressed:

To further corroborate the generality of the authors' important findings, the inclusion of at least some data for other eukaryotic species, such as yeast, would be of great value. The authors could also separately analyze the effects of DNA sequence by assembling unmodified nucleosomes on genomic DNA. This could be achieved either with a library of 150-bp DNA fragments or a long-fragment (e.g., plasmid) library treated with MNase.

The authors should also consider isolating the effects from different PTMs by treating the samples with the respective erasers before condensation.

Is there data on the PTMs in polyamine-depleted cells? It would be very helpful to have information on the general levels of different PTMs to understand how hyperpolarization is achieved. Perhaps this can be addressed using mass spectrometry?

Based on their findings, could the authors provide a cartoon figure illustrating how they envision that chromatin compartmentalization is achieved? How do nucleosomes physically interact in the condensed regions?

Is condensability indeed a property of an individual nucleosome, independent of other nucleosomes or free DNA present in solution? The authors should explicitly clarify this and discuss to what extent this assumption is true, and spell out potential limitations stemming from it. Given that nucleosome-nucleosome interactions probably play an important role in condensation, one could imagine that the concentration and nature of other nucleosomes could affect the condensation of a given nucleosome species. The acidic patch as an epitope might have an effect. Would condensation be affected if the surrounding nucleosomes had mutations in the acidic patch? The authors should discuss this and potentially consider addressing this point experimentally using acidic patch-mutated (APM) nucleosomes.

Some additional minor points

Abstract: "... information to form compartments is all contained in single native nucleosomes" - the authors could consider rephrasing this statement to be less categorical

"nucleosome condensability is a very meaningful axis onto which to project the high dimensional cellular chromatin state" - the authors should clarify this statement

Main text:

It would be helpful if the authors mentioned the analyzed species earlier on in the text

Line 112: "clear correlation with condensability" - here and in lines 157 and 162, it would be helpful if the correlation coefficients could be provided in the text.

Line 120: "For example, although AT content is also the lowest around TSS in genes with the highest expression, its dip is much narrower than the condensability dip"

While this statement makes sense based on visual inspection of the plot, it would be helpful to corroborate this more quantitatively.

Fig. 1c would benefit from results of statistical tests for the differences between states

Line 367 "immunology" - immunity?

Line 372: the existence of a detailed protocol in the SI should be mentioned in this section

Extended data Figure 3a: scales should to be provided for these plots in some form, same for Fig. 1c, it's not fully clear to me how the coloring scheme works.

Referee #2

(Remarks to the Author)

The authors investigate a long-standing question – namely that of how the inherent nucleosome properties lead to chromatin regulation. They take an innovative approach to this, combining an in vitro condensation assay on native nucleosomes with RNA-seq, and global chromatin data to assess the correlation of nucleosome condensability with chromatin state and gene regulation, and do so across different cell types. They amass an impressive amount of data and uncover a number of correlations between condensability, histone PTMs, genomic elements, and gene transcription that are logical within the framework of what is known regarding chromatin character. The results are intriguing and potentially of high significance, however I'm not sure they were truly able to build a model of how the biophysical properties of nucleosomes regulate chromatin structure. I also have concerns about the statistical significance of some comparatives. A minor point, but this reviewer finds the anthropomorphizing of nucleosomes to be quite distracting (e.g. the nucleosomes "know" where to go).

Major points are below:

1) It is not noted how error bars in Fig1b were calculated, neither was it noted if the two curves are statistically significantly different. This is critical because if they are not then it appears that it is largely the DNA condensability that is the determining factor.

2) Statistical significance should in fact be computed for and reported for all comparative results.

3) In relation to DNA composition results are confusing and appear contradictory. On one hand the authors argue that since condensability is seen to be cell type dependent it cannot be due to AT-rich character – however, they then see a major correlation between AT content and condensability. This seeming contradiction is not discussed.

4) The authors suggest that since changes in HP1-mediated condensability is modulated by a number of histone PTMs outside of the expected H3K9me3 that HP1 must be interacting with these if only weakly. However, it is now fairly well documented that histone tail PTMs can alter the conformational dynamics of the nucleosome including DNA accessibility. Furthermore, HP1 is known to interact with other components of the nucleosome including the DNA. All of these factors need to be considered in interpreting this data.

5) The authors conclude that the major biophysical determinant of nucleosome condensability is electrostatic character of the mono-nucleosome. However, they do not present a good test of this model.

Referee #3

(Remarks to the Author)

In this manuscript, the authors describe a new sequencing-based genomic method called condense-seq, which is aimed at measuring nucleosome condensability, i.e. ability to precipitate upon the addition of some condensing agent (e.g. a polyamine). Based on their measurements applying condense-seq to human cell lines, they calculate a condensability score, and show this score is correlated with genomic features such as chromatin accessibility, gene activity, GC content and A/B genomic compartments. They also apply condense-seq to a synthetic library of histone post-translational modifications (PTMs), and show that different PTMs have different condensability scores. In addition, the authors compare the effects of different condensing agents, including HP1 and observe a number of differences between agents. Finally, they examine a polyamine depletion mouse model and observe hyperpolarization of condensability.

I must admit that initially I found it difficult to understand what the real claims of this manuscript actually mean, as the framing was problematic. The authors write that they ask if “individual nucleosomes in vivo know where to go” and “do individual nucleosomes in the cell intrinsically encode information important for their participation in large scale organizations such as A and B compartments”. Of course it is known that nucleosomes within different compartments are associated with compartment-specific features such as specific PTMs and tend to be associated differentially with certain DNA sequences in vivo. Also, Multiple genomic assays can produce measurements that will be correlated with A/B compartments. So clearly the claim cannot be that individual nucleosomes contain information about whether they belong to A/B nor that condense-seq is unique in its ability to measure this. Thus, in my view the real claim here is that the information on individual nucleosomes is sufficient to form A/B compartment-like interactions spontaneously upon the addition of a condensing agent simply based on electrostatic interactions without any chromatin readers, chromatin remodellers, or additional investment of energy.

If this is indeed the main claim, there are two central questions for in my opinion. First, whether this claim is supported by the experiments and data, and second whether this mechanism is relevant to what actually happens in cells. Unfortunately, I think that currently I am not convinced on either of these points.

Major comments:

1. The condensability score on which much of the paper is based on is problematic. This score is calculated as $-\log(\text{supernatant}/\text{input})$ or equivalently $\log(\text{input}/\text{supernatant})$. The highest condensability score possible will be achieved if the supernatant has one read (let's assume zeros are not corrected – this is not mentioned in the methods either way), therefore it can be at most $\log(2/1)$. However, the dynamic range for low condensability is not bounded in this way. If this is correct it, it means that it would be impossible to get high condensability scores in regions which initially had low nucleosome occupancy, and create an artificial correlation between condensability and nucleosome occupancy. Is this indeed what happens? Yes, we can see a clear example in figure 2e (potentially related issues also in 3a). Suspiciously, the upper right corner, in which chromatin accessibility is high (i.e. low nucleosome occupancy) and condensability is high, is suspiciously empty of any datapoints. This is critical, as it means that even if the supernatant was just a random sample from the input, we would still get a correlation between the condensability score and nucleosome occupancy (and A/B compartments). Ultimately, it will be important to understand whether any condensability score is correlated with input MNase occupancy, as this alone can drive correlation with A/B compartments.
2. More generally, this just one example of how the analysis details, and specifically the calculation of condensability, are extremely important and must be justified and done correctly. What happens to zero values? How are missing values treated? Why are nucleosomes peaks “called” and what are the effects of this on the data? What is the justification for an additional condensability score ($C1/2$)? Currently, these details are not provided and each of these may result in an artificial correlation with other genomic features such as A/B compartments. Details are also missing from the methods in other sections (especially analysis), e.g. how were machine learning hyperparameters fit, why only 100000 nucleosomes were used, etc.
3. What is actually being measured in condense-seq? Is it actually the propensity of specific nucleosomes to precipitate? The authors take extra care to isolate mononucleosomes, which is good. But DNA can also precipitate with polyamines, and couldn't the nucleosome potentially unwrap? Perhaps crosslinking the nucleosomes to the DNA could reduce the chances of this happening.
4. The authors make conclusions based on differences between groups, but no statistical tests were performed on the results in the paper. It is also not mentioned what error bars represent (SD/SE/CI). In other cases, the authors argue that data are similar/correlated but do not provide a proper quantification (e.g. condensability vs A/B compartments).
5. The fact that using condensability score as pairwise energy can reconstruct genomic compartment patterns in interaction maps is expected given the previous work in polymer models of genome organization, it was shown several times that having some energy which favours self-interaction can create these patterns.
6. The condensability measurements on the PTM library are interesting, but they do not seem to explain well the notion that A/B compartments are mostly driven by condensability: why do A-specific PTMs have similar condensability to B-specific PTMs? Ideally, we would like to find some model that could roughly predict the condensability of in vivo nucleosomes from these synthetic scores. Could DNA sequence play a role as well? One way to learn about this might be to perform condense-seq on nucleosomes (without PTMs) that were reconstituted in vitro on genomic DNA. At this stage I think the PTM data still does not sufficiently support the hypothesis that this is what drives compartmentalization.
7. The measurements in mice attempt to show biological relevance of condensability, which is potentially a good idea, but the results are somewhat counterintuitive. If polyamines drive condensability and A/B compartments, naively one would expect that their removal would cause less compartmentalization and condensability (by the way, Hi-C was not measured here so we do not know how genome organization changed). It is possible that the cell tries to compensate for this effect as the authors suggest, but there is no direct evidence for this here. I believe that currently the results of this experiment do not provide strong support for the authors' hypothesis.

8. As I mentioned, I suggest stating the hypotheses and claims more accurately so they would be easier for the reader to understand. It may also be appropriate to tone down some of the more speculative conclusions such as “thus, electrostatic interaction between nucleosomes mediated by multivalent ions is probably the main driving force behind large-scale genomic compartmentalization”.

Minor comments:

9. Extended Figure 5C is not really explained in the main text.
10. Why doesn't HP1 condensation show AT content as a strong factor like spermine? This seems counterintuitive.
11. “the correlation was biphasic due to the sparsity of accessible regions revealed by ATAC-seq” – this was unclear to me.
12. The text should be checked for typos (e.g. “chromatins”, “multilayer perception”).

(Remarks on code availability)

I reviewed the code only briefly, but I found the documentation to be lacking. I was hoping that the analysis details explaining how things were calculated - which did not appear in the Methods section - would at least be explained in the code documentation but this was not the case. I did not try to run the code myself.

Referee #4

(Remarks to the Author)

The manuscript by Park et al. reports a new deep sequencing method they refer to as condense-seq. Briefly, their approach as I understand it is to use micrococcal nuclease to digest chromatin from cell lines down to single nucleosomes and then purify them. They condense the single nucleosomes with different reagents, then centrifuge the sample and subject the DNA in the pellet to deep sequencing. This allows them to map the propensity of genomic regions to condense in the form of single nucleosomes. The authors refer to this as the intrinsic condensability of single nucleosomes. They compare the condensability to a wide range of published mapped features. They find that the reduction in condensability correlates with active genomic regions including promoters and enhancers, while increase condensability correlates with silenced genomic regions such as heterochromatin and insulators. Accessibility as measured by ATACseq also correlates with the reduction in accessibility as well as A/B compartmentalization as measured by HiC. To better understand how histone PTMs influence accessibility they used a nucleosome library of different histone PTM combinations and determined that histone acetylation largely determines condensability. However, other PTMs such as ubiquitylation significantly reduces condensability as well. They then investigated the impact of disrupting the expression of polyamines on condensability since polyamines help with global condensation of chromatin. They find that the range of condensability is significantly increased when polyamines are not expressed, which indicates that the cell responds to this mutation by increasing nucleosome condensability to compensate for the loss of polyamines.

Overall, this is an extensive study that provides strong evidence that their method condense-seq maps the genomes propensity to condense, which will certainly be another impactful deep sequence method for genomic characterization. The method is synergistic with Chip-seq, ATAC-seq, HiC, and other genome mapping techniques. It is also interesting how different compacting reagents can be used, which could be used to characterize condensability by different distinct chromatin architectural proteins and small molecules. This is an important new and elegant method that will likely be impactful and useful for studying different cell types, mutations, and disease. However, there are issues with the presentation of the work and their conclusions, which I list below.

- 1) A primary concern is the authors repeatedly make bold and general conclusions in the paper. Here are some examples:
 - a. These findings are surprising because they indicate that single native nucleosomes ‘know’ if they are in highly transcribed regions or gene promoters by way of their reduced condensability and vice versa.
 - b. Overall, our results imply that the native mono-nucleosomes intrinsically possess, even in the absence of other factors, essentially all the biophysical properties needed for the large-scale A/B compartmentalization of the eukaryote genome.
 - c. Thus, electrostatic interaction between nucleosomes mediated by multivalent ions is probably the main driving force behind large-scale genomic compartmentalization.

These are bold conclusions and implicit to these conclusions is that there is a cause relationship between the mononucleosome condensability and other features such as transcription level and compartmentalization. However, based on my understanding of these studies, the results only provide correlations between these features. Also, it is possible that other factors that are driving chromatin compaction then influence histone acetylation. For example, the linker histone H1 can influence the level of histone acetylation (PMID: 10611231). The authors need to adjust the text to reflect that they have not shown that electrostatic driven condensation is the driver of these genomic features, but instead are correlated with them.

- 2) Another concern, which is connected to the first bullet point, is that there are numerous papers that have shown that histone acetylation influences chromatin compaction. So, it is not that surprising that electrostatics and histone acetylation is a key determinant of mononucleosome condensability. In fact, previous liquid crystal and TEM studies have shown that mononucleosomes compact similarly to that of nucleosome arrays, where mononucleosomes make long columnar structures and are packed face to face (PMID: 16973479). This is most likely due to the H4 tail interacting with the acidic patch of an adjacent nucleosome. Therefore, these condense-seq studies nicely confirm previous studies that have shown electrostatics is key to condensation and that the neutralization of positive charge by acetylation reduces the propensity of

chromatin to compact. The text should be adjusted to reflect better what is known about how histone acetylation influences chromatin and nucleosome compaction.

3) The authors should consider alternative interpretations on their HP1 measurements where they find that HP1 has a different and smaller impact on condensability than spermine. HP1 has been shown to compact chromatin through interactions with adjacent nucleosomes that contain H3K9me3 (PMID: 32493764). The experiments with mononucleosomes will largely prevent the assembly of HP1 compacted chromatin because there are not chains of nucleosomes with H3K9me3. This should be added and expanded on in the manuscript.

4) The authors should address how compaction of single nucleosomes is different than the compaction of chromatin where nucleosomes are contained within a single fiber. For example, it is known that the DNA repeat spacing between nucleosome significantly impacts chromatin compaction (PMID: 30630950). The authors should address in the manuscript that nucleosome spacing is another regulator of chromatin compaction, which could function synergistically and antagonistically with histone acetylation.

5) It is interesting they have evidence that methylation might directly influence compaction. In fact, methylation on H4K20 has been shown to directly induce decompaction within nucleosome arrays (PMID: 34417450) by disrupting the H4 tail conformation and thus its interaction with the acidic patch. However, they do not seem to observe reduced compaction by spermine with the ENCODE mapped H4K20me1 PTM. This suggests direct impact of this methylation might impact condensation within chains of nucleosomes but not within isolated mononucleosomes. Both this example and the mono vs. polynucleosomes should be discussed in the manuscript.

In addition to these major issues, I list below some minor issues.

1) In many of the main and supplemental figures the text is very small and nearly impossible to read. Please improve this. I found it very frustrating when I was trying to read the text in the figures.

2) Figure caption 1, has H3K25ac in the text. I assume they mean H3K27ac.

Additional feedback:

1) One natural extension of these studies is to carry out condense-seq with dinucleosomes, trinucleosomes, etc. It would be extremely interesting to compare condensability of different repeat numbers. I am not suggesting this be done for this paper, which already contains a tremendous number of results. But this work provides the foundation for further important studies of chromatin compaction. This is another reason why the authors should adjust their conclusions of the single nucleosome studies. There is tremendous potential for using this method of studying short sections of chromatin.

2) I am surprised the authors do not map the positions of nucleosomes from their data. It would be interesting to learn how the condensed nucleosomes are positioned relative to input and the soluble fraction. I don't think this needs to be done. But it would be interesting to investigate and should come along without the need for additional experiments.

Version 1:

Reviewer comments:

Referee #1

(Remarks to the Author)

I commend the authors for their exceptionally thorough and thoughtful responses to the reviewers' comments. They have addressed every concern comprehensively, providing clear, well-supported explanations that further elevate an already outstanding manuscript.

Referee #2

(Remarks to the Author)

The authors have addressed all of my original comments.

Referee #3

(Remarks to the Author)

The authors have done an excellent job in clarifying their claims and findings. The additional experiments and analysis are also very helpful in addressing some of the points I made. Specifically, the nucleosome unwrapping experimental controls (Extended Figure 1) and the in vitro reconstitution experiments (Ext. Data Fig. 7) are excellent. I am still not convinced that this actually happens in cells (e.g. the results in mice are very indirect), but the bottom line for me is that the findings are sufficiently supported, interesting and thought-provoking for publication.

One issue which is important to clarify:

I mentioned that the condensability scores are constrained by the input levels and the authors thought I misunderstood something, but I still think I am correct here: condensability is defined by the authors as $-\log(\text{nucleosome counts in supernatant} / \text{nucleosome counts in input control})$, and they say the ratio is always between 0 and 1, so the condensability is always positive. First, note that Ext. Data Fig. 2b (which explains how condensability is calculated) shows a negative score, but let's assume this is a typo. If the ratio is always between 0 and 1, it means that supernatant counts cannot be higher than input counts. Now let's consider a case in which there is low input, say only 1 read. Now, the supernatant can have 0 or 1 reads, and after adding the pseudocounts we get that the condensability in this case can be $-\log((0+1)/(1+1))$ or $-\log((1+1)/(1+1))$. In this case, $\log(2)$ is the highest condensability value that can be achieved when the number of input reads is 1. Similarly, when the input counts is 2, the highest condensability is $-\log((0+1)/(2+1))$, and so on. This means that when the input is low, you can't get high condensability values, and this creates an artificial correlation which could seriously bias the data as I explained previously.

To save us all time in back-and-forth, I took the liberty of performing the following small simulation: I randomly generated 10,000 input nucleosome read count values, and then randomly generated 10,000 supernatant values where the only constraint is that they cannot be higher than the input values. I then calculated the nucleosome condensability scores, and then the Pearson correlation between the input and the condensability. The resulting correlation was around $r = \sim 0.1$. Given these results, which I think are in line with the correlation the authors observe with the input, I think this concern can be put aside. However, it would be useful to add this type of analysis to the paper in case other readers are also concerned about how strongly this artifact can bias the condensability score.

(Remarks on code availability)

The code looks well-documented and organized. I did not run it though.

Referee #4

(Remarks to the Author)

The authors have extensively revised the manuscript, which has significantly improved the manuscript.

I will not repeat the summary provided. However, I think it is worth pointing out that there are two major outcomes of this work. The first outcome is that native mononucleosomes can be condensed into A and B compartments without additional factors. This reveals that features within single nucleosomes play an important role in genome organization. I recommend the authors consider pointing out that their observation is synergistic with SAXS and super-resolution imaging that indicates that the 30 nm chromatin fiber does not form in cells since they find that connectivity is not critical for heterochromatin-associated nucleosomes to condense more readily than euchromatin-associated nucleosomes. The second aspect is that Condense-seq could develop into an important technique for genome profiling since it appears to be a straightforward sequencing technique. My sense is that Condense-seq is synergistic with other genome organization methods, including Hi-C, MNase-seq, and ATAC-seq. This aspect is not discussed much and could be emphasized more.

One final note is that the authors state it would be difficult to purify di- and tri-nucleosomes. Sucrose gradients should work well for purifying mono-, di-, and tri-nucleosomes. They would just need to optimize the MNase digestion. The authors might consider trying this in their future studies.

Response to reviewer comments. Author responses are in blue. Updated text is green.

Referees' comments:

Referee #1 (Remarks to the Author):

In this paper, S. Park et al. report an extremely elegant new approach, condense-seq, to study fundamental principles of genome organization. Condense-seq involves extracting and digesting nuclear chromatin to purify native mononucleosomes. Upon exposure of these native mononucleosomes to condensating agents, the authors use next-generation sequencing to derive, for each sequence, the fraction of nucleosomes in the supernatant (in comparison to the condensed precipitate) versus the input. This 'condense-seq' workflow thus yields a genome-wide measure of condensability that the authors correlate with chromatin organization. Specifically, they demonstrate a striking anticorrelation of this condensability with expression levels as per RNA-seq data. The authors demonstrate that the information contained in these mono-nucleosomes is an important determinant for sorting them into either A or B compartments. The biophysical properties of mono-nucleosomes are determined by their DNA sequence and epigenetic marks. In order to separately probe the contributions from these two factors, the authors therefore analyze the condensability of nucleosomes from two different human cell lines. Additionally, they employ a barcoded library of nucleosomes carrying post-translational modifications (PTMs) to elucidate their effect on condensability. Intriguingly, these experiments demonstrated that all PTMs included in this study attenuate condensation.

This is a well-executed study which provides extremely timely and novel information about one of the most fundamental questions in the organization of genetic information. The data in this study are of exceptional quality, and the methodology is novel and unique. I have no doubt that this will be a landmark paper and it is a pleasure to enthusiastically recommend it for publication, after the following points have been addressed:

//Thank you very much for concisely summarizing our findings and highlighting the importance and novelty of our discoveries.

To further corroborate the generality of the authors' important findings, the inclusion of at least some data for other eukaryotic species, such as yeast, would be of great value. The authors could also separately analyze the effects of DNA sequence by assembling

unmodified nucleosomes on genomic DNA. This could be achieved either with a library of 150-bp DNA fragments or a long-fragment (e.g., plasmid) library treated with MNase.

//Thank you very much for your great suggestions. In response, (1) we performed condense-seq analysis of another eukaryotic species, in this case, of mouse embryonic stem cells (E14 mESC), and (2) conducted condense-seq on reconstituted nucleosomes with ~150-bp genomic nucleosomal DNA fragments from human lymphocytes (GM12878) and recombinant PTM-free histone octamers.

(1) Condense-seq of mouse E14 mESC (included as new Extended Data Figure 3 d-g). We found that mESCs exhibit nucleosome condensability characteristics very similar to those of H1 human embryonic stem cells (hESCs), such as condensability changes associated with ChromHMM chromatin states (Extended Figure 3e) and the correlation between nucleosome condensability and gene expression near TSSs (Extended Data Figure 3f,g). We added the following to the main text.

“We also applied condense-seq to mouse embryonic stem cells (E14 mESC) and found similar results, such as the dependence of condensability on chromatin states, anti-correlation between condensability and gene expression, and its cell type specificity (Extended Data Fig.3 d-g).”

(2) Condense-seq of the reconstituted nucleosomes (included as new Extended Data Figure 7). We purified mono-nucleosomal genomic DNA from GM12878 cells following MNase treatment and reconstituted the nucleosomes using recombinant PTM-free histone octamers (new Extended Data Figure 7a). After extensive optimization of this new protocol, we applied several size-selection steps to obtain pure reconstituted nucleosome products (new Extended Data Figure 7c). Our analysis revealed that reconstituted nucleosomes composed of genomic DNA but completely free of PTMs exhibited a dramatic loss of condensability changes associated with chromatin states (new Extended Data Figure 7d), correlations with gene expression (new Extended Data Figure 7f), and genome-wide fluctuations (new Extended Data Figure 7e). These effects were particularly pronounced when compared to native nucleosomes from GM12878 cells (also presented in Extended Data Fig. 7). This data clearly indicates that histone PTMs are among the most critical factors in determining the condensability of native genomic nucleosomes. We added the following to the main text.

“Next, to examine the effects of genomic DNA sequences on nucleosome condensation, we synthesized a ‘reconstituted’ nucleosome library composed of the genomic nucleosomal DNA purified from GM12878 cells reconstituted with

recombinant canonical histone octamers which are devoid of PTMs (Extended Data Fig. 7a,b). Remarkably, the reconstituted nucleosomes show higher condensability overall compared to native nucleosomes (Extended Data Fig. 7c) and lost the chromatin state dependence (Extended Data Fig. 7d). They also lost the correlation with gene expression in genome-wide scale (Extended Data Fig. 7e) and individual genes near TSS (Extended Data Fig 7f). These results show the primary importance of histone PTMs for determining genomic nucleosome condensability.”

The authors should also consider isolating the effects from different PTMs by treating the samples with the respective erasers before condensation.

//Although we appreciate this suggestion, we do not believe such an experiment is necessary because our data already showed that PTMs are important for condensability based on the comparison to the PTM-based ChromHMM classification (Figure 1 c), the cell-type dependent condensability (Figure 1 f), and synthetic histone PTM library experiment (Figure 3 d-f). Furthermore, our new condense-seq data of the reconstituted nucleosome with genomic DNA and recombinant PTM-free histone octamers, described above, clearly showed the primary importance of PTMs (new Extended Data Figure 7).

Is there data on the PTMs in polyamine-depleted cells? It would be very helpful to have information on the general levels of different PTMs to understand how hyperpolarization is achieved. Perhaps this can be addressed using mass spectrometry?

//Thank you very much for this great suggestion. Indeed, we were wondering what the epigenetic basis is for the observed hyperpolarization. In response to this suggestion, we performed ChIP-seq for histone modifications after polyamine depletion using the ODC inhibitor DFMO (new Extended Data Figure 10 m, n). We found that K27me3 is enhanced in the genomic regions where they were enriched before the treatment, consistent with the hyperpolarization we proposed (new Extended Data Figure 10 n). We also observed that K27ac is not affected significantly (new Extended Data Figure 10 m). As an alternative way of polyamine depletion, we tested ODC flox to knockout ODC in T cells and used immunofluorescence flow cytometry to show that they too exhibit similar behaviors for K27Ac and K27me3 as in inhibitor treated cells (new Extended Data Figure 10 l). In addition, the ODC KO cells allowed us to look at the global changes in several other epigenetic markers. In particular, we observed a global increase in K36me3, a marker that correlates well with high condensability. Overall, our new data provides information about epigenetic basis for hyperpolarizability. We added the following to the main text.

“The importance of the H3K27me3 mark is validated by histone PTM immunostaining screen using flow-cytometry that showed a global increase of H3K27me3 in ODC KO cell lines which also showed a global increase of H3K36me3 (Extended Data Fig. 10l). This is further analyzed by calibrated ChIP-seq experiments, which show a small but significant increase of H3K27me3 upon DFMO treatment, more so at active chromatin regions, while H3K27ac level is almost unchanged with very mild decrease at heterochromatin regions (Extended Data Fig. 10m,n).”

Based on their findings, could the authors provide a cartoon figure illustrating how they envision that chromatin compartmentalization is achieved? How do nucleosomes physically interact in the condensed regions?

//We revised text and included a figure (new Extended Data Figure 8 e) to clarify how we think about this point.

Is condensability indeed a property of an individual nucleosome, independent of other nucleosomes or free DNA present in solution? The authors should explicitly clarify this and discuss to what extent this assumption is true, and spell out potential limitations stemming from it. Given that nucleosome-nucleosome interactions probably play an important role in condensation, one could imagine that the concentration and nature of other nucleosomes could affect the condensation of a given nucleosome species. The acidic patch as an epitope might have an effect. Would condensation be affected if the surrounding nucleosomes had mutations in the acidic patch? The authors should discuss this and potentially consider addressing this point experimentally using acidic patch-mutated (APM) nucleosomes.

//This is a very good point. Acid patch mutated nucleosomes were already in the barcoded nucleosome library we used (Figure 4e, labeled as “H2A/H2B AP mutant” in the tabular bar-plot). The data indeed showed that acidic patch mutant nucleosome is one of the top modifications increasing the condensability compared to unmodified nucleosomes. We revised the text to include the following.

“Intriguingly, the charge-swap mutations on the acidic patch of histone H2A/B, which was previously suggested to be the nucleosome–nucleosome interaction

interface, induced the largest condensability increase among the PTM library members for all ionic condensing agents (Fig 3e).”

We also tested the possibility that when nucleosomes condense, DNA molecules depart from the histone core so that what’s spun down during the protocol is primarily naked DNA molecules (new Extended Data Figure 1f-h). When we redissolved the condensed fraction and ran a gel, we could show that most of the molecules show up as intact mononucleosomes (new Extended Data Figure 1h). This was true for native nucleosomes from mESCs and GM12878 as well as reconstituted nucleosomes made from GM12878 nucleosomal DNA and unmodified histone octamers. Therefore, we conclude that each nucleosome maintains its mononucleosome status even when it becomes part of a condensate in the presence of other nucleosomes. We added the following to the main text.

“We showed that the native mono-nucleosomes remain intact upon condensation and used single molecule FRET to show that spermine, at concentrations that induce formation of large nucleosome condensates, does not induce significant unwrapping of nucleosomal DNA (Extended Data Fig. 1).”

Some additional minor points

Abstract: “... information to form compartments is all contained in single native nucleosomes” - the authors could consider rephrasing this statement to be less categorical

//We agree with the suggestion and made a change as follow.

“In silico chromatin polymer simulations using condensability as the only input showed that much of biophysical information needed to form compartments is contained in single native nucleosomes even in the absence of other factors.”

“nucleosome condensability is a very meaningful axis onto which to project the high dimensional cellular chromatin state” - the authors should clarify this statement

//Another great suggestion. We expanded on this in the main text as follows. In addition, we changed 'very meaningful axis' to 'natural axis' in the abstract.

“In the cellular context, because genomic nucleosomes are decorated with the combinations of multiple PTMs and cytosine methylation in different sequence contexts as shown in the NMF clustering (Extended Data Fig. 5e), nucleosome condensation properties are likely to be a complex emergent outcome of the combined effects of the individual genetic and epigenetic features. If so, we may conclude that nucleosome condensability is a natural axis onto which to project the high dimensional cellular chromatin state.”

Main text:

It would be helpful if the authors mentioned the analyzed species earlier on in the text

//We have mentioned the species for which condense-seq was performed at the end of introduction as follows.

“To address these questions, we developed an assay to measure their intrinsic condensability mediated by physiological condensing agents and applied the assay to human and mouse embryonic stem cells and differentiated cells.”

Line 112: “clear correlation with condensability” - here and in lines 157 and 162, it would be helpful if the correlation coefficients could be provided in the text.

//We have provided the numbers in figure (new Figure 1d) and the text as follows.

“Gene expression, as reported by RNA-seq, shows a clear anticorrelation with condensability (Spearman correlation = -0.8).”

Line120: “For example, although AT content is also the lowest around TSS in genes with the highest expression, its dip is much narrower than the condensability dip”

While this statement makes sense based on visual inspection of the plot, it would be helpful to corroborate this more quantitatively.

//We provided the numeric values in the text for easy comparison as follows.

“For example, although AT content is also the lowest around TSS in genes with the highest expression, its dip is approximately two-fold narrower than the condensability dip (Fig. 1e).”

Fig. 1c would benefit from results of statistical tests for the differences between states

//We have provided the statistical significance (p-value) as well as the effect size (Cohen’s d) values for comparing condensability in every pair of ChromHMM states (new Figure 1c and new Extended Data Figure 2i).

Line 367 “immunology” - immunity?

//We fixed it.

Line 372: the existence of a detailed protocol in the SI should be mentioned in this section

//Thank you. We have done that.

Extended data Figure 3a: scales should to be provided for these plots in some form, same for Fig. 1c, it’s not fully clear to me how the coloring scheme works.

//We have provided scales for both condensability scores (blue) and RNA-seq level (red) in revised manuscript (new Extended Data Figure 3a). We also provided color-scale for ChromHMM emission matrix values (new Figure 1 c).

Referee #2 (Remarks to the Author):

The authors investigate a long-standing question – namely that of how the inherent nucleosome properties lead to chromatin regulation. They take an innovative approach to this, combining an in vitro condensation assay on native nucleosomes with RNA-seq, and global chromatin data to assess the correlation of nucleosome condensability with chromatin state and gene regulation, and do so across different cell types. They amass an impressive amount of data and uncover a number of correlations between condensability, histone PTMs, genomic elements, and gene transcription that are logical within the framework of what is known regarding chromatin character. The results are intriguing and potentially of high significance, however I'm not sure they were truly able to build a model of how the biophysical properties of nucleosomes regulate chromatin structure. I also have concerns about the statistical significance of some comparatives. A minor point, but this reviewer finds the anthropomorphizing of nucleosomes to be quite distracting (e.g. the nucleosomes "know" where to go).

//Thank you very much for your enthusiasm and constructive critique. We removed the anthropomorphizing in the revised manuscript.

Major points are below:

1) It is not noted how error bars in Fig1b were calculated, neither was it noted if the two curves are statistically significantly different. This is critical because if they are not then it appears that it is largely the DNA condensability that is the determining factor.

//We provided statistical significance (p-value) for comparing two titration curves (new Figure 1b, new Extended Data Fig. 7c, 10a).

2) Statistical significance should in fact be computed for and reported for all comparative results.

//We have revised the manuscript to give full statistical analyses, including the statistical significance (p-value) as well as effect size (Cohen's d), to all comparative graphs presented in the revised manuscript.

3) In relation to DNA composition results are confusing and appear contradictory. On one hand the authors argue that since condensability is seen to be cell type dependent it

cannot be due to AT-rich character – however, they then see a major correlation between AT content and condensability. This seeming contradiction is not discussed.

//We thank the reviewer for this comment. From the correlation data between condensability and AT content of DNA, we did not mean to claim that the condensability is directly determined by AT content, and we should have made it clear. Even though the conditional correlation analysis (Figure 3 b) and the boosting analysis (new Extended Data Figure 5 h) on the genomic nucleosome data show AT content is one of the strongest determinants for condensability, AT content is not the sole determinant of condensability as shown by the strong cell type dependence (Figure 1 f). In addition, our new data with reconstituted nucleosome clearly show the primary importance of PTMs in determining nucleosome condensability (new Extended Data Fig. 7). The AT content dependence is at least in part due to the confounding effects of various genetic and epigenetic factors since genomic nucleosomes with low AT content tends to be more highly decorated with acetylation and vice-versa (new Extended Data Fig. 5e). In new revised manuscript, we added new Supplementary Note 4 to discuss these points in a single location (reproduced below).

“We could see a clear correlation between nucleosome condensability and AT content of DNA in many data sets, such as the simple correlation (Extended Data Fig. 5a), conditional correlation (Fig. 3 b), boosting analysis (new Extended Data Fig. 5h), the meta-gene profile near TSS (Fig. 1e). However, these correlations do not mean that AT content alone determines the nucleosome condensability. As shown by the strong cell type dependence (Fig. 1f), DNA sequence cannot be a sole determinant of condensability. In addition, the condense-seq data with reconstituted nucleosome clearly show the primary importance of PTMs in determining nucleosome condensability (Extended Data Fig. 7). The AT content dependence is at least in part due to the confounding effects of various genetic and epigenetic factors since genomic nucleosomes with low AT content tend to be more highly decorated with acetylation and vice-versa (Extended Data Fig. 5e).”

4) The authors suggest that since changes in HP1-mediated condensability is modulated by a number of histone PTMs outside of the expected H3K9me3 that HP1 must be interacting with these if only weakly. However, it is now fairly well documented that histone tail PTMs can alter the conformational dynamics of the nucleosome including DNA accessibility. Furthermore, HP1 is known to interact with other components of the nucleosome including the DNA. All of these factors need to be considered in interpreting this data.

//We agree with the reviewer and apologize for giving the wrong impression. We revised the text accordingly as shown below.

“Interestingly, regardless of PTM type, most PTMs on the H3 tail also show a slight increase in HP1-induced condensation, and this trend is stronger at locations farther from the nucleosome core. This finding suggests that HP1 α could also recognize other PTMs on the H3 tail in a non-specific manner, and/or that these H3 tail modifications may also affect nucleosome dynamics, thus indirectly influencing interactions with HP1 α (PMID: 25424540). Other than H3 tail modifications, most PTMs showed similar effects between HP1 α and ionic agents, reducing condensability.”

5) The authors conclude that the major biophysical determinant of nucleosome condensability is electrostatic character of the mono-nucleosome. However, they do not present a good test of this model.

//We thank the reviewer for this comment which allowed us to present the data in a logical manner in the revision to support the statement that the major biophysical determinant of condensability is the electrostatic character of the mono-nucleosomes. In the synthetic histone PTM library data (Figure 3 d-f), we showed that acetylation has the largest effect of decreasing condensation among the small chemical modifications and, especially, poly-acetylation gives the strongest effects. This strong acetylation dependency trend is also further consistent with our mass spectrometry data (new Extended Data Figure 6a,b) and conditional correlation analysis on genomic nucleosome (Figure 3b). Furthermore, we showed that acidic patch mutations that reverse the charge from negative to positive of the histone core greatly increases the condensability (“H2A/H2B AP mutant” in tabular bar plot of Figure 3 e). Finally, condensability scores are cross-correlated with each other when we used different polyamines (spermine vs spermidine) or non-polyamine-based multivalent cations such as Cobalt hexamine (Figure 3a, new Extended Data Figure 8b, and new Extended Data Figure 9f), which are condensing agents known for charge-charge interaction-based mechanism. We have revised the text throughout to make these points clearer and created a supplementary text (Supplementary Note 4, reproduced below).

“Our condense-seq data using purified native genomic nucleosomes and synthetic histone PTM library using a variety of condensing agents suggest that electrostatic interactions appear to be a major determinant for nucleosome condensability for the following reasons.

(1) In the synthetic histone PTM library data (Fig. 3d-f), acetylation has the largest effect of decreasing condensation among all possible modifications (except ubiquitylation) and, especially, poly-acetylation gives the strongest effects. This strong acetylation dependency trend is also further consistent with our mass spectrometry data (Extended Data Fig 6a,b) and conditional correlation analysis on genomic nucleosome (Fig. 3b). Because acetylation removes a positive charge, our data support the importance of electrostatic interactions in determining nucleosome condensability.

(2) The acidic patch mutations that reverse the charge from negative to positive of the histone core greatly increase the condensability (“H2A/H2B AP mutant” in tabular bar plot of Fig. 3e), further supporting the electrostatic basis for nucleosome condensability.

(3) Condensability scores are cross-correlated with each other when we used different polyamines (spermine vs spermidine) or non-polyamine-based multivalent cations such as Cobalt hexamine (Fig. 3a, Extended Data Fig 8b, and Extended Data Fig. 9f), which are condensing agents known for charge-charge interaction-based mechanism. All this data supports the idea that electrostatics, which is a ubiquitous force regardless of the identity of protein interactors, is a major determinant of nucleosome condensability.

Combined with our observation that polymer simulations using nucleosome condensability as the sole input can reproduce the large-scale genome organization into A and B compartments, our data indicate that the electrostatic character of the mono-nucleosome determines nucleosome condensability and thereby the formation of A/B compartments.”

Referee #3 (Remarks to the Author):

In this manuscript, the authors describe a new sequencing-based genomic method called condense-seq, which is aimed at measuring nucleosome condensability, i.e. ability to precipitate upon the addition of some condensing agent (e.g. a polyamine). Based on their measurements applying condense-seq to human cell lines, they calculate a condensability score, and show this score is correlated with genomic features such as chromatin accessibility, gene activity, GC content and A/B genomic compartments. They also apply condense-seq to a synthetic library of histone post-translational modifications (PTMs), and show that different PTMs have different condensability scores. In addition, the authors compare the effects of different condensing agents, including HP1 and observe a number of differences between agents. Finally, they examine a polyamine depletion mouse model and observe hyperpolarization of condensability.

I must admit that initially I found it difficult to understand what the real claims of this manuscript actually mean, as the framing was problematic. The authors write that they ask if “individual nucleosomes in vivo know where to go” and “do individual nucleosomes in the cell intrinsically encode information important for their participation in large scale organizations such as A and B compartments”. Of course it is known that nucleosomes within different compartments are associated with compartment-specific features such as specific PTMs and tend to be associated differentially with certain DNA sequences in vivo. Also, Multiple genomic assays can produce measurements that will be correlated with A/B compartments. So clearly the claim cannot be that individual nucleosomes contain information about whether they belong to A/B nor that condense-seq is unique in its ability to measure this. Thus, in my view the real claim here is that the information on individual nucleosomes is sufficient to form A/B compartment-like interactions spontaneously upon the addition of a condensing agent simply based on electrostatic interactions without any chromatin readers, chromatin remodellers, or additional investment of energy.

//We thank the review for crystallizing our main proposal in such an elegant format.

If this is indeed the main claim, there are two central questions for in my opinion. First, whether this claim is supported by the experiments and data, and second whether this mechanism is relevant to what actually happens in cells. Unfortunately, I think that currently I am not convinced on either of these points.

Major comments:

1. The condensability score on which much of the paper is based on is problematic. This score is calculated as $-\log(\text{supernatant}/\text{input})$ or equivalently $\log(\text{input}/\text{supernatant})$. The highest condensability score possible will be achieved if the supernatant has one read (let's assume zeros are not corrected – this is not mentioned in the methods either way), therefore it can be at most $\log(2/1)$. However, the dynamic range for low condensability is not bounded in this way. If this is correct it, it means that it would be impossible to get high condensability scores in regions which initially had low nucleosome occupancy, and create an artificial correlation between condensability and nucleosome occupancy. Is this indeed what happens? Yes, we can see a clear example in figure 2e (potentially related issues also in 3a). Suspiciously, the upper right corner, in which chromatin accessibility is high (i.e. low nucleosome occupancy) and condensability is high, is suspiciously empty of any datapoints. This is critical, as it means that even if the supernatant was just a random sample from the input, we would still get a correlation between the condensability score and nucleosome occupancy (and A/B compartments). Ultimately, it will be important to understand whether any condensability score is correlated with input MNase occupancy, as this alone can drive correlation with A/B compartments.

//We thank the reviewer for this suggestion to address a very important issue that we failed to mention. Regarding the quantification of the condensability score, we apologize if this was not clear in the original submission. We defined the condensability as $-\log(\text{nucleosome counts in supernatant divided by nucleosome counts in input control})$ or $-\log(\text{survival probability of nucleosome in supernatant after condensation})$. Because this survival probability is between 0 and 1 ($0 < P < 1$), $-\log P$ should be positive and not bounded. We have better clarified the quantification of condensability in the resubmission. In the new analysis (new Extended Figure 2) in the revised manuscript, the input nucleosome coverage across the genome is uniform but the supernatant nucleosome coverage after condensation is getting more biased at the higher spermine concentration (new Extended Data Figure 2 d). In addition, nucleosome number in supernatant is much more correlated with the condensability score in single-nucleosome level (new Extended Figure 2 f) and in ChromHMM states (new Extended Data Figure 2 g) compared to the input nucleosome number. All these results strongly indicate that the condensability contrast is primarily driven by how many nucleosomes remain in the supernatant after condensation, instead of input nucleosome occupancy. We added the following to the main text.

“We also validated that our condensability metric is indeed a measure tightly associated with how many nucleosomes survived in supernatant after

condensation by showing that the nucleosome counts in the supernatant, not those of the input, are primarily responsible for the condensability contrast (Extended Data Fig. 2 d-g).”

2. More generally, this just one example of how the analysis details, and specifically the calculation of condensability, are extremely important and must be justified and done correctly. What happens to zero values? How are missing values treated? Why are nucleosomes peaks “called” and what are the effects of this on the data? What is the justification for an additional condensability score($C_{1/2}$)? Currently, these details are not provided and each of these may result in an artificial correlation with other genomic features such as A/B compartments. Details are also missing from the methods in other sections (especially analysis), e.g. how were machine learning hyperparameters fit, why only 100000 nucleosomes were used, etc.

//We thank the reviewer for giving us the opportunity to improve the manuscript in terms of rigor of analysis and presentation. Since we used input nucleosome occupancy for peak calling nucleosome positions, input nucleosome counts are always larger than zero for each nucleosome peak and this is rationale behind for doing peak calling to focus on genomic locations having more nucleosomes as input for fair testing their condensability. To avoid taking the ratio of zero counts at supernatant over input, we added one pseudo-count to each supernatant and input read counts before estimating nucleosome numbers and condensability computation. We also tested the sliding window method (171bp window and 25 bp step) to compute condensability scores of each genomic bins, and peak calling method and sliding window approach gave almost identical result in the ChromHMM genome segmentation (new Extended Data Figure 2 h). This result implies that condensability score computation is not very sensitive to the choice of peak calling or genomic binning method, probably because nucleosomes are almost everywhere across the genome. The condensability point ($c_{1/2}$) was computed using multiple condensation data in different spermine concentrations, which could be more robust than the previous condensability score metric, the negative log of the survival probability at a single spermine concentration point, but this new metric should be only complementary and indeed it has inverse relationship with the original condensability score metric as shown in the scatter plot (new Extended Data Figure 10 i), which indicates the self-consistency between condensability point ($c_{1/2}$) and condensability score metrics. Finally, we have updated the method section of the revised manuscript for more detailed information about machine learning training parameters and other analysis details. We also provided open source IPython scripts of all our data analysis in our GitHub repository

(<https://github.com/spark159/condense-seq>). We have added the following relevant statements to the main text and Materials & Methods.

“We also checked the reproducibility and robustness against the choice of nucleosome peak calling methods (Extended Data Fig. 2e,j,h).”

“For a quantitative analysis of subtle differences across different conditions, we used another metric, condensation point ($c_{1/2}$), a spermine concentration in which the soluble fraction is half the input (Extended Data Fig. 10b). Thus, $c_{1/2}$ is inversely correlated with the previously defined condensability score (Extended Data Fig. 10i).”

“To avoid taking log of zero values, we added one pseudo-count to each input and supernatant reads counts during the condensability calculation.”

3. What is actually being measured in condense-seq? Is it actually the propensity of specific nucleosomes to precipitate? The authors take extra care to isolate mononucleosomes, which is good. But DNA can also precipitate with polyamines, and couldn't the nucleosome potentially unwrap? Perhaps crosslinking the nucleosomes to the DNA could reduce the chances of this happening.

//We thank the reviewer for raising these important points. We are also glad to learn that the reviewer appreciates the fact that we took efforts to isolate mono-nucleosomes to avoid confounding factors coming from oligo-nucleosomes. We have performed additional experiments to test if (a) naked DNA is the main species that condenses with polyamines and (b) DNA is unwrapped from the histone core.

(a) We tested the possibility that when nucleosomes condense, DNA molecules depart from the histone core so that what's spun down during the protocol is primarily naked DNA molecules (new Extended Data Figure 1f-h). When we redissolved the condensed fraction and ran a gel, we could show that most of the molecules show up as intact mono-nucleosomes (new Extended Data Figure 1h). This was true for native nucleosomes from mESCs and GM12878 as well as reconstituted nucleosomes made from GM12878 nucleosomal DNA and unmodified histone octamers. Therefore, we conclude that each nucleosome maintains its mono-nucleosome status even when it becomes part of a condensate in the presence of other nucleosomes.

(b) We performed single molecule FRET experiments using a nucleosome labeled with a donor and an acceptor so that FRET decreases report on DNA unwrapping from histone core (new Extended Data Figure 1f). For the concentrations used for condense-seq (up to 2 mM), spermine did not induce any DNA unwrapping. As a separate control, we tested that 0.4 mM spermine induces large assemblies of mononucleosomes that can be detected via total internal reflection fluorescence microscopy (new Extended Data Figure 1g).

We have added the following to the main text.

“We showed that the native mono-nucleosomes remain intact upon condensation and used single molecule FRET to show that spermine, at concentrations that induce formation of large nucleosome condensates, does not induce significant unwrapping of nucleosomal DNA (Extended Data Fig. 1).”

4. The authors make conclusions based on differences between groups, but no statistical tests were performed on the results in the paper. It is also not mentioned what error bars represent (SD/SE/CI). In other cases, the authors argue that data are similar/correlated but do not provide a proper quantification (e.g. condensability vs A/B compartments).

//We thank the reviewer for this important point. We provided the information of error bars in the corresponding figure captions (most of them are standard deviations). We also provided the numerical correlation, and the statistical significance (p-value) as well as the effect-size (Cohen's d) for all comparative analysis in the revised manuscript.

5. The fact that using condensability score as pairwise energy can reconstruct genomic compartment patterns in interaction maps is expected given the previous work in polymer models of genome organization, it was shown several times that having some energy which favours self-interaction can create these patterns.

//We agree with the reviewer that previous polymer simulations have demonstrated the ability to reproduce compartment patterns using block copolymer models. In these previous models, chromatin is typically divided into a few distinct types, with favorable interaction energies assigned between chromatin beads of the same type. However, a significant distinction—and a key contribution of our study—lies in the interpretation of these interaction energies. To our knowledge, prior studies have not explicitly addressed the physical origin of these energies. This lack of clarity has introduced considerable confusion in the field, with various hypotheses proposed to explain the

molecular factors underlying the interaction energies. These include physical interactions mediated by chromatin regulators that recognize histone modifications [PMID: 28636604, 28636597], DNA-RNA interactions [34739832], and non-equilibrium processes associated with transcription [33649325].

Importantly, most previous studies have overlooked the intrinsic physicochemical interactions between nucleosomes, which we have measured in our work. These interactions can vary among nucleosomes due to differences in DNA sequence and histone modifications. Our findings are noteworthy in that these intrinsic interactions, without accounting for additional *in vivo* complexities, can fully explain the compartment patterns observed.

Thus, our study presents a novel conclusion: genome organization is encoded by the physicochemical properties of nucleosomes. To our knowledge, no other study provides this level of support for such a claim.

6. The condensability measurements on the PTM library are interesting, but they do not seem to explain well the notion that A/B compartments are mostly driven by condensability: why do A-specific PTMs have similar condensability to B-specific PTMs? Ideally, we would like to find some model that could roughly predict the condensability of *in vivo* nucleosomes from these synthetic scores. Could DNA sequence play a role as well? One way to learn about this might be to perform condense-seq on nucleosomes (without PTMs) that were reconstituted *in vitro* on genomic DNA. At this stage I think the PTM data still does not sufficiently support the hypothesis that this is what drives compartmentalization.

//We thank the reviewer for the comment and suggestion. In response, we performed the condense-seq of reconstituted nucleosome composed of genomic nucleosomal DNA from human lymphocytes (GM12878) and recombinant canonical histone octamer. (included as new Extended Data Fig. 7). We purified mono-nucleosomal genomic DNA from GM12878 cells following MNase treatment and reconstituted the nucleosomes using recombinant PTM-free histone octamers (Extended Data Figure 7a). After extensive optimization of this new protocol, we applied several size-selection steps to obtain pure reconstituted nucleosome products (Extended Data Figure 7 b,c). Our analysis revealed that reconstituted nucleosomes composed of genomic DNA but completely free of PTMs exhibited a dramatic loss of condensability changes associated with chromatin states

(Extended Figure 7d), correlations with gene expression (Extended Figure 7f), and genome-wide fluctuations (Extended Figure 7e). These effects were particularly pronounced when compared to native nucleosomes from GM12878 cells (also presented in the new Extended Data Fig. 7). This data clearly indicates that histone PTMs are among the most critical factors in determining the condensability of native genomic nucleosomes. To further consolidate this idea, we built a simple linear regression model only trained with synthetic PTM library condensability data and successfully predicted genomic nucleosome condensability at least qualitatively. The results are included in our revised manuscript (new Extended Data Fig. 6d-f). We added the following to the main text.

“Next, to examine the effects of genomic DNA sequences on nucleosome condensation, we synthesized a ‘reconstituted’ nucleosome library composed of the genomic nucleosomal DNA purified from GM12878 cells reconstituted with recombinant canonical histone octamers which are devoid of PTMs (Extended Data Fig. 7a,b). Remarkably, the reconstituted nucleosomes show higher condensability overall compared to native nucleosomes (Extended Data Fig. 7c) and lost the chromatin state dependence (Extended Data Fig. 7d). They also lost the correlation with gene expression in genome-wide scale (Extended Data Fig. 7e) and individual genes near TSS (Extended Data Fig 7f). These results show the primary importance of histone PTMs for determining genomic nucleosome condensability.”

“A linear regression model trained with only the PTM library condensability data could qualitatively predict genomic nucleosome condensability (Extended Data Fig. 6d-f).”

7. The measurements in mice attempt to show biological relevance of condensability, which is potentially a good idea, but the results are somewhat counterintuitive. If polyamines drive condensability and A/B compartments, naively one would expect that their removal would cause less compartmentalization and condensability (by the way, Hi-C was not measured here so we do not know how genome organization changed). It is possible that the cell tries to compensate for this effect as the authors suggest, but there is no direct evidence for this here. I believe that currently the results of this experiment do not provide strong support for the authors’ hypothesis.

//We realized that we did not explain our logic clearly in our original submission. Condensability is a biophysical property of a nucleosome itself. Because polyamines induce attractive interactions between nucleosomes, cells can rely on endogenous

polyamines to bring together more condensable nucleosomes to form B compartments. When polyamines are depleted, i.e. lower in concentration, a higher condensability contrast would be needed to achieve the same average degree of attractive interactions, leading to hyperpolarization. That is, nucleosomes with biophysical properties associated with high condensability acquire changes to make the condensability even higher and vice versa. Our data indeed show hyperpolarization upon polyamine depletion achieved in two complementary manners. We rewrote the section as follows to make this point clearer.

“We suggest that when cells cannot rely on endogenous polyamines to bring together more condensable nucleosomes to form B compartments to induce promoter condensation, they modify the nucleosomes to accentuate the condensability contrast. That is, upon polyamine depletion, nucleosomes with biophysical properties associated with high condensability acquire changes to make the condensability even higher and vice versa.”

8. As I mentioned, I suggest stating the hypotheses and claims more accurately so they would be easier for the reader to understand. It may also be appropriate to tone down some of the more speculative conclusions such as “thus, electrostatic interaction between nucleosomes mediated by multivalent ions is probably the main driving force behind large-scale genomic compartmentalization”.

//This point is well taken. We have rephrased our statements as follows.

“Intriguingly, the charge-swap mutations on the acidic patch of histone H2A/B, which was previously suggested to be the nucleosome–nucleosome interaction interface, induced the largest condensability increase among the PTM library members for all ionic condensing agents (Fig 3e). Thus, this trend, combined with our observation that polymer simulations using nucleosome condensability as the sole input can predict A/B compartments (Fig. 2), further points to the electrostatic interaction between nucleosomes mediated by multivalent ions as a major driving force behind large-scale genomic compartmentalization (see Supplementary Note 4 for additional discussion).”

Minor comments:

9. Extended Figure 5C is not really explained in the main text.

//We have added its mention in the revised manuscript (new Extended Data Fig 4e,f) as follows.

“Indeed, when AT-rich DNA and GC-rich DNA are co-condensed in the presence of spermine, they spontaneously formed a spatially segregated structure, where AT-rich DNA core is surrounded with GC-rich DNA, likely through their differential condensabilities²⁷ (Extended Data Fig. 4e,f).”

10. Why doesn't HP1 condensation show AT content as a strong factor like spermine? This seems counterintuitive.

//HP1 being a basic protein, it may work to a certain degree like a multivalent cation. In fact, we find that PTM condense-seq data also show that HP1 condenses most PTM-containing nucleosomes in a similar direction as polyamines except for H3K9me3 (Fig 3f and new Extended Data Fig 9d). As noted by the reviewer, AT content does not appear to be important for HP1. The reason is presently unknown, but one possibility is that the strong AT content dependence is polyamine specific. In a previous publication (Yoo et al, 2016, PMID 27001929), we proposed that methyl groups of thymine act as a steric block, relocating spermine from major grooves to interhelical regions, thereby increasing DNA-DNA attraction.

11. “the correlation was biphasic due to the sparsity of accessible regions revealed by ATAC-seq” – this was unclear to me.

//We have revised the text as follows.

“Genomic accessibility measured by ATAC-seq also shows an anti-correlation with condensability, in which more accessible/opened genomic regions are less condensable, and vice versa (Fig. 2e). This inverse relationship between chromatin openness and condensability is even more pronounced when compared across chromatin states (Fig. 2f and Extended Data Fig. 4c,d).”

12. The text should be checked for typos (e.g. “chromatins”, “multilayer perception”).

//Fixed.

Referee #3 (Remarks on code availability):

I reviewed the code only briefly, but I found the documentation to be lacking. I was hoping that the analysis details explaining how things were calculated - which did not appear in the Methods section - would at least be explained in the code documentation but this was not the case. I did not try to run the code myself.

//We thank the reviewer for this feedback. We have fully documented all analysis pipelines and also generated IPython notebooks so that a reader can start with the raw data and create all of the plots included in the manuscript. All scripts and analysis details could be found in our GitHub repository (<https://github.com/spark159/condense-seq>).

Referee #4 (Remarks to the Author):

The manuscript by Park et al. reports a new deep sequencing method they refer to as condense-seq. Briefly, their approach as I understand it is to use micrococcal nuclease to digest chromatin from cell lines down to single nucleosomes and then purify them. They condense the single nucleosomes with different reagents, then centrifuge the sample and subject the DNA in the pellet to deep sequencing. This allows them to map the propensity of genomic regions to condense in the form of single nucleosomes. The authors refer to this as the intrinsic condensability of single nucleosomes. They compare the condensability to a wide range of published mapped features. They find that the reduction in condensability correlates with active genomic regions including promoters and enhancers, while increase condensability correlates with silenced genomic regions such as heterochromatin and insulators. Accessibility as measured by ATACseq also correlates with the reduction in accessibility as well as A/B compartmentalization as measured by HiC. To better understand how histone PTMs influence accessibility they used a nucleosome library of different histone PTM combinations and determined that histone acetylation largely determines condensability. However, other PTMs such as ubiquitylation significantly reduces condensability as well. They then investigated the impact of disrupting the expression of polyamines on condensability since polyamines help with global condensation of chromatin. They find that the range of condensability is significantly increased when polyamines are not expressed, which indicates that the cell responds to this mutation by increasing nucleosome condensability to compensate for the loss of polyamines.

Overall, this is an extensive study that provides strong evidence that their method condense-seq maps the genomes propensity to condense, which will certainly be another impactful deep sequence method for genomic characterization. The method is synergistic with Chip-seq, ATAC-seq, HiC, and other genome mapping techniques. It is also interesting how different compacting reagents can be used, which could be used to characterize condensability by different distinct chromatin architectural proteins and small molecules. This is an important new and elegant method that will likely be impactful and useful for studying different cell types, mutations, and disease. However, there are issues with the presentation of the work and their conclusions, which I list below.

//We thank the reviewer for a concise summary of our advances and findings, and for highlighting the significance of the work.

1) A primary concern is the authors repeatedly make bold and general conclusions in the

paper. Here are some examples:

a. These findings are surprising because they indicate that single native nucleosomes 'know' if they are in highly transcribed regions or gene promoters by way of their reduced condensability and vice versa.

b. Overall, our results imply that the native mono-nucleosomes intrinsically possess, even in the absence of other factors, essentially all the biophysical properties needed for the large-scale A/B compartmentalization of the eukaryote genome.

c. Thus, electrostatic interaction between nucleosomes mediated by multivalent ions is probably the main driving force behind large-scale genomic compartmentalization.

These are bold conclusions and implicit to these conclusions is that there is a cause relationship between the mononucleosome condensability and other features such as transcription level and compartmentalization. However, based on my understanding of these studies, the results only provide correlations between these features. Also, it is possible that other factors that are driving chromatin compaction then influence histone acetylation. For example, the linker histone H1 can influence the level of histone acetylation (PMID: 10611231). The authors need to adjust the text to reflect that they have not shown that electrostatic driven condensation is the driver of these genomic features, but instead are correlated with them.

//We thank the reviewer for pointing out several statements that may appear to be bold. We have revised them as follows.

“These findings are surprising because they indicate that single native nucleosomes isolated from the cell possess biophysical properties, high or low condensability that are associated with low and high transcription, respectively, even though the condensability was determined in vitro in the absence any other factors normally present in vivo.”

“Overall, our results imply that the native mono-nucleosomes intrinsically possess, even in the absence of other factors, much of the biophysical properties needed for the large-scale A/B compartmentalization (~80% in the case of GM12878 cells).”

“Thus, this trend, combined with our observation that polymer simulations using nucleosome condensability as the sole input can predict A/B compartments (Fig. 2), further points to the electrostatic interaction between nucleosomes mediated by multivalent ions as a major driving force behind large-scale genomic compartmentalization (see Supplementary Note 4 for additional discussion).”

We would also like to point out that although H1 may influence histone acetylation, we can reproduce the A/B compartments in the Hi C experiments from the condense-seq data alone (with ~80% accuracy) so at least at the length scale of compartments, most of the chromosomal organization can be predicted from nucleosome level interactions.

Finally, we agree with the reviewer that we cannot claim causality and revised the text to avoid giving such an impression. Our main conclusion is that single native mono-nucleosomes contain biophysical properties that correlate with A/B compartments and gene expression, and these are intrinsic properties found within single nucleosomes, i.e. in the absence of other factors or even physical connectivity between the nucleosomes.

2) Another concern, which is connected to the first bullet point, is that there are numerous papers that have shown that histone acetylation influences chromatin compaction. So, it is not that surprising that electrostatics and histone acetylation is a key determinant of mononucleosome condensability. In fact, previous liquid crystal and TEM studies have shown that mononucleosomes compact similarly to that of nucleosome arrays, where mononucleosomes make long columnar structures and are packed face to face (PMID: 16973479). This is most likely due to the H4 tail interacting with the acidic patch of an adjacent nucleosome. Therefore, these condense-seq studies nicely confirm previous studies that have shown electrostatics is key to condensation and that the neutralization of positive charge by acetylation reduces the propensity of chromatin to compact. The text should be adjusted to reflect better what is known about how histone acetylation influences chromatin and nucleosome compaction.

//We thank the reviewer for this comment. We have revised the text as follows.

“Although the nucleosome core particle (NCP), lacking DNA linker connecting nucleosomes in chromatin fiber, appears to contain sufficient information for large-scale genomic compartmentalization and electrostatics can drive compaction of NCPs similar to that of nucleosome arrays (PMID:16973479), we do not neglect the possibility that the linker DNA must play an important role for genome organization through modulation of nucleosome spacing (PMID: 30630950), synergizing with the intrinsic condensabilities of individual NCPs.”

“Electrostatic interaction is a key determinant as shown by the strong impact of acetylation and crotonylation which add negative charges that would require more polyamines to neutralize the net negative charged nucleosomes during

condensation. Acetylation on histone tails has a much stronger effect than acetylation on the histone fold domain (Fig. 3d), having the strongest effect on the H4 tail, followed by the H2A, H2B, and H3 tails, respectively.”

3) The authors should consider alternative interpretations on their HP1 measurements where they find that HP1 has a different and smaller impact on condensability than spermine. HP1 has been shown to compact chromatin through interactions with adjacent nucleosomes that contain H3K9me3 (PMID: 32493764). The experiments with mononucleosomes will largely prevent the assembly of HP1 compacted chromatin because there are not chains of nucleosomes with H3K9me3. This should be added and expanded on in the manuscript.

//HP1, being a basic protein, may act as a multivalent cation as polyamines but not as effectively, explaining the more moderate contrast we observed. We would like to note, however, that we do not need physical connectivity between mononucleosomes to observe H3K9me3-dependent condensation mediated by HP1. This is most clearly shown in the PTM library data where H3K9me3 nucleosomes show greatly elevated condensability in HP1 condense-seq. Even in the native nucleosome HP1 condense-seq experiments, we observed that H3K9me3 is the best predictor of high condensability. We have revised the text as follows to highlight these points.

“We also performed condense-seq of the PTM library using HP1 α as the condensing agent. H3K9me3 profoundly increases nucleosome condensation by HP1 α (Extended Data Fig. 9d, Fig. 3f), which is consistent with HP1 α 's role as an H3K9me3 heterochromatin mark reader.”

“Heterochromatin, nucleolar, and lamina-associated marks show a positive correlation with condensability, with the strongest correlation being observed between HP1-mediated condensability and the H3K9me3 marks.”

4) The authors should address how compaction of single nucleosomes is different than the compaction of chromatin where nucleosomes are contained within a single fiber. For example, it is known that the DNA repeat spacing between nucleosome significantly impacts chromatin compaction (PMID: 30630950). The authors should address in the manuscript that nucleosome spacing is another regulator of chromatin compaction, which could function synergistically and antagonistically with histone acetylation.

//This is an excellent point. We added the following to Discussion to address this point.

“Although the nucleosome core particle (NCP), lacking DNA linker connecting nucleosomes in chromatin fiber, appears to contain sufficient information for large-scale genomic compartmentalization and electrostatics can drive compaction of NCPs similar to that of nucleosome arrays (PMID:16973479), we do not neglect the possibility that the linker DNA must play an important role for genome organization through modulation of nucleosome spacing (PMID: 30630950), synergizing with the intrinsic condensabilities of individual NCPs.”

5) It is interesting they have evidence that methylation might directly influence compaction. In fact, methylation on H4K20 has been shown to directly induce decompaction within nucleosome arrays (PMID: 34417450) by disrupting the H4 tail conformation and thus its interaction with the acidic patch. However, they do not seem to observe reduced compaction by spermine with the ENCODE mapped H4K20me1 PTM. This suggests direct impact of this methylation might impact condensation within chains of nucleosomes but not within isolated mononucleosomes. Both this example and the mono vs. polynucleosomes should be discussed in the manuscript.

//We thank the reviewer for this important point. In fact, we do not expect condensability to have an explanatory power for all histone modifications' effect on chromatin organization. The noted effect of H4K20 methylation may indeed be specific to oligonucleosomes as shown by the reference noted by the reviewer. We added the following statement to Discussion.

“For example, the very small reduction in condensability we observed for NCPs with H4K20me1 (Extended Data Fig. 6D), a modification known to induce decompaction in nucleosome arrays (PMID: 34417450), suggests that some histone modifications may primarily impact condensation within arrays.”

In addition to these major issues, I list below some minor issues.

1) In many of the main and supplemental figures the text is very small and nearly impossible to read. Please improve this. I found it very frustrating when I was trying to read the text in the figures.

//We thank the reviewer for this feedback and sorry for the inconvenience since we had to smooch lots of key data into the format. We reorganized the figures and re-scaled them to improve readability.

2) Figure caption 1, has H3K25ac in the text. I assume they mean H3K27ac.

//Thank you. This has been fixed.

Additional feedback:

1) One natural extension of these studies is to carry out condense-seq with dinucleosomes, trinucleosomes, etc. It would be extremely interesting to compare condensability of different repeat numbers. I am not suggesting this be done for this paper, which already contains a tremendous number of results. But this work provides the foundation for further important studies of chromatin compaction. This is another reason why the authors should adjust their conclusions of the single nucleosome studies. There is tremendous potential for using this method of studying short sections of chromatin.

//This is a very good point, and it will indeed be informative to perform condense-seq using dinucleosomes. Presently, this will be a very challenging experiment because we need to purify dinucleosomes from other species and because heterogeneous spacing between nucleosomes will make it difficult to obtain purely dinucleosome preparations etc. We added the following statements to state the limitations of our study as we discussed in response to earlier comments by the reviewer.

“Although the nucleosome core particle (NCP), lacking DNA linker connecting nucleosomes in chromatin fiber, appears to contain sufficient information for large-scale genomic compartmentalization and electrostatics can drive compaction of NCPs similar to that of nucleosome arrays (PMID:16973479), we do not neglect the possibility that the linker DNA must play an important role for genome organization through modulation of nucleosome spacing (PMID: 30630950), synergizing with the intrinsic condensabilities of individual NCPs. For example, the very small reduction in condensability we observed for NCPs with H4K20me1 (Extended Data Fig. 6D), a modification known to induce decompaction in nucleosome arrays (PMID: 34417450), suggests that some histone modifications may primarily impact condensation within arrays.”

2) I am surprised the authors do not map the positions of nucleosomes from their data. It would be interesting to learn how the condensed nucleosomes are positioned relative to input and the soluble fraction. I don't think this needs to be done. But it would be interesting to investigate and should come along without the need for additional experiments.

//We agree that this will be an excellent topic for future investigations. The raw data we provide may also be segmented and analyzed by interested researchers as the first step.

Rebuttal letter

Reviewer 3:

I mentioned that the condensability scores are constrained by the input levels and the authors thought I misunderstood something, but I still think I am correct here: condensability is defined by the authors as $-\log(\text{nucleosome counts in supernatant} / \text{nucleosome counts in input control})$, and they say the ratio is always between 0 and 1, so the condensability is always positive. First, note that Ext. Data Fig. 2b (which explains how condensability is calculated) shows a negative score, but let's assume this is a typo. If the ratio is always between 0 and 1, it means that supernatant counts cannot be higher than input counts. Now let's consider a case in which there is low input, say only 1 read. Now, the supernatant can have 0 or 1 reads, and after adding the pseudocounts we get that the condensability in this case can be $-\log((0+1)/(1+1))$ or $-\log((1+1)/(1+1))$. In this case, $\log(2)$ is the highest condensability value that can be achieved when the number of input reads is 1. Similarly, when the input counts is 2, the highest condensability is $-\log((0+1)/(2+1))$, and so on. This means that when the input is low, you can't get high condensability values, and this creates an artificial correlation which could seriously bias the data as I explained previously.

To save us all time in back-and-forth, I took the liberty of performing the following small simulation: I randomly generated 10,000 input nucleosome read count values, and then randomly generated 10,000 supernatant values where the only constraint is that they cannot be higher than the input values. I then calculated the nucleosome condensability scores, and then the Pearson correlation between the input and the condensability. The resulting correlation was around $r = \sim 0.1$. Given these results, which I think are in line with the correlation the authors observe with the input, I think this concern can be put aside. However, it would be useful to add this type of analysis to the paper in case other readers are also concerned about how strongly this artifact can bias the condensability score.

Response to Reviewer 3

They pointed out a potential typo in our Extended Data Figure and we fixed it. They also mentioned their own simulations that showed their earlier concern is unlikely to be valid. We have already addressed this concern in our previous revision.

Reviewer 4:

I will not repeat the summary provided. However, I think it is worth pointing out that there are two major outcomes of this work. The first outcome is that native mononucleosomes can be condensed into A and B compartments without additional factors. This reveals that features within single nucleosomes play an important role in

genome organization. I recommend the authors consider pointing out that their observation is synergistic with SAXS and super-resolution imaging that indicates that the 30 nm chromatin fiber does not form in cells since they find that connectivity is not critical for heterochromatin-associated nucleosomes to condense more readily than euchromatin-associated nucleosomes. The second aspect is that Condense-seq could develop into an important technique for genome profiling since it appears to be a straightforward sequencing technique. My sense is that Condense-seq is synergistic with other genome organization methods, including Hi-C, MNase-seq, and ATAC-seq. This aspect is not discussed much and could be emphasized more.

One final note is that the authors state it would be difficult to purify di- and tri-nucleosomes. Sucrose gradients should work well for purifying mono-, di-, and tri-nucleosomes. They would just need to optimize the MNase digestion. The authors might consider trying this in their future studies.

Response to Reviewer 4

In response to the reviewer's comments, we added the following sentences to the Results and Discussion sections.

"We envision condense-seq as a readily adoptable tool for studying functional genome organization in a variety of contexts."

"By finding that connectivity is not critical for heterochromatin-associated nucleosomes to condense more readily than euchromatin-associated nucleosomes, our data are synergistic with studies that showed that 30 nm fibers do not form in cells."